# FAITHFUL AND EFFICIENT EXPLANATIONS FOR NEURAL NETWORKS VIA NEURAL TANGENT KERNEL SURROGATE MODELS

Andrew Engel[1]     Zhichao Wang[2]     Natalie S. Frank[3]     Ioana Dumitriu[2]
Sutanay Choudhury[1]     Anand Sarwate[4]     Tony Chiang[1,5,6]

[1]Pacific Northwest National Laboratory     [2]University of California, San Diego
[3]Courant Institute, NYU     [4]Rutgers University     [5]University of Washington     [6]University of Texas, El Paso
`{andrew.engel,sutanay.choudhury,tony.chiang}@pnnl.gov;`
`{zhw036,idumitriu}@ucsd.edu;`
`nf1066@nyu.edu; ads221@soe.rutgers.edu`

## ABSTRACT

A recent trend in explainable AI research has focused on surrogate modeling, where neural networks are approximated as simpler ML algorithms such as kernel machines. A second trend has been to utilize kernel functions in various explain-by-example or data attribution tasks. In this work, we combine these two trends to analyze approximate empirical neural tangent kernels (eNTK) for data attribution. Approximation is critical for eNTK analysis due to the high computational cost to compute the eNTK. We define new approximate eNTK and perform novel analysis on how well the resulting kernel machine surrogate models correlate with the underlying neural network. We introduce two new random projection variants of approximate eNTK which allow users to tune the time and memory complexity of their calculation. We conclude that kernel machines using approximate neural tangent kernel as the kernel function are effective surrogate models, with the introduced trace NTK the most consistent performer. Open source software allowing users to efficiently calculate kernel functions in the PyTorch framework is available here[*].

## 1 INTRODUCTION

Explainability remains a critical open problem for applications of deep neural networks (NNs) (Leavitt & Morcos, 2020). Explain-by-example techniques (Lai et al., 2021; Yang et al., 2020) have emerged as a major category of algorithms for explainability, including prototype examples (Chen et al., 2019), Deep K-Nearest Neighbors (Papernot & McDaniel, 2018; Wang et al., 2021; Dziedzic et al., 2022), and Representer Points (Yeh et al., 2018; Tsai et al., 2023). These techniques explain models by providing example(s) that capture model behavior on new data. Kernel functions (Alvarez et al., 2011) are a natural choice for building explain-by-example algorithms (Yeh et al., 2018); a kernel measures the similarity between individual data points via an inner product in a reproducing kernel Hilbert space (RKHS) (Hilbert, 1912; Ghojogh et al., 2021). A RKHS that faithfully represents a linearized NN feature space can be used in a kernel machine to explain (model) the NN decision as a weighted sum of similarities to training data.

In this work, we investigate computationally efficient approximations to the empirical neural tangent kernel (eNTK), which is a kernel function motivated by advances in the theory of deep learning (Jacot et al., 2018). It is well established that NNs trained using gradient descent are equivalent to kernel machines (Schölkopf & Smola, 2002) with a kernel constructed from a sum over eNTK (Lee et al., 2020) computed at each gradient step (Domingos, 2020; Bell et al., 2023). Given this equivalence, we would like to evaluate the eNTK as the kernel function for an explain-by-example algorithm; however, computing eNTK is computationally expensive (Novak et al., 2022; Chen et al., 2022),

---

[*]https://github.com/pnnl/projection_ntk

so low computational cost approximations have been developed instead (Mohamadi & Sutherland, 2022). We are the first to define and evaluate one such approximate kernel, the trace neural tangent kernel (trNTK). Additionally, we build from the work of Park et al. (2023) to provide software to compute random-projection variants that can be computed and stored with lower time and memory cost over traditional eNTK. Using these approximations, we build low-cost and faithful surrogate models for neural network classifiers.

Our methodology improves over the past evaluation of kernel surrogate models. We measure the faithfulness of a kernel function by assessing how well a kernel generalized linear model (kGLM) (Hofmann et al., 2007) correlates with the softmax probabilities of the original NN using a rank correlation. Previous evaluations relied on test accuracy (Mohamadi & Sutherland, 2022; Long, 2021), or having high similarity to the correct class (Hanawa et al., 2021), which are both flawed. Our approach and accompanying code-repository will allow users to evaluate how close their own NNs are to kernel machines in the PyTorch framework with limited overhead (Paszke et al., 2019).

### CONTRIBUTIONS

We make three major contributions in this work:

1. We define and evaluate new kernel functions for faithful approximation of an underlying neural network; we are the first to analyze random projection variants that permit tuning the computational and memory expense of approximate eNTK.

2. We are the first to show that approximate eNTK kernel surrogate models are consistently correlated to the underlying neural network across experiments including ResNet18 on CIFAR10 and Bert-base on COLA.

3. We compare explanations of NN decisions generated from each kernel function through a data attribution strategy and through an explain-by-example strategy; this is the first such qualitative evaluation between approximate eNTK.

### RELATED WORK

**Surrogate Models for Explaining Neural Network Behavior.** Recent work in explainable AI has focused on determining when NNs are exactly equivalent to other common ML algorithms (Lee et al., 2018; Balestriero & Baraniuk, 2018; Schmitz et al., 1999), including kernel machines. It has been shown that infinitely wide NNs are equivalent to a kernel machine with kernel function chosen as the neural tangent kernel (Jacot et al., 2018). These infinitely wide models, however, do not replicate the feature learning behavior seen in finite-width networks (Chizat et al., 2018; Yang & Hu, 2021; Wang et al., 2022). Subsequently, researchers turned to investigate properties of finite-width models with NTK computed at various checkpoints (Domingos, 2020; Bell et al., 2023) and/or after training (Long, 2021). This framework was used to explore inductive biases (Ortiz-Jiménez et al., 2021), feature learning (Radhakrishnan et al., 2022), learning dynamics (Fort et al., 2020; Atanasov et al., 2022), and adversarial faithfulness (Tsilivis & Kempe, 2023; Loo et al., 2022). Support vector machines (Vapnik, 1999) using eNTK or approximate eNTK kernels computed after training were shown to achieve the same test accuracy as the underlying NN (Atanasov et al., 2022; Long, 2021; Vyas et al., 2022; Mohamadi & Sutherland, 2022). Our work builds upon this by evaluating whether kernel machines can approximate the underlying neural network function itself, rather than simply reproduce the same test accuracy.

**Kernels for Explainability.** Kernel functions defined from various RKHS have been proposed to explain the behavior of NN in different contexts, (Park et al., 2023; Koh & Liang, 2017; Pruthi et al., 2020; Akyürek et al., 2023), but in each of these works the kernel studied is loss-based and relies upon the availability of labels at inference time. We differ in that our goal is to model/explain the classification behavior on any new data, including unlabeled data where the loss is incalculable. Most relevant to our work, Yeh et al. (2018) (hereafter Representer Points) used a kernel formed from the NN final embedding in what we call the data attribution task (see section 2). We build from Representer Points by evaluating their assumptions under new approximate eNTK kernels.

**Computationally Feasible Approximations of the eNTK** The computational cost of the eNTK is prohibitively high for large models and datasets. Advances on this issue have been two-pronged:

Some groups focus on algorithmic improvements to calculate the eNTK directly (Novak et al., 2022). An alternative strategy has been to avoid eNTK calculation and instead compute kernel functions that share a similar structure to the eNTK (Mohamadi & Sutherland, 2022). One such approximate kernel was introduced quietly in Chen et al. (2022) which we refer to as the trace-NTK ($\mathrm{trNTK}$). We are the first to explicitly investigate the $\mathrm{trNTK}$'s properties. Finally, Park et al. (2023), hereafter TRAK, utilized random projection matrices to scale the computation of a loss-based kernel function. We modify TRAK to compute projected variants of approximate eNTK.

**Evaluating Kernel Attribution.** In this paper, we use three evaluation strategies. The first focuses on evaluating the faithfulness of the surrogate model through rank correlation. The second evaluates surrogate model performance on a data-attribution task. We follow the methodology in Shan et al. (2022) to evaluate the model via precision and recall in tracing decisions on poisoned test data back to poisoned training data. Finally, we compare kernels qualitatively via explain-by-example. Previous work evaluated kernels through whether the attributions trace to training data of the correct class (Hanawa et al., 2021), whether surrogate models replicate NN test accuracy (Mohamadi & Sutherland, 2022; Long, 2021). These are insufficient: our goal is that kernel functions reflect the neural network behavior, but test accuracy is invariant to the specific classification on individual datapoints. Representer Points used Pearson correlation as a faithfulness measure, but Pearson correlation can conflate covariance with faithfulness (see Appendix H). We will demonstrate that our methodology is more secure measurement of faithfulness.

## 2 PRELIMINARIES

**Neural Networks for Classification.** We consider the supervised classification problem with $C$ classes. Consider a data input $\boldsymbol{x} \in \mathcal{X} \subseteq \mathbb{R}^n$ with $n$ the dimensionality of inputs, and a one-hot encoded data label vector $\boldsymbol{z} \in \mathcal{Z} \subseteq \mathbb{R}^C$. We define a neural network $F(\boldsymbol{x}\,;\boldsymbol{\theta}) : \mathcal{X} \to \mathcal{Y}$ where the output space $\mathcal{Y} \subseteq \mathbb{R}^C$ is an intermediary step in our classification called a "logit." The NN $F(\boldsymbol{x}\,;\boldsymbol{\theta})$ is parameterized by the vector $\boldsymbol{\theta}$ and was learned via back-propagation to minimize the cross entropy loss between the target label vector $\boldsymbol{z}$ and softmax probability vector $\sigma(F(\boldsymbol{x}\,;\boldsymbol{\theta}))$, with $\sigma : \mathcal{Y} \to \mathcal{Z}$ the softmax function. We denote the $c$-th scalar output of the network as $F^c$. We interpret the predicted confidence for the $c$-th class for input $\boldsymbol{x}$ as $\sigma(F(\boldsymbol{x}\,;\boldsymbol{\theta}))^c$.

**Kernel Functions.** Kernel functions implicitly map the data vector $\boldsymbol{x}$ to a feature vector $\rho(\boldsymbol{x})$ in a higher dimensional RKHS $\mathcal{V}$ for which the kernel function $\boldsymbol{\kappa}(\cdot, \cdot)$ evaluates the inner product of two feature vectors in $\mathcal{V}$. We will notate the data matrix $\boldsymbol{X} = [\boldsymbol{x}_1, \ldots, \boldsymbol{x}_N] \in \mathbb{R}^{N \times n}$ with N the number of training samples. With some abuse of notation, we will write $\boldsymbol{\kappa}(\boldsymbol{x}, \boldsymbol{X}) \in \mathbb{R}^N$ for the vector whose $j$-th component is $\boldsymbol{\kappa}(\boldsymbol{x}, \boldsymbol{x}_j)$ and $\boldsymbol{\kappa}(\boldsymbol{X}, \boldsymbol{X}) \in \mathbb{R}^{N \times N}$ for the matrix whose $(i, j)$-th entry is $\boldsymbol{\kappa}(\boldsymbol{x}_i, \boldsymbol{x}_j)$.

**Kernel General Linear Models as Surrogate Models** We limit our investigation of surrogate models to kernel general linear models. We define a general kernel linear model $\mathrm{kGLM} : \mathcal{X} \to \mathcal{Y}$ as:

$$\mathrm{kGLM}(\boldsymbol{x}) := \boldsymbol{W}\boldsymbol{\kappa}(\boldsymbol{x}, \boldsymbol{X}) + \boldsymbol{b}, \tag{1}$$

where $\boldsymbol{W} \in \mathbb{R}^{C \times N}$ is a learnable weight matrix, $\boldsymbol{\kappa}$ is the kernel function, and $\boldsymbol{b} \in \mathbb{R}^C$ is a learnable bias vector. We compute classifications from kGLM by mapping the final activations to softmax confidences. The parameters $\boldsymbol{W}$ and $\boldsymbol{b}$ are learned using an optimizer to minimize the cross entropy loss using the same dataset upon which the NN is trained. Given an input $\boldsymbol{x}$, the softmax activation $\sigma$, and a NN $F(\boldsymbol{x}\,;\boldsymbol{\theta})$, the ideal surrogate modeling goal is to find a kGLM that satisfies:

$$\sigma(\mathrm{kGLM}(\boldsymbol{x})) = \sigma(F(\boldsymbol{x}, \boldsymbol{\theta}))), \tag{2}$$

for all $\boldsymbol{x}$. Keeping this ideal in mind is useful for building intuition, but in practice, we will relax from this ideal goal for reasons described below.

**Data Attribution with Kernels.** Our main motivation is to explain neural networks through data attribution, i.e., by computing "a score for each training datapoint indicating its importance to the output of interest" (TRAK). Given the choice of kernel function $\boldsymbol{\kappa}$, the scalar valued data attribution for the $c$-th class for a test input $\boldsymbol{x}$ and a training datapoint $\boldsymbol{x}_i$ is given by:

$$A(\boldsymbol{x}, \boldsymbol{x}_i)^c := \boldsymbol{W}_{c,i}\,\boldsymbol{\kappa}(\boldsymbol{x}, \boldsymbol{x}_i) + \frac{\boldsymbol{b}_c}{N}. \tag{3}$$

Where the $\frac{b_c}{N}$ term is necessary to ensure that the sum over the attributions for the entire training dataset is equal to the kGLM's logit for class $c$, $\sum_{i=1}^{N} A(\boldsymbol{x}, \boldsymbol{x}_i)^c = \text{kGLM}(\boldsymbol{x})^c$. If the kGLM is an ideal surrogate model Eq. 2, then the softmax function applied to the vector created from each class attribution will equal the NN confidence in each class. Consequently, we will have decomposed the reasoning for the NN's specific confidence in each class to a linear combination of similarities between $\boldsymbol{x}$ and each training datapoint $\boldsymbol{x}_i$. We emphasize that Eq. 3 is our definition of data attribution. Attribution is a weighted sum of kernel/similarity values.

## 3 Methods

We now turn towards the novel work of this research. In the following sections we describe our measure of faithfulness then introduce the kernel functions.

**Evaluating the Faithfulness of Surrogate Models.** Given many choices of kernel functions we require a measure to determine which surrogate models have higher approximation quality (i.e., faithfulness) to the NN. We relax from the ideal surrogate model goal Eq. 2 and instead evaluate kernel functions by how well they are correlated with the neural network using the Kendall-$\tau$ rank correlation.

To assess the faithfulness of a surrogate model, we compute $\tau_K$ between the softmax probability of the neuron representing the correct class, $\sigma(F(\boldsymbol{x}\,;\boldsymbol{\theta}))^c$, and the kGLM softmax probability for the output representing the correct class, $\sigma(\text{kGLM}(\boldsymbol{x}))^c$. $\tau_K$ was chosen for two reasons; First, $\tau_K$ has a range $[-1, 1]$ with $\pm 1$ representing a monotonic relationship and a value of 0 representing no correlation. Second, if the relationship between the kGLM and NN is strictly monotonic, then an invertible mapping function exists between the kGLM softmax probabilities and the NN's (Bartle & Sherbert, 2011). Therefore, for a $\tau_K = 1$ we would recover the one-to-one ideal surrogate model relationship given by Eq. 2. In Appendix L, we demonstrate how to find these mapping functions with iterative optimizers (Virtanen et al., 2020). We provide a formal definition of Kendall-$\tau$ rank correlation in appendix G.

We additionally report two more complementary metrics. While we have argued that the test accuracy is flawed to measure faithfulness, we will report the test accuracy differential to be complete with prior works. We define test accuracy differential (TAD) as:

$$\text{TAD} := \text{TestAcc}_{\text{kGLM}} - \text{TestAcc}_{\text{NN}}.$$

A fundamental limitation of $\tau_K$ is that it can only be computed over a set of scalar outputs so does not take advantage of the vectorized output of classification networks. To compensate, we will also report the misclassification coincidence rate, ($R_{\text{miss}}$), which captures whether two models both misclassify the same datapoints as the same class, which is an intuitive property $\tau_K$ misses. A formal definition of $R_{\text{miss}}$ is available in appendix G. We now turn to defining the specific kernel functions we evaluate.

**Trace Neural Tangent Kernel.** For any two data inputs $\boldsymbol{x}_i$ and $\boldsymbol{x}_j$, we define the Jacobian of the NN's $c$-th output neuron with respect to $\boldsymbol{\theta}$ at datapoint $\boldsymbol{x}_i$ as $\boldsymbol{g}^c(\boldsymbol{x}_i;\boldsymbol{\theta}) = \nabla_{\boldsymbol{\theta}} F^c(\boldsymbol{x}_i;\boldsymbol{\theta})$. Then, for choice of class $c$ and $c'$, the eNTK is a kernel function defined as:

$$\text{eNTK}(\boldsymbol{x_i}, \boldsymbol{x_j}) := \langle \boldsymbol{g}^c(\boldsymbol{x}_i;\boldsymbol{\theta}) \boldsymbol{g}^{c'}(\boldsymbol{x}_i;\boldsymbol{\theta}) \rangle. \tag{4}$$

For $C$ classes and $N$ datapoints, the full eNTK can be evaluated for each choice of $(c, c')$ and $(i, j)$ resulting in a large $NC \times NC$ total size matrix. This matrix is often too expensive to compute or manipulate in memory, leading researchers to seek approximations.

We introduce now the trace neural tangent kernel (trNTK) approximation, which removes the $C^2$ scaling in memory by effectively performing a "block-trace" operation on the original eNTK. The trNTK is a kernel function defined as:

$$\text{trNTK}(\boldsymbol{x_i}, \boldsymbol{x_j}) := \frac{\sum\limits_{c=1}^{C} \langle \boldsymbol{g}^c(\boldsymbol{x}_i;\boldsymbol{\theta}), \boldsymbol{g}^c(\boldsymbol{x}_j;\boldsymbol{\theta}) \rangle}{(\sum\limits_{c=1}^{C} ||\boldsymbol{g}^c(\boldsymbol{x}_i;\boldsymbol{\theta})||^2)^{\frac{1}{2}} (\sum\limits_{c=1}^{C} ||\boldsymbol{g}^c(\boldsymbol{x}_j;\boldsymbol{\theta})||^2)^{\frac{1}{2}}}. \tag{5}$$

The denominator of Eq. 5 is a normalization that makes the trNTK a kernel of cosine-similarity values. It has been suggested that this normalization helps smooth out kernel mass over the entire training dataset (Akyürek et al., 2022). The normalization ensures that two identical inputs always have maximum similarity value 1. Additional intuition about how this kernel relates to the geometry of the neural network function surface is available in Appendix C. We provide additional details about these definitions in Appendix D. In the following section, we relate this kernel to another approximate eNTK kernel, the pseudo neural tangent kernel.

Wei et al. (2022)

**Relationship to the Pseudo Neural Tangent Kernel.** We can understand the motivation for the trNTK in the context of another approximate eNTK, called the pseudo neural tangent kernel (pNTK). The pNTK computed between inputs $\boldsymbol{x}_i$ and $\boldsymbol{x}_j$ is a kernel function defined as:

$$\text{pNTK}(\boldsymbol{x}_i, \boldsymbol{x}_j) := \frac{1}{C}\left(\nabla_{\boldsymbol{\theta}}\sum_{c=1}^{C}F(\boldsymbol{x}_i;\boldsymbol{\theta})^c\right)^{\top}\left(\nabla_{\boldsymbol{\theta}}\sum_{c=1}^{C}F(\boldsymbol{x}_j;\boldsymbol{\theta})^c\right). \tag{6}$$

Mohamadi & Sutherland (2022) showed that the product of the $\text{pNTK}(\boldsymbol{x}_i, \boldsymbol{x}_j)$ with the $C \times C$ identity matrix is bounded in Frobenius norm to the eNTK by $\mathcal{O}(\frac{1}{\sqrt{n}})$, with $n$ the width parameter of a feed forward fully connected NN with ReLU activation (Nair & Hinton, 2010; Glorot et al., 2011) and He-normal (He et al., 2015a) initialization, with high probability over random initialization.

We can frame the critical differences between the pNTK and trNTK by how each approximate the eNTK. The pNTK approximates the eNTK as a constant diagonal matrix with constant equal to the scalar kernel function given in Eq. 6. In contrast, the trNTK allows the diagonal elements of the eNTK approximation to vary, and in fact, calculates these values directly. Both the pNTK and trNTK perform a simplifying sum over the diagonal elements, which reduces the memory footprint of the approximations by a factor $C^2$ compared to the eNTK. We choose not to compare directly with the pNTK because the trNTK is a higher cost, but more precise, approximation of the eNTK. Instead, we focus our comparisons to much lower cost alternatives, including a projection variant of the pNTK.

**Projection trNTK and Projection pNTK.** For large number of parameters $P$ and large datasets $N$, computing approximate eNTK remain expensive, therefore, we explore a random projection variant that allows us to effectively choose $P$ regardless of architecture studied. Let $\boldsymbol{P}$ be a random projection matrix $\boldsymbol{P} \in \mathbb{R}^{K \times P}$, $K \ll P$, with all entries drawn from either the Gaussian $\mathcal{N}(0, 1)$ or Rademacher (with p=0.5 for all entries) distribution. $K$ is a hyperparameter setting the projection matrix dimension. We set $K = 10240$ for all experiments. We use $\boldsymbol{P}$ to project the Jacobian matrices to a lower dimension, which reduces the memory needed to store the Jacobians and reduce the time complexity scaling. The Johnson-Lindenstrauss lemma ensures that most of the information in the original Jacobians is preserved when embedded into the lower dimensional space (Johnson & Lindenstrauss, 1984). We define the proj-trNTK and proj-pNTK as random projection variants of the trNTK and pNTK:

$$\text{proj-pNTK}(\boldsymbol{x}_i, \boldsymbol{x}_j) := \frac{\left\langle \boldsymbol{P}\sum_{c=1}^{C}\boldsymbol{g}^c(\boldsymbol{x}_i, \boldsymbol{\theta}), \boldsymbol{P}\sum_{c=1}^{C}\boldsymbol{g}^c(\boldsymbol{x}_j, \boldsymbol{\theta})\right\rangle}{\left\|\boldsymbol{P}\sum_{c=1}^{C}\boldsymbol{g}^c(\boldsymbol{x}_i, \boldsymbol{\theta})\right\| \cdot \left\|\boldsymbol{P}\sum_{c=1}^{C}\boldsymbol{g}^c(\boldsymbol{x}_j, \boldsymbol{\theta})\right\|} \tag{7}$$

$$\text{proj-trNTK}(\boldsymbol{x}_i, \boldsymbol{x}_j) := \frac{\sum_{c=1}^{C}\langle \boldsymbol{P}\boldsymbol{g}^c(\boldsymbol{x}_i;\boldsymbol{\theta}), \boldsymbol{P}\boldsymbol{g}^c(\boldsymbol{x}_j;\boldsymbol{\theta})\rangle}{(\sum_{c=1}^{C}||\boldsymbol{P}\boldsymbol{g}^c(\boldsymbol{x}_i;\boldsymbol{\theta})||^2)^{\frac{1}{2}}(\sum_{c=1}^{C}||\boldsymbol{P}\boldsymbol{g}^c(\boldsymbol{x}_j;\boldsymbol{\theta})||^2)^{\frac{1}{2}}}, \tag{8}$$

where both definitions include the cosine-normalization.

Random projection variants can improve the time complexity scaling for computing approximate eNTK under large dataset size and large number of parameters. Assuming computation via Jacobian contraction and time $[FP]$ for a forward pass, the eNTK time complexity is: $NC[FP] + N^2C^2P$ (Novak et al., 2022). The pNTK computation reduces this to $N[FP] + N^2P$; while the trNTK computation only reduces to $NC[FP] + N^2CP$. In contrast, the proj-pNTK costs $N[FP] + N^2K +$

$NKP$, and the proj-trNTK costs $NC[FP] + CN^2K + CNKP$. The final term in the projection variants is the cost of the extra matrix multiplication with the random projection matrix $\boldsymbol{P}$ and the Jacobian matrix. For $K \ll P$ and $N$ large, projection variants reduce the time complexity.

**Additional Kernel Functions.** We also evaluate the conjugate kernel (CK) formed from the Gram matrix of the final embedding vector (Fan & Wang, 2020; Yeh et al., 2018), the un-normalized trNTK (trNTK$^0$) which is equal to the numerator of Eq. 5, and the embedding kernel (Akyürek et al., 2023), formed from a sum over the Gram matrices of embedding vectors from various layers in the network architecture. See Appendix B for formal definition of these kernels.

## 4 RESULTS

**Experiments.** Classification NNs with architectures and datasets (MNIST (Lecun et al., 1998), FM-NIST (Xiao et al., 2017), CIFAR10 (Krizhevsky & Hinton, 2009), and COLA (Warstadt et al., 2018)) shown in Table 1 are trained using standard techniques. Additional details regarding datasets are provided in Appendix K.1. Models that have a value of more than 1 in the column '# Models' in Table 1 are trained multiple times with different seeds to generate uncertainty estimates. The ResNet18 (He et al., 2015b), ResNet34, and MobileNetV2 (Sandler et al., 2018) models were trained by an independent research group with weights downloaded from an online repository (Phan, 2021). Bert-base (Devlin et al., 2019) weights were downloaded from the HuggingFace (Wolf et al., 2019) repository then transferred onto the COLA dataset, as is common practice for foundation models (Bommasani et al., 2021). After training, we calculate the trNTK and alternative kernels using PyTorch automatic differentiation (Paszke et al., 2019). We train a kGLM (`sklearn.SGDclassifier`) (Pedregosa et al., 2011) for each $\boldsymbol{\kappa}$ using the same training dataset for training the NN model. All computation was completed on a single A100 GPU with 40GB memory. Details such as specifics of architecture and choice of hyperparameters are available in Appendix K.

**Faithful Surrogate Modeling via** trNTK**.** We calculate the $\tau_K$ correlation between the surrogate model and underlying NN and report the results in Table 1. We find that the efficacy of our surrogate model as measured by the correlation to the NN changes depending on architecture and dataset; though remarkably, $\tau_K$ is consistently high, with a lower bound value of 0.7 across all experiments, indicating high faithfulness. To demonstrate high $\tau_K$ implies we can achieve a point-for-point linear realization of the NN, we learn a non-linear mapping from the kGLM to the NN (Figure 1 for Bert-base. (Additional visualizations for the remainder of experiments are available in Appendix L.) Finally, we observe that the kGLM with choice of $\boldsymbol{\kappa} = $ trNTK achieves comparable test accuracy as the underlying NN, which replicates the observations of prior work (Long, 2021; Vyas et al., 2022; Mohamadi & Sutherland, 2022) using our trNTK.

**Data Attribution with** trNTK**.** Accepting that the trNTK is a faithful kernel function for a kGLM surrogate model, we can use the data attribution formalism to analyze the importance of individual training datapoints to the classification. In Figure 2 we present the visualization of data attribution for one test input and provide additional visualizations in Appendix M.1. The distribution of attribution follows a regular pattern in every visualization generated: the central value of attribution mass for each logit from each class is centered on the distribution of all training data from that class. We emphasize that in no cases have we observed a sparse number of training datapoints dominate the data attribution.

**Comparison of Faithfulness between Kernels Functions.** For ResNet18 and Bert-base models, we evaluate our choice of trNTK against alternative kernel functions, reporting $\tau_K$ and test accuracy differential in Table 2. Across both ResNet18 and Bert-base experiments, we observe that the trNTK forms surrogate models with the highest correlation to the underlying NN decision function and is furthermore consistent in replicating the performance of these networks (TAD nearly 0). The embedding kernel (Em) does not perform as consistently between both tasks, but for its intuitive connection to the internal representation of the neural network may warrant further investigation.

**Faithful Surrogates in Data Poisoning Regime.** Next, we evaluate whether surrogate models can be extended to analyze network behavior on poisoned data. We train a 21-layer CNN (details available in Appendix K.2.5) using BadNet CIFAR10 data (Gu et al., 2019; Shan et al., 2022). We randomly perturb training data by placing a yellow square in a tenth of training images from CIFAR10 and modify the label of these perturbed images to a targeted label (see example in Appendix N). We

Table 1: **Choice of $\kappa = \mathrm{trNTK}$ faithfully forms a surrogate model of underlying NN.** We perform each experiment with '# Models' independent seeds. For each model and dataset we train and extract the $\mathrm{trNTK}$, train a kGLM, then calculate and report the $\tau_K$ correlation between the kGLM softmax probability and NN softmax probability for the correct class. The NN test accuracy column shows that training terminates with a highly performant model, and the test accuracy differential (TAD) columns reports the difference between the kGLM test accuracy and the NN test accuracy. We report the leading digit of error (standard error of the mean) as a parenthetical, when available.

| Model (Dataset) | # Models | NN test acc (%) | TAD (%) | $\tau_K$ |
|---|---|---|---|---|
| MLP (MNIST2) | 100 | 99.64(1) | +0.03(5) | 0.708(3) |
| CNN (MNIST2) | 100 | 98.4(1) | -0.2(2) | 0.857(7) |
| CNN (CIFAR2) | 100 | 94.94(5) | -2.1(5) | 0.711(3) |
| CNN (FMNIST2) | 100 | 97.95(4) | -2.2(2) | 0.882(3) |
| ResNet18 (CIFAR10) | 1 | 93.07 | -0.28 | 0.776 |
| ResNet34 (CIFAR10) | 1 | 93.33 | -0.29 | 0.786 |
| MobileNetV2 (CIFAR10) | 1 | 93.91 | -0.4 | 0.700 |
| BERT-base (COLA) | 4 | 83.4(1) | -0.1(3) | 0.78(2) |

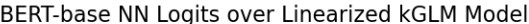

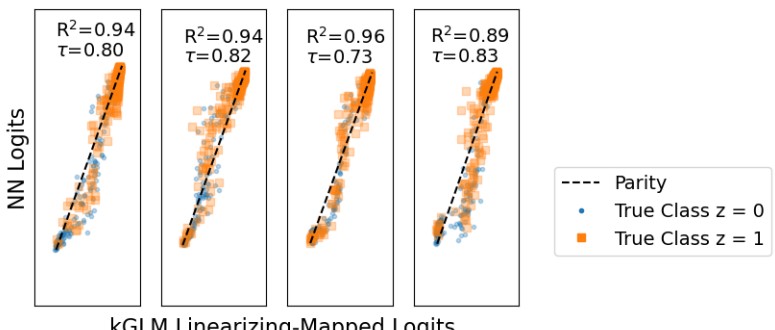

Figure 1: **Linear Realization of Bert-base Model.** Each panel shows a linearization of a Bert-base transfer model, initialized from a different seed. An invertible mapping is fit between the kGLM and NN to transform the kGLM's final activations to the NN's, described in Appendix L. Both $\tau_K$ and the Coefficient of Determination ($R^2$) are shown for each model.

Table 2: **Comparison across surrogate feature spaces.** For ResNet18 and Bert-base experiments we report the faithfulness as $\tau_K$, test-accuracy-differential (TAD), and misclassification coincidence rate ($R_{\mathrm{Miss}}$) for each kernel function: the trace-NTK ($\mathrm{trNTK}$), unnormalized trace-NTK ($\mathrm{trNTK}^0$, the projection trace NTK ($\mathrm{proj\text{-}trNTK}$), the projection pseudo NTK ($\mathrm{proj\text{-}pNTK}$), the embedding kernel (Em) and the conjugate kernel (CK). If available, we report leading digit of error (standard error of the mean) as a parenthetical.

| Exp Name | Metric | $\kappa$ | | | | | |
|---|---|---|---|---|---|---|---|
| | | trNTK | trNTK$^0$ | proj-trNTK | proj-pNTK | Em | CK |
| ResNet18 | $\tau_K$ | 0.776 | 0.658 | 0.737 | 0.407 | 0.768 | 0.630 |
| | TAD (%) | -0.30 | -0.52 | -0.20 | -0.30 | -0.32 | -0.20 |
| | $R_{\mathrm{Miss}}$ | 0.75 | 0.65 | 0.77 | 0.71 | 0.80 | 0.73 |
| Bert-base | $\tau_K$ | 0.809(9) | 0.5(1) | 0.800(9) | 0.72(2) | 0.65(2) | 0.52(4) |
| | TAD (%) | +0.1(3) | +0.6(2) | +0.1(2) | +0.5(2) | -0.3(5) | -0.1(1) |
| | $R_{\mathrm{Miss}}$ | 0.67(2) | 0.71(5) | 0.61(2) | 0.86(3) | 0.86(2) | 0.91(2) |

create a "clean" test dataset from CIFAR10's normal test dataset, and a "poisoned" test dataset by placing yellow squares into each image of CIFAR10's test dataset. At test time, perturbed test data tricks the model into producing labels of the targeted label. We train a model on this poisoned dataset, compute each kernel function, measure faithfulness, and report our results in Table 3. We find that the

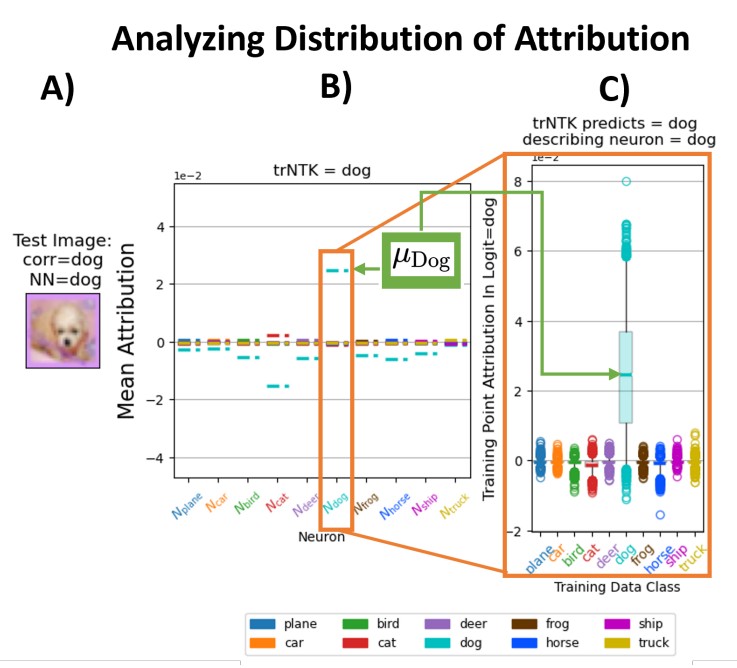

Figure 2: **Overview of Using Kernel Functions for Data Attribution** A) An image from the test dataset of CIFAR10 is chosen. B) We propagate the test image through the NN and plot the mean attribution of the training points from each class for each output neuron. C) Zooming into the neuron representing class "dog", we view the distribution of attributions as a modified box-plot with central lines the mean and outliers shown as flier points. The mean lines are always observed to be within the inner quartile, suggesting that no sparse number of datapoints dominate the central value, and therefore, do not dominate the data attribution.

$\mathrm{trNTK}$ is most faithful to the NN on the clean test data, but the $\mathrm{proj}$-pNTK is most faithful when evaluated on the poisoned test data. Overall in comparison to the non-poisoned set of experiments each kGLM is less faithful, except for the $\mathrm{proj}$-pNTK. We also point out that the kGLM with overall highest faithfulness are the kernel functions with our cosine-normalization applied.

In addition, we show an application of our surrogate modeling approach enabled by kernel-techniques. Forensics models trace NN behavior on unseen poisoned data to the poisoned data source in a training set (Shan et al., 2022). We treat each kernel as a forensic model: for each image in the clean and poisoned test dataset we compute the top 5 most similar training datapoints. If 3/5 of these training datapoints are poisoned we flag the test image as poisoned. In doing so, we can filter poisoned images from clean images. We report the performance of our forensic models using precision and recall (see Appendix G) in table 3. Each kernel, except for the conjugate kernel, are all comparable in performance as forensics models. Appendix N provides examples of multiple forensic models acting on poisoned and clean versions of CIFAR10 data.

Table 3: **Poisoned data attribution forensics.** We compute each kernel function between all poisoned training data and the clean test dataset. We report $\tau_K$, TAD, and $R_{\mathrm{Miss}}$ between the kGLM and NN for both the poisoned (poi.) and clean set of unseen test images. Finally, we evaluate each kernel as a filter for identifying unseen poisoned data through high similarity to poisoned training data and report the performance as Precision and Recall.

| Method | Precision (%) | Recall (%) | $\tau_K$ | TAD (%) | $R_{\mathrm{Miss}}$ | poi. $\tau_K$ | poi. TAD(%) | poi. $R_{\mathrm{Miss}}$ |
|---|---|---|---|---|---|---|---|---|
| $\mathrm{trNTK}$ | 99.99 | 100.00 | 0.643 | +0.45 | 0.44 | 0.569 | +0.09 | 0.12 |
| $\mathrm{trNTK}^0$ | 99.99 | 99.97 | 0.344 | +0.87 | 0.20 | 0.125 | +0.13 | 0.01 |
| proj-$\mathrm{trNTK}$ | 99.99 | 99.97 | 0.565 | +0.09 | 0.45 | 0.418 | +1.3 | 0.12 |
| proj-pNTK | 99.99 | 100.00 | 0.554 | +0.07 | 0.59 | 0.665 | -1.3 | 0.11 |
| Embedding | 99.71 | 100.00 | 0.430 | -2.73 | 0.07 | 0.261 | -13.98 | 0.22 |
| CK | 1.65 | 50.61 | 0.552 | -3.50 | 0.38 | 0.454 | -81.25 | 0.00 |

## 5 SUMMARY AND CONCLUSIONS

**Impact of Linear Surrogate Modeling for Explainability.** We have shown evidence supporting the choice of the $\mathrm{trNTK}$ as a consistently faithful choice of kernel function for a surrogate model (table 1). We made this determination by measuring the correlation between the kGLM surrogate and the NN, which is an improvement over past methodologies. Our choice of a linear model as surrogate model allows us to separate the attribution terms from each training datapoint, and ensures the central value of the attribution distribution is coupled to the kGLM's logit, and therefore the NN which it approximates (Section 2). We observed that the highest attributed images from the $\mathrm{trNTK}$ have relatively small mass compared to the bulk contribution, suggesting that the properties of the bulk, rather than a few outliers, are the main source driving decision making. We believe this is a result of the cosine normalization we apply in our definition of the $\mathrm{trNTK}$, as the unnormalized $\mathrm{trNTK}^0$ shows a much tighter IQR of attribution (see appendix M.1.2), and in fact, this pattern exists between all normalized vs un-normalized kernel functions. This directly visualizes the intuition that the cosine normalization "smooths-out" the attribution (Akyürek et al., 2022). Because the properties of the bulk drive classification, we conclude that presenting the top highest attribution training images without the context of the entire distribution of attribution is potentially misleading as a form of explanation, i.e., the assumption of sparsity in explain-by-example strategies is misguided.

**Comparison of Kernel Functions for Surrogate Models.** Our quantitative experiments showed the $\mathrm{trNTK}$ as more consistently correlated to the NN model compared to the unnormalized $\mathrm{trNTK}$, Embedding kernel, and CK. We observe qualitative differences between these kernel's attributions (Appendix M.1) and which training datapoints have highest similarity (Appendix N). As a qualitative comparison between kernel functions, in Appendix M.2 we visualize the top-5 most similar datapoints evaluated by each kernel function. This further reveals the similarities and differences between kernel functions. Overall, we observe that the $\mathrm{trNTK}$ is more sensitive to conceptual similarities between test and train examples than the CK. The embedding kernel is consistently sensitive to background pixel values, though this may be an artifact from our specific choice of layers to sample from. The $\mathrm{proj\text{-}trNTK}$, as expected, follows closely with the regular $\mathrm{trNTK}$. These differences could be used to tied to interesting phenomena: for example, because the CK is computed from the final embedding it is likely more sensitive to the effects of neural-collapse (Papyan et al., 2020) than the NTK, which is computed from Jacobians of weight tensors across the entire architecture. We believe this fact explains why the highest similar images measured by the $\mathrm{trNTK}$ are more conceptually tied to the specific test image, while the CK has collapsed that inner-class variance away.

**Computational Feasibility.** Finally, we comment on the computational feasibility of each of the kernel functions. Table 4 reports the time to compute each kernel, and Appendix F shows that the empirical residual distribution between the $\mathrm{trNTK}$ and $\mathrm{proj\text{-}trNTK}$ falls exponentially. The projection-trNTK and projection-pNTK have efficient computation thanks to software made available in Park et al. (2023). The full $\mathrm{trNTK}$ is by far the slowest. As implemented, our $\mathrm{trNTK}$ computation was layerwise (see Appendix D), except in the Poisoning experiment, which we now believe is sub-optimal. Both the $\mathrm{trNTK}$ and projection-trNTK computation scales with the number of output neurons linearly, so for models with large output space the projection-pNTK may remain the only feasible option. Finally, because the residuals between the $\mathrm{trNTK}$ and $\mathrm{proj\text{-}trNTK}$ are small and decay rapidly, we believe using the projected variants are well justified. In total, we believe the differences between the $\mathrm{trNTK}$ and $\mathrm{proj\text{-}trNTK}$ are small enough that for small number of outputs, our recommendation is to utilize the $\mathrm{proj\text{-}trNTK}$. Finally, see Appendix A for limitations.

Table 4: **Computational Complexity of Large Model Experiments.** We report time to compute each of the $\mathrm{trNTK}$, $\mathrm{proj\text{-}trNTK}$, and $\mathrm{proj\text{-}pNTK}$ for the large model large dataset experiments are shown.

| Exp Name | trNTK | proj-trNTK | proj-pNTK |
|---|---|---|---|
| ResNet18 | 389h | 1.12h | 7.4m |
| BertBase | 1200h | 22m | 12m |
| Poisoning | 50h | 9.3m | 1m |

ACKNOWLEDGMENTS

The authors thank Panos Stinis, Mark Raugas, Saad Qadeer, Adam Tsou, Emma Drobina, Amit Harlev, Ian Meyer, and Luke Gosink for varied discussions while preparing the draft. This work would not have been possible without the help from Wendy Cowley in helping navigate the release protocol. The authors thank Davis Brown for discussions regarding TRAK. A.W.E., Z.W., S.C., N.F., and T.C. were partially supported by the Mathematics for Artificial Reasoning in Science (MARS) initiative via the Laboratory Directed Research and Development (LDRD) Program at PNNL and A.D.S. and T.C. were partially supported by the Statistical Inference Generates kNowledge for Artificial Learners (SIGNAL) Program at PNNL. A.D.S. was partially supported by the US NSF under award CNS-2148104. PNNL is a multi-program national laboratory operated for the U.S. Department of Energy (DOE) by Battelle Memorial Institute under Contract No. DE-AC05-76RL0-1830.

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

## A    LIMITATIONS

We point out previous works using support vector machines (SVM) kernel surrogate models report limitations that we believe extend to kGLM models. We know of two such limitations. We found that SVM surrogate models fail to replicate NN behavior under gradient-based adversarial attacks Appendix J. In addition, SVM surrogate models do not have the same scaling relationships as underlying NNs (Vyas et al., 2022). Our conclusions are limited to kGLM surrogate models; an interesting follow-on work would investigate using kernel functions in K-Nearest Neighbors surrogate models which may recover a sparse explanation.

A fundamental limitation of our choice of Kendall-$\tau$ was discussed in section 3 and we expand upon it here. Kendall-$\tau$ requires a set of scalars, which forces us to reduce the naturally vector output space of classification networks to a single value. We choose to use the logit representing the correct ground-truth class. This is reasonable given that the confidence given by the neural network in the correct class is an interesting behavior with consequences to the classification task; however, this choice does not leverage the total amount of information given by the output soft-max vector. To compensate for this, we report the misclassification coincidence rate, $R_{\text{Miss}}$, which utilizes the intuition that coupled models should also be wrong in the same way, at the same time. While this added metric provides an additional powerful line of evidence demonstrating the coupling between kGLM and NN, it also clouds our analysis on which Kernel function represents the best choice. Therefore future work should continue to improve upon Kendall-$\tau$ as a metric for faithfulness.

While many explainability techniques now exist, its not always clear how useful any technique actually is until a human reviewer attempts to utilize the technique. In this work we do not perform any human subjects testing to evaluate each kernel, but in principle this would be an interesting direction for future work.

This work's premise is limited in that we have no guarantee that the surrogate model is performing "reasoning" in the same manner as the underlying neural network. We have only worked to establish that the kGLMs are highly coupled to NNs and evaluate this coupling between different choices of kernel functions. Because we find evidence for a high correlation between NN and kGLM models, we suggest that structure of kGLMs serve as a potential explanation of NNs in a way that connect decisions made on new inputs to specific training data. In the most limited view of this work, this is simply a fundamental assumption that must be empirically evaluated for each new network-kGLM pair. Follow on work could look to compute the eNTK at multiple times throughout training to form an approximation to the path kernel (Domingos, 2020).

Finally, our evaluations between the trNTK and the pNTK are limited in extent to which either are a true approximation of the eNTK. For example, we are guaranteed that the $tr(\text{trNTK})$ $tr(\text{eNTK})$ at all times, but the $tr(\text{pNTK})$ does not necessarily equal the $tr(\text{eNTK})$ at all times. An interesting direction of future work would be to evaluate to what extent the trNTK reproduced the eNTK in a similar manner as Mohamadi & Sutherland (2022). In any case, given the computational difficulty of the eNTK we believe the more interesting questions are for what behavior/phenomena are the approximations "close-enough" to model the eNTK. This has recently been explored in Qadeer et al. (2023).

## B    DEFINITION OF KERNELS

In this Appendix we provide the definition of each of the kernel functions evaluated. For convenience we restate the definition of the trNTK.

**trNTK** Recall the definition of the total gradient with respect to $\boldsymbol{\theta}$ at datapoint $\boldsymbol{x}_i$ by

$$\boldsymbol{g}(\boldsymbol{x}_i; \boldsymbol{\theta})^c = \nabla_{\boldsymbol{\theta}} F^c(\boldsymbol{x}_i; \boldsymbol{\theta}).$$

Then the trNTK evaluated at datapoints $\boldsymbol{x}_i$ and $\boldsymbol{x}_j$ is given by

$$\text{trNTK}(\boldsymbol{x_i}, \boldsymbol{x_j}) := \frac{\sum\limits_{c=1}^{C} \langle \boldsymbol{g}^c(\boldsymbol{x}_i; \boldsymbol{\theta}), \boldsymbol{g}^c(\boldsymbol{x}_j; \boldsymbol{\theta}) \rangle}{(\sum\limits_{c=1}^{C} ||\boldsymbol{g}^c(\boldsymbol{x}_i; \boldsymbol{\theta})||^2)^{\frac{1}{2}} (\sum\limits_{c=1}^{C} ||\boldsymbol{g}^c(\boldsymbol{x}_j; \boldsymbol{\theta})||^2)^{\frac{1}{2}}}.$$

We provide additional details about the exact calculation in Appendix D.

**Projection Trace Neural Tangent Kernel.** We restate our definition of the proj-trNTK kernel function:

$$\text{proj-trNTK}(\boldsymbol{x}_i, \boldsymbol{x}_j) := \frac{\sum\limits_{c=1}^{C} \langle \boldsymbol{P}\boldsymbol{g}^c(\boldsymbol{x}_i; \boldsymbol{\theta}), \boldsymbol{P}\boldsymbol{g}^c(\boldsymbol{x}_j; \boldsymbol{\theta}) \rangle}{(\sum\limits_{c=1}^{C} ||\boldsymbol{P}\boldsymbol{g}^c(\boldsymbol{x}_i; \boldsymbol{\theta})||^2)^{\frac{1}{2}} (\sum\limits_{c=1}^{C} ||\boldsymbol{P}\boldsymbol{g}^c(\boldsymbol{x}_j; \boldsymbol{\theta})||^2)^{\frac{1}{2}}},$$

We remind the reader that $\boldsymbol{P}$ is a Rademacher or Gaussian random projection matrix $\in \mathbb{R}^{K \times P}$, with $K$ a hyperparameter, $P$ the number of model parameters, and K chosen to be $K \ll P$. In all experiments K = 10240.

**Projection Pseudo Neural Tangent Kernel.**

$$\text{proj-pNTK}(\boldsymbol{x}_i, \boldsymbol{x}_j) := \frac{\langle \boldsymbol{P}\sum\limits_{c=1}^{C} \boldsymbol{g}^c(\boldsymbol{x}_i, \boldsymbol{\theta}), \boldsymbol{P}\sum\limits_{c=1}^{C} \boldsymbol{g}^c(\boldsymbol{x}_j, \boldsymbol{\theta}) \rangle}{||\boldsymbol{P}\sum\limits_{c=1}^{C} \boldsymbol{g}^c(\boldsymbol{x}_i, \boldsymbol{\theta})|| \cdot ||\boldsymbol{P}\sum\limits_{c=1}^{C} \boldsymbol{g}^c(\boldsymbol{x}_j, \boldsymbol{\theta})||}$$

**Embedding** Akyürek et al. (2022) defines the embedding kernel, which we restate here. The embedding kernel is computed from the correlation of the activations following each layer. Let $\lambda_\ell(\boldsymbol{x}; \boldsymbol{\theta})$ be the output of the $\ell$-th hidden layer of $F(\boldsymbol{x}; \boldsymbol{\theta})$. We denote the $\ell$-th embedding kernel at datapoints $\boldsymbol{x}_i$ and $\boldsymbol{x}_j$ by

$$E_\ell(\boldsymbol{x}_i, \boldsymbol{x}_j) = \frac{\langle \lambda_l(\boldsymbol{x}_i; \boldsymbol{\theta}), \lambda_l(\boldsymbol{x}_j; \boldsymbol{\theta}) \rangle}{||\lambda_l(\boldsymbol{x}_i; \boldsymbol{\theta})|| ||\lambda_l(\boldsymbol{x}_j; \boldsymbol{\theta})||}.$$

Let the full embedding kernel be defined by the normalized sum over the unnormalized embedding kernel at each layer of the NN

$$E(\boldsymbol{x}_i, \boldsymbol{x}_j) = \frac{\sum_{\ell=1}^{L} \langle \lambda_\ell(\boldsymbol{x}_i; \boldsymbol{\theta}), \lambda_\ell(\boldsymbol{x}_j; \boldsymbol{\theta}) \rangle}{\sqrt{\sum_{\ell=1}^{L} ||\lambda_\ell(\boldsymbol{x}_i; \boldsymbol{\theta})||^2 ||\lambda_\ell(\boldsymbol{x}_j; \boldsymbol{\theta})||^2}}.$$

Embedding kernels are an interesting comparison for the data attribution task when we consider the prominent role they play in transfer learning and auto-encoding paradigms. In both, finding an embedding that can be utilized in down-stream tasks is the objective.

**Conjugate Kernel** We utilize an the empirical conjugate kernel (CK) to compare to the trNTK. Let the normalized CK be defined by

$$\text{CK}(\boldsymbol{x}_i, \boldsymbol{x}_j) = \frac{\langle \lambda_L(\boldsymbol{x}_i; \boldsymbol{\theta}), \lambda_L(\boldsymbol{x}_j; \boldsymbol{\theta}) \rangle}{||\lambda_L(\boldsymbol{x}_i; \boldsymbol{\theta})|| ||\lambda_L(\boldsymbol{x}_j; \boldsymbol{\theta})||}.$$

The CK is an interesting comparison for a couple of reasons: first, for any network that ends in a fully connected layer, the CK is actually an additive component of the trNTK; therefore, we can evaluate whether a smaller amount of the total trNTK can accomplish the same task. Second, the CK is computed from the final feature vector before a network makes a decision; the NN is exactly a linear model with respect to this final feature vector. NN architectures typically contain bottlenecks that project down to this final feature vector. These projections remove information. While that information might be of no use to the classification task, it may be useful for the attribution task. We can think of the the final information presented to the NN as the CK, and the information contained before these projections as the trNTK, though more work is needed to formalize and explore this comparison.

**Unnormalized Pseudo Neural Tangent Kernel** To evaluate the effect of the normalization in the trNTK definition we will evaluate the kernel without normalizing. let the unnormalized trNTK be defined as:

$$\text{trNTK}^0(\boldsymbol{x}_i, \boldsymbol{x}_j) = \boldsymbol{g}(\boldsymbol{x}_i; \boldsymbol{\theta})^\top \boldsymbol{g}(\boldsymbol{x}_j; \boldsymbol{\theta}).$$

While neural tangent kernels are not typically cosine-normalized kernels we were drawn to investigate such normalized kernels for a few reasons: Akyürek et al. (2022) remarked that cosine normalization

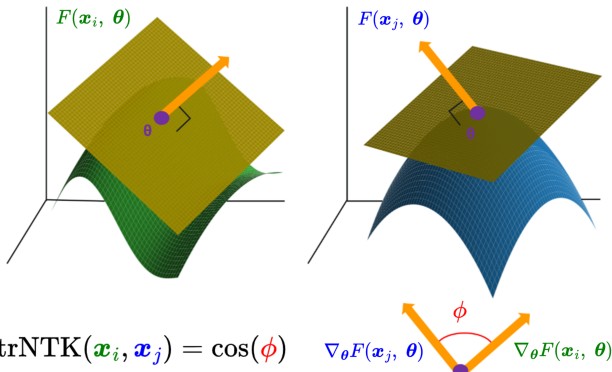

$$\text{trNTK}(\boldsymbol{x}_i, \boldsymbol{x}_j) = \cos(\phi)$$

**Figure 3: Geometric intuition behind the** trNTK. A NN function is evaluated at two points creating surfaces $F(\boldsymbol{x}_i\,;\boldsymbol{\theta})$ and $F(\boldsymbol{x}_j\,;\boldsymbol{\theta})$. These surfaces are shown with a tangent hyper plane at the same point ($\boldsymbol{\theta}$) in parameter space coinciding with the end of training. The Jacobian vector defines the tangent hyperplane's orientation in parameter space. The trNTK is a kernel whose $(i, j)$-th element is the cosine angle between averaged Jacobian vectors. The more similar the local geometry between $\boldsymbol{x}_i$ and $\boldsymbol{x}_j$ local to $\boldsymbol{\theta}$ in parameter-space , the higher the value of trNTK($\boldsymbol{x}_i, \boldsymbol{x}_j$).

could prevent training data with large magnitude Jacobian vectors from dominating the kernel, and Hanawa et al. (2021) notes a cosine-similarity kernel achieves the best performance among alternative kernels on a data attribution task. Key motivators for our study included that the cosine normalized values are intuitive geometrically, and that it is standard practice to ensure feature matrices such as $\boldsymbol{\kappa}$ are in a small range (such as [-1,1]) for machine learning.

## C  GEOMETRIC INTUITION BEHIND NEURAL TANGENT KERNELS

In figure 3 we provide a pictorial representation of the geometric interpretation behind the trNTK.

## D  ADDITIONAL DETAILS REGARDING THE TRACE NEURAL TANGENT KERNEL

In this Appendix we provide an expanded definition of the trNTK that highlights how the trNTK is actually computed from a series of individual contributions from each learnable tensor. This layerwise decomposition has been pointed out in previous work (Novak et al., 2022). Let $\boldsymbol{\theta}_l$ be the parameter vector consisting of only the parameters from the $l$-th layer. Let the number of parameters in the $l$-th layer be $p_l$. A Jacobian is a vector of first-order partial derivatives of the NN with respect to the parameters. We will specify each Jacobian through the $c$-th scalar function (equivalently, $c$-th output neuron) for the parameters in the $l$-th layer as:

$$\mathbf{g}_l^c(\boldsymbol{x}_i) = \frac{\partial F(\boldsymbol{x}_i\,;\boldsymbol{\theta})}{\partial \boldsymbol{\theta}_l} \in \mathbb{R}^{1 \times P_l}. \tag{9}$$

Note that we have intentionally broken our notation for the vector by using the Gothic capital $\mathbf{g}$ for the Jacobian vector. We do this to avoid confusion with the lowercase $j$ used as an index. Let $\mathbf{g}_l(\boldsymbol{x}_i)$ be the concatenation of all such $\mathbf{g}_l^c(\boldsymbol{x}_i)$ for all $c \in \{1, 2, \dots, C\}$:

$$\mathbf{g}_l(\boldsymbol{x}_i) = \left[\mathbf{g}_l^1(\boldsymbol{x}_i), \mathbf{g}_l^2(\boldsymbol{x}_i), \dots, \mathbf{g}_l^C(\boldsymbol{x}_i)\right] \in \mathbb{R}^{1 \times CP_l}. \tag{10}$$

Let $\boldsymbol{J}_l(\boldsymbol{X})$ be the matrix formed from column vectors $\mathbf{g}_l(\boldsymbol{x}_i)^\top$ over each training data point $\boldsymbol{x}_i$, where $i \in \{1, 2, \dots, N\}$:

$$\boldsymbol{G}_l(\boldsymbol{X}) = \left[\mathbf{g}_l(\boldsymbol{x}_1)^\top, \mathbf{g}_l(\boldsymbol{x}_2)^\top, \dots, \mathbf{g}_l(\boldsymbol{x}_N)^\top\right] \in \mathbb{R}^{CP_l \times N}. \tag{11}$$

Let the $l$-th unnormalized pseudo-Neural Tangent Kernel, or $\mathrm{trNTK}_l$, be the Gram matrix formed from the products of $\boldsymbol{J}_l(\boldsymbol{X})$ matrices:

$$\mathrm{trNTK}_l^0 = \boldsymbol{G}_l(\boldsymbol{X})^\top \boldsymbol{G}_l(\boldsymbol{X}) \in \mathbb{R}^{N \times N}. \tag{12}$$

As a Gram matrix, $\mathrm{trNTK}_l^0$ is symmetric and positive semi-definite. Let $\mathrm{trNTK}^0 \in \mathbb{R}^{N \times N}$ be the matrix formed from summing the contributions from all $\mathrm{trNTK}_l^0$. Consider

$$\mathrm{trNTK}^0 = \sum_{l=1}^{L} \mathrm{trNTK}_l^0 \in \mathbb{R}^{N \times N}. \tag{13}$$

Here, $\mathrm{trNTK}^0$ itself is symmetric, as the sum of symmetric matrices is symmetric. Finally, we must apply the normalization. Let the matrix B be defined as the element-wise product of the $\mathrm{trNTK}$ with the identity:

$$\boldsymbol{B} = \boldsymbol{I} \odot \mathrm{trNTK}^0. \tag{14}$$

Then the normalized $\mathrm{trNTK}$ can be computed form the unnormalized $\mathrm{trNTK}$ by the following relationship:

$$\mathrm{trNTK} = \boldsymbol{B}^{\frac{-1}{2}} \, \mathrm{trNTK}^0 \, \boldsymbol{B}^{\frac{-1}{2}}. \tag{15}$$

The relationship between the full neural tangent kernel and the $\mathrm{trNTK}$ is described in Appendix E.

## E  RELATIONSHIP TO THE EMPIRICAL NTK

To calculate the full eNTK, first find the $c$-th class Jacobian vector, $\mathbf{g}^c$, with respect to $\boldsymbol{\theta}$ backwards through the network for each $\boldsymbol{x}_i$ in the data matrix $\boldsymbol{X}$. Explicitly, the $c$-th logit's Jacobian $i$-th column-vector corresponds to datapoint $\boldsymbol{x}_i$ and is defined:

$$\mathbf{g}^c(\boldsymbol{x}_i) = \frac{\partial F^c(\boldsymbol{x}_i, \boldsymbol{\theta})}{\partial \boldsymbol{\theta}}. \tag{16}$$

From which we can define the Jacobian matrix as:

$$\boldsymbol{G}^c = [\mathbf{g}^c(x_0), \, \mathbf{g}^c(x_1), \, \ldots, \, \mathbf{g}^c(x_N)] \tag{17}$$

The eNTK is the block-matrix whose (k,j)-th block, where both $k, j = \{1, 2, \ldots, C\}$, is the linear kernel formed between the Jacobians of the (k,j)-th logits:

$$\mathrm{NTK}_{k,j} = (\boldsymbol{G}^j)^\top (\boldsymbol{G}^k). \tag{18}$$

The NTK is therefore a matrix $\in \mathbb{R}^{CN \times CN}$. The relationship between the unnormalized $\mathrm{trNTK}$ and the NTK is simply

$$\mathrm{trNTK}^0 = \sum_{c=1}^{C} \mathrm{NTK}_{c,c}. \tag{19}$$

We chose to study the $\mathrm{trNTK}$ instead of the NTK for simplicity, computational efficiency, and reduced memory footprint. Follow on work could attempt to use the entire NTK to form the surrogate models. We were additionally motivated by the approach taken in Chen et al. (2022) and Chen et al. (2022), and we refer the reader to Mohamadi & Sutherland (2022) for a deeper discussion of the qualities of similar approximations.

## F  NOTES ON THE PROJECTED VARIANTS OF THE NTK

For TRAK, Park et al. (2023) utilized the Johnson Lindenstrauss lemma (Johnson & Lindenstrauss, 1984) to justify the use of the projection matrix K. The Johnson Lindenstrauss lemma bounds the error between any two vectors and the same two vectors projected under a projection matrix $\boldsymbol{P}$. The lemma can be used to show a bound on the cosine similarity between two vectors and two projected vectors (Lin et al., 2019). However, this bound relates the probability of the residual for all vectors being less than some small $\epsilon$. From an applied perspective we might care only that the residuals of cosine similarity are small with high probability. We empirically observe that the

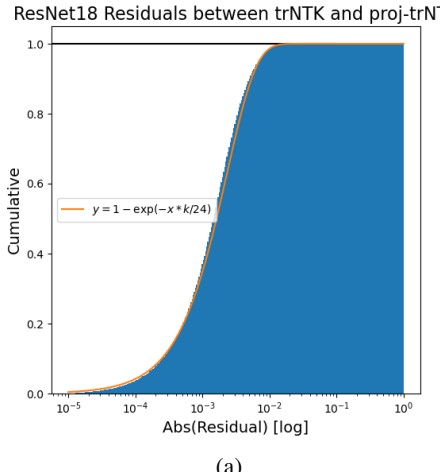 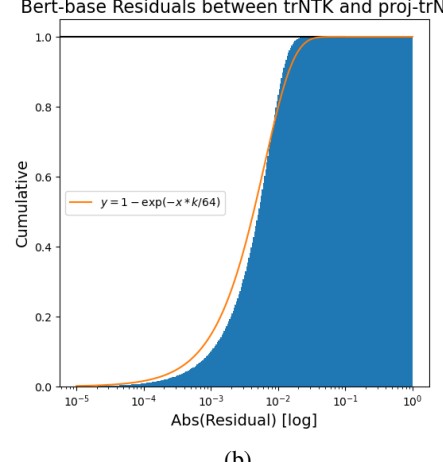

(a)                                                           (b)

Figure 4: **trNTK and** proj-trNTK **cosine-similarity residuals fall exponentially.** For both ResNet18 Eq. 4a and Bert-base Eq. 4b we plot the cumulative histogram of residuals between the trNTK and proj-trNTK. The orange line is an exponential function with k=10240. The orange line is fit "by eye" rather than some best-fit, the objective being to reference the exponential shape of the residual distribution.

absolute residuals of of the trace-NTK and proj-trNTK fall away as $\exp(-x * \beta)$, where $\beta$ is the decay rate. In Figures 4a and 4b, we show the residuals for our ResNet18 and Bert-base experiments, with an overlaid exponential decay model for reference. We are unaware of a formal proof that would dictate the form of the distribution of residuals, but we use these plots to empirically justify the exploration of the projected-variants as close approximations for the original kernels with large enough K. Intuitively, we expect that there is a trade-off between size of the dataset, size of the model, and K.

## G  FORMAL DEFINITION OF EVALUATION METRICS

In this Appendix we restate all the metrics used throughout this study.

**Kendall-$\tau$ rank correlation**

For a paired sequence $S_\tau = \{(a_1, b_1), \ldots, (a_N, b_N)\}$ a pair $(a_i, b_i)$ and $(a_j, b_j)$ with $i \neq j$ are concordant if either both $a_i > a_j$ and $b_i > b_j$ or $a_i < a_j$ and $b_i < b_j$. Otherwise, the pair is discordant. We count the total number of concordant, NC, and number of discordant pairs, ND. Then, $\tau_K$ is defined as

$$\tau(S_\tau) = \frac{(\text{NC} - \text{ND})}{\text{NC} + \text{ND}}.$$

**Test accuracy differential (TAD)**  We track the test accuracy differential, or TAD, given by the difference between the kGLM and NN's test accuracy,

$$\text{TAD} = \text{TestAcc}_{\text{kGLM}} - \text{TestAcc}_{\text{NN}}, \tag{20}$$

to demonstrate that kGLM have similar performance to the underlying NN. A value of $0$ is preferred.

**Misclassification Coincidence Rate**  we compute the intersection of misclassifications between each kGLM model and the NN where both the NN and kGLM predict the same class, over the union of all misclassifications of either the NN or kGLM models as a decimal. A value of 1.0 indicates that in all cases where the NN is wrong, the kGLM is also wrong and predicts the same class as the NN.

$$R_{\text{Miss}} = \frac{|\{f(\boldsymbol{x}_i, \boldsymbol{\theta}) \neq \boldsymbol{z}_i\} \cap \{\text{kGLM}(\boldsymbol{x}_i) \neq \boldsymbol{z}_i\} \cap \{f(\boldsymbol{x}_i, \boldsymbol{\theta}) = \text{kGLM}(\boldsymbol{x}_i)\}|}{|\{f(\boldsymbol{x}_i, \boldsymbol{\theta}) \neq \boldsymbol{z}_i\} \cup \{\text{kGLM}(\boldsymbol{x}_i) \neq \boldsymbol{z}_i\}|}. \tag{21}$$

**Precision and Recall** To evaluate whether our attributions are performant at discriminating between perturbed and unperturbed test datapoints, we use precision as a measure of how valid the flags given by our attribution model are, and recall as a measure of how complete these attributions were at identifying poisoned test data. A perfect model would have both precision and recall $= 1$. Precision and recall are defined:

$$\text{Precision} = \frac{\text{TP}}{(\text{TP} + \text{FP})}$$
$$\text{Recall} = \frac{\text{TP}}{(\text{TP} + \text{FN})},$$

where TP is the true positive rate, FP is the false positive rate, and FN is the false negative rate.

**Coefficient of Determination $R^2$** The coefficient of determination is used as a goodness-of-fit to assess the viability of our linearization of the NN (described below in Appendix L). It is possible to have a high $\tau_K$ but small $R^2$ if the choice of invertible mapping function is wrong or if the fit of said function does not converge. Such cases can be inspected visually to determine the relationship between the logits.

For a sequence of observations (in the context of this paper, the natural logarithm of probability of the correct class for the NN and kGLM) $S_{R^2} = \{(x_1, y_1), \ldots, (x_N, y_N)\}$, let the sample average of the $y_i$ observations be $\bar{y} = \frac{1}{N} \sum_i^N y_i$. Then let the total sum of squares be $\text{SS}_{\text{tot}} = \sum_i^N (y_i - \bar{y})^2$, and the sum of squared residuals be $\text{SS}_{\text{res}} = \sum_i^N (y_i - x_i)^2$. Then let the goodness-of-fit $R^2$ function be defined by

$$R^2(S_{R^2}) = 1 - \frac{\text{SS}_{\text{res}}}{\text{SS}_{\text{tot}}}.$$

## H    ALTERNATIVE MEASURES OF CORRELATION

To justify our choice of Kendall-$\tau$ as the measure of correlation, we compare to other choices of correlation, the Pearson-R and Spearman-$\rho$. We wrote that Pearson-R is unsuitable as a measure of correlation because it conflates the covariance between models with the correlation between models. Consider the thought experiment to see this is true: $F_A$ and $F_B$ are independent models, both of which for any input $X_i$ are correct at a rate $P_A$ and $P_B$, with $P_A$ and $P_B$ nearly one. When the models are correct, the output is $Y_i + N(0, \sigma)$, with $Y_i \in \{0, 1\}$, and when incorrect are $|Y_i - 1| + N(0, \sigma)$. Furthermore, assume an even class distribution, and that $\sigma \ll 1$. The result of the paired set of evaluations from $F_A$ and $F_B$ is a point cloud with most points centered at 0 and 1, as in figure 5. Because both models are correct with high probability, the probability that $F_B$'s output is centered at zero is high if $F_A$'s output is centered at zero; likewise, the probability that $F_B$'s output is centered at one is high if $F_A$'s output is centered at one. These point clouds act as anchor points that sway the Pearson-R correlation to values of 1, even though there is no real coupling between the models. To the point: because the kGLM and NN are highly performant models, we must distinguish from correlation from this fact and their independence, from true kGLM dependence on the NN itself. While rank-based correlations are sensitive to this phenomena, the expected value of Kendall-$\tau$ would only be 0.5 in this experiment.

To complete the thought experiment, consider if $F_B$ is dependent on $F_A$: $F_B(X_i) = F_A(X_i) + N(0, \sigma)$ (visualized in right panel of figure 5). In the limit $\sigma \to 0$, we would like to choose the correlation measure that most slowly converges to 1. This is because we want to maximize the interval over which out faithfulness measure discriminates between models. We complete the numerical experiment and visualize the result in figure 6, showing the Kendall-$\tau$ converges to value one slowest.

## I    USE THE NN OUTPUTS FOR THE KGLM TARGETS

To evaluate our methodology of training the kGLM using the ground truth labels, we compare to training using the neural network model output as the label for the kGLM. This is a reasonable choice, framing the surrogate model's learning as a teacher-student model. Contemporaneous work investigates kernel based data attribution using this method, (Tsai et al., 2023). We report the result of this experiment in table 5. Compared to our methodology (table 2), We generally see an training with

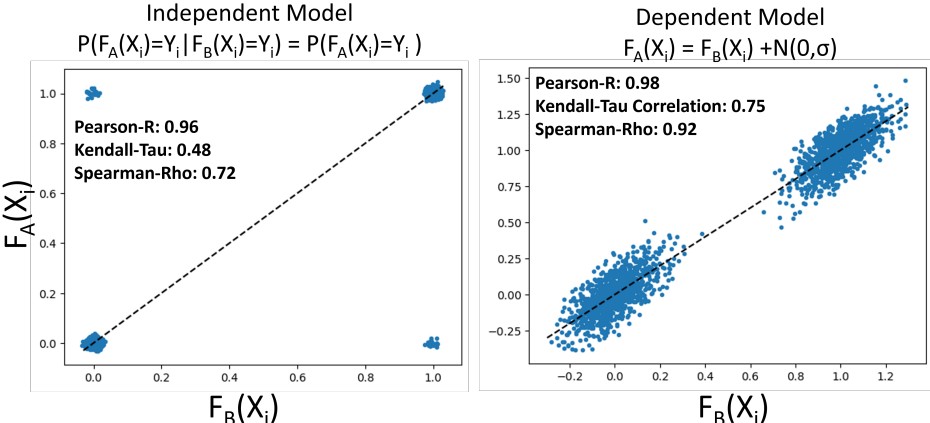

Figure 5: **Distinguishing between independence but high covariance from true dependence** Left: Plotting the confidence-confidence scatter plot using two independent models which both have a high probability of correct classification results in a point cloud with high density at (0,0) and (1,1). These point clouds act as anchors that force the Pearson correlation measure to be nearly 1, but because there is no underlying structure the rank-correlation $\tau$ is only 0.5. Right: We visualize the dependent case, which is an ideal form of our surrogate model definition. We see that the anchor point structure is still present forcing the Pearson to be nearly 1, and now the rank correlation $\tau$ has grown to 0.75. Our main point is that Kendall-$\tau$ is not so affected by the issue of separating covariance from dependence as Pearson.

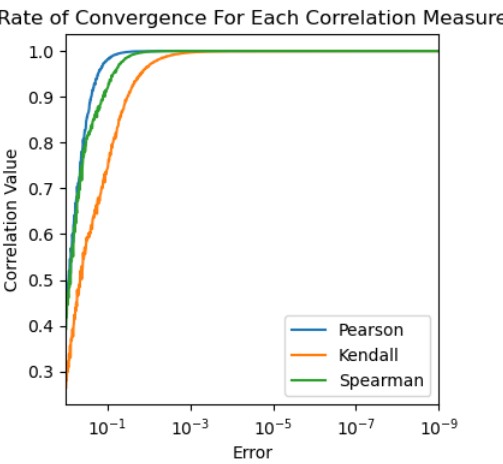

Figure 6: **Comparison of Rate of Convergence of Correlation Measures.** Using the dependent models thought experiment, we reduce the $\sigma$, or error (x-axis), and plot the correlation value. The Kendall-$\tau$ is the slowest to converge to values of 1, meaning its the most sensitive measure of correlation over the interval studied.

Table 5: **Using the NN outputs as labels for training kGLM.** We report our modified experiment results for ResNet18 and Bert-base. For the Bert-base model where multiple models are trained, we report the leading digit of the standard error of the mean as a parenthetical.

| Exp Name | Metric | $\kappa$ | | | | | |
|---|---|---|---|---|---|---|---|
| | | trNTK | trNTK$^0$ | proj-trNTK | proj-pNTK | Em | CK |
| ResNet18 | $\tau_K$ | 0.44166 | * | 0.4443 | 0.6707 | 0.47159 | 0.62874 |
| | TAD (%) | -0.66 | * | -0.68 | -0.01 | -0.18 | -0.02 |
| Bert-base | $\tau_K$ | 0.50(3) | 0.31(2) | 0.50(3) | 0.43(4) | 0.40(4) | 0.38(4) |
| | TAD (%) | 0(2) | 0.1(2) | 0(2) | -0.2(1) | -0.9(2) | -0.3(2) |

the original ground truth labels increases Kendall-$\tau$. We speculate this is because the optimization problem are shared between the kGLM and the NN training if the original ground truth labels are utilized.

## J  ADVERSARIAL ATTACKS

We trained NN models on the MNIST dataset. In order to avoid combinatorial considerations, the classifier was trained on just two classes– we used 7's and 1's because these digits look similar. Subsequently, we extracted the NTKs and used these kernels to train SVMs. To attack both types of models, we considered $\ell_\infty$ perturbations, computed using the projective gradient descent algorithm (Madry et al., 2019) with 7 steps (PGD-7). Our experiments leverage PyTorch's auto-differentiation engine to compute second-order derivatives to effectively attack the SVMs. In contrast, prior work (Tsilivis & Kempe, 2023) derived an optimal one-step attack for the NTK at the limit and and used this approximation to compute adversarial examples. To compare neural nets with kernel regression, (Tsilivis & Kempe, 2023) compute the cosine similarity between the FGSM adversarial attack and the optimal 1-step attack for kernel machine, computed analytically by taking the limit for an infinitely wide neural net. Their results show (Figures 3 and 7 of (Tsilivis & Kempe, 2023)) that throughout training, the cosine similarity of this optimal 1-step attack and the empirical attack on the neural net decreases. This observation suggests that in practice, the NTK limit is not a good surrogate model for a neural net under an adversarial attack. Our plots (Figure 7) confirm this observation as SVMs are much more vulnerable to attacks that the associated neural nets. To better compare with prior work, we trained our SVMs using NTKs rather than pNTKs.

In considering security of neural nets, attacks are categorized as either *white-box* or *black-box*. White-box attacks assume that the adversary has access to all the weights of a neural net while black box attacks do not assume that an adversary has this information. A common strategy for creating a black box attack is training an independent NN and then using perturbations calculated from attacking this new NN to attack the model in question. Such attacks are called *transfer attacks*; see (Papernot et al., 2016b;a) for examples of successful black-box and transfer attacks.

In line with this framework, we test our models against two white-box attacks and a black box attack. First, we test neural nets and SVMs by directly attacking the models. Next, to better understand the similarities between a neural net and the associated SVM, we evaluate the SVM on attacks generated from the associated neural net and the neural net on attacks generated from the associated SVM. For the black box attacks, we test: 1) neural nets on adversarial examples generated from independently trained neural nets, 2) SVMs on adversarial examples from SVMs trained with an NTK from an independently trained neural net, 3) Neural nets on adversarial examples from SVMs trained with an NTK from and independently trained neural net, 4) SVMs on adversarial examples from independently trained neural nets.

The error bars for all three figures are on 10 trials. For the black box figure, each model was tested against 9 other independently trained models; the plotted quantities are the average of all these black box attacks.

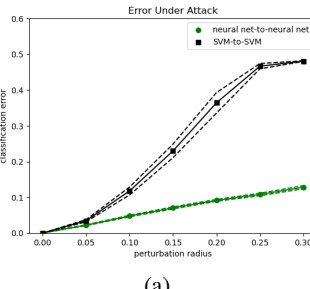 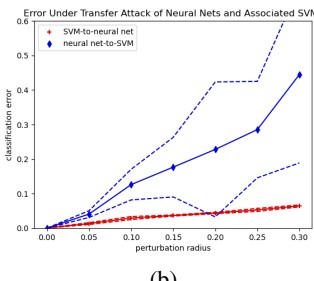 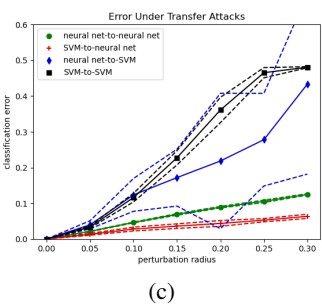

      (a)                        (b)                       (c)

Figure 7: **Error Under Adversarial Attacks: Eq. 7a** White-box attack: Attacking a neural net and the associated NTK SVM directly. **Eq. 7b** White-box attack: Attacking a neural net using perturbed examples for the associated SVM and attacking an NTK SVM by using perturbed examples for the associated neural net. **Eq. 7c** Black-box attack: Attacking neural nets and SVMs using perturbed examples from independently trained SVMs and neural nets. This demonstrates a limitation of our surrogate model method: we find that the SVM's performance does not scale the same as the NN's performance with increasing perturbation radius, across multiple kinds of attack.

### J.1 ADVERSARIAL EXPERIMENT DETAILS

When performing PGD to find adversarial examples to our models, we did not restrict pixel values of the attacked images to the interval $[0, 1]$. See (Madry et al., 2019) for more information on using the PGD algorithm in an adversarial context. Notice that in the PGD algorithm, attacking the SVM trained with the NTK involves computing second derivatives of the neural net. Due to this consideration, using ReLUs as neurons in this experiment was impractical– the second derivative of a piecewise linear function at any point is either zero or non-existent. Hence the nets are constructed from sigmoid neurons.

The model architecture was 3 fully connected layers of 100 neurons. The models were trained for 100 epochs and with learning rate $10^{-4}$ with AdamW optimizer and minibatches of size 64 in PyTorch on the cross-entropy loss. The error bars in both Figures 7a and 7b figures are computed from the standard deviation calculated from 10 independent experimental trials set with different random seeds.

The SVMs were trained using `sklearn`'s SVM package.

## K ADDITIONAL EXPERIMENTAL DETAILS

In this Appendix we detail the specific choice of architecture, hyperparameters, and training times for each experiment.

### K.1 DATASETS

Our experiments utilize common benchmark datasets: MNIST, FMNIST, CIFAR10, and COLA. we will quickly introduce each in turn. The Modified National Institute of Science Technology (MNIST) (Lecun et al., 1998) handwritten digit database is a grey-scale image classification dataset comprised of handwritten numerical digits and label pairs created from combining the National Institute of Science and Technology special datasets 1 and 3. MNIST has over 50,000 training and 10,000 testing data-label pairs. "Fashion"-MNIST (FMNIST) (Xiao et al., 2017) is another image classification dataset that was specifically introduced to serve as drop in replacement to MNIST. It was created by reducing images from an online European fashion catalogue to the same 28x28 pixel resolution as MNIST and to grey-scale. FMNIST has 10 classes of different kinds of garments, with 7,000 examples of each garment, split into 60,000 training and 10,000 test data. Canadian Institute for Advanced Research-10 (CIFAR10) is a 10-class supervised image classification dataset comprised of 32x32 pixel 3-color channel hand-labeled subset of the TinyImages dataset (Torralba et al., 2008) featuring everyday objects and animals. CIFAR10 is composed of 50,000 training and

10,000 test data, evenly split among the 10 classes. Finally, the Corpus of Linguistic Acceptability (CoLA) (Warstadt et al., 2018) is a dataset composed of sentences and labels corresponding to the grammatical correctness of the sentence compiled from texts on grammar. CoLA includes 9515 training sentences and 1049 test sentences. CoLA was included in the original GLUE (Wang et al., 2019) set of benchmarks for NLP, which became the de-facto benchmark set of tasks for general language modeling.

## K.2 Experiments

### K.2.1 100 Fully Connected MNIST2 Models

Using the first two classes of MNIST, (MNIST2), we train 100 independent 4-layer fully connected NNs using PyTorch. The network layer widths were [100,100,100,1], and each had a Rectified Linear Unit (ReLU) activation function, except for the final layer. We define all of our networks to terminate without a final activation for the sake of calculating our $\mathrm{trNTK}$; however, we use the sigmoid link function to map the activations onto a value we interpret as probability of class 1. As is typical in NTK parameterization, we divided each activation map by the square root of the preceding layer's width. The input space of MNIST was modeled as a 784-size feature vector that we preprocessed to have values between 0 and 1 by dividing by the maximum pixel value of 255. For simplicity, we down sampled the test dataset to share an equal amount of class 0 and class 1 examples, giving 980 examples of each class. We initialized the layers using the normal distribution.

Each model instance had the same hyperparameters, architecture, and approximate training time. The only differences were the initialization given by seed and the stochastic sequence of datapoints from a standard PyTorch data-loader. We trained our model to minimize the binary cross entropy loss between the true labels and the prediction function. We chose to optimize our model using stochastic gradient descent with no momentum and static learning rate 1e-3. Training 100 models sequentially takes approximately 8 hours on a single A100 GPU.

### K.2.2 100 CNN MNIST2, FMNIST2, and CIFAR2 Models

We use the same CNN architecture for our 100 MNIST2, FMNIST2, and CIFAR2 models; for brevity, we will describe the model once. Each model is a 12-layer NN where the first 9 layers are a sequence of 2D convolutional layers and 2D Batch Normalization layers. The final 3 layers are fully connected. The first nine layers are split into three sections operating on equal feature map sizes (achieved with padding). The first layer in each section is a convolutional layer with kernel size 3 and padding size 1 followed by a batch normalization layer, followed by a second convolutional layer with kernel size 3 and padding size 1 but with stride = 2 to reduce the feature map in half. The number of filters steadily increases throughout each convolutional layer as [8,8,16,24,32,48,64]. After the convolutional layers, a flattening operation reduces the image dimensions into a 1-dimensional vector. Next, fully connected layers of widths [256, 256, 1] are applied. After each convolutional layer and fully connected layer we apply the rectified linear unit (ReLU) activation. Training times for 100 models on MNIST2, CIFAR2, and FMNIST 2 were 15 hours (100 epochs), 5 hours (100 epochs), and 48 hours (200 epochs), respectfully, on a single A100 GPU. The difference in times can be explained by the different choices of batch size and number of epochs, which were 4, 64, and 4, respectfully. We chose these batch sizes, and all other hyperparameters, by hand after a small search that stopped after achieving comparable performance the many examples of models available online for these benchmark tasks. One oddity we believe worth mentioning is that we subtract the initial model's final activation vector for the CIFAR2 model, after observing that this lead to a modest improvement. Initial LRs were 1e-3 for each model, but the optimizers were chosen as SGD, Adam, and Adam for MNIST2, CIFAR2, and FMNIST2, respectfully.

### K.2.3 4 COLA BERT-base Models

To train the 4 BERT-base models, we downloaded pre-trained weights available on the HuggingFace repository for BERT-base no capitalization. We then replaced the last layer with a two-neuron output fully connected layer using HuggingFace's API for classification tasks. We set different seeds for each model instance, which sets the random initialization for the final layer. We train our model on the COLA dataset for binary classification of sentence grammatical correctness. We train our model using the the AdamW optimizer (Loshchilov & Hutter, 2017) with an initial learning rate $\eta$

= 2e-5. We allow every layer to update. Training is done over 10 epochs after which the training accuracy is seen to exceed 99% performance on each model. Training takes a few minutes on an A100 GPU. Calculating the NTK is achieved by splitting the parameter vector into each learnable tensor's contribution, then parallelizing across each tensor. Each tensor's trNTK computation time depends upon the tensor's size. In total the computation takes 1200 GPU hours, on single A100 GPUs.

### K.2.4 LARGE COMPUTER VISION MODELS

We downloaded 3 pre-trained model weights files from an independent online repository (Phan, 2021). ResNet18 and Resnet34 architectures can be found described in He et al. (2015b), and MobileNetV2 can be found described in Sandler et al. (2018). Each model's trNTK was computed by parallelizing the trNTK computation across each learnable tensor. the computation time varies as a function of the learnable tensor's size, but the total time to compute each of ResNet18, ResNet34, and MobilenetV2 was 389, 1371, and 539 GPU hours, respectfully, on single A100 GPUs.

### K.2.5 CNN FOR POISONED DATA EXPERIMENT

We trained a 22 layer CNN with architecture described in the repository alongside Shan et al. (2022) and restated here. The architecture's first 15 layers are composed of a 5 layer repeating sequence of convolution, batch normalization, convolution, batch normalization, and max pooling. After the 15th layer, we flatten the feature vector, apply another max pooling operation, and then apply dropout with probability 0.2. The next parameterized layers consist of the sequence fully connected layer, batch normalization, fully connected layer, batch normalization and final fully connected layer. A ReLU activation is applied between each hidden layer. The repository of Shan et al. (2022) generates BadNet cifar10 images as a data artifact. We translate their architecture to PyTorch and train our own model. The model was trained to minimize the cross entropy loss on the poisoned image dataset with stochastic gradient descent with an initial learning rate of 1e-2. The total number of parameters for this model is 820394. We take a different approach to calculate the trNTK of this model and choose not to parallelize the computation across each learnable tensor. The total trNTK calculation completed in 8 hours on a single A100 GPU.

### K.3 COMPUTING EMBEDDING KERNELS

To compute an embedding kernel we must make a choice of what constitutes a "layer". This has some slight nuance, as for example, the most complete Embedding kernel would be computed after every modification to the feature space. In a typical fully connected layer there would be 2-3 modifications that occur: 1) the weight matrix multiplication; 2) the bias vector addition; 3) the activation function. Typically, we would take each of these modifications as part of the same fully connected layer and sample an activation for the Embedding following all three. Next, consider residual blocks and similar non-feed forward or branching architectures. We must make a choice of where to sample in the branch that may have an impact on how the final Embedding kernel behaves. In this Appendix, we list our choice of layers to sample the activation for each experiment. We chose to balance completeness and computation time. Follow on work could investigate how these choices affect the final embedding kernel.

### K.3.1 RESNET18

Table 6 shows where the components of the embedding kernel were calculated.

### K.3.2 BERT-BASE

The layers used to calculate Bert-base embedding kernel are shown in Table 7.

### K.3.3 POISONED CNN

Table 8 shows after which modules the embedding kernel was calculated for the data poisoning CNN.

Table 6: Embedding Layers ResNet18 with $x \in \{1, 2, 3, 4\}$

| Layername |
| --- |
| conv1 |
| bn1 |
| maxpool |
| layer.$x$ |
| layer.$x$ .0 |
| layer.$x$.0.conv1 |
| layer.$x$.0.bn1 |
| layer.$x$.0.conv2 |
| layer.$x$.0.bn2 |
| layer.$x$.1 |
| layer.$x$.1.conv1 |
| layer.$x$.1.bn1 |
| layer.$x$.1.conv2 |
| layer.$x$.1.bn2 |
| avgpool |
| fc |

Table 7: Bert-base Layers with Embedding Kernel calculation, $x \in \{0, 1, 2, 3, 4, 5, 6, 7, 8, 9, 10, 11\}$

| Layername |
| --- |
| bert.embeddings |
| bert.embeddings.word_embeddings |
| bert.embeddings.position_embeddings |
| bert.embeddings.token_type_embeddings |
| bert.embeddings.LayerNorm |
| bert.encoder.layer.x |
| bert.encoder.layer.x.attention |
| bert.encoder.layer.x.attention.self |
| bert.encoder.layer.x.attention.self.query |
| bert.encoder.layer.x.attention.self.key |
| bert.encoder.layer.x.attention.self.value |
| bert.encoder.layer.x.attention.output |
| bert.encoder.layer.x.attention.output.dense |
| bert.encoder.layer.x.attention.output.LayerNorm |
| bert.encoder.layer.x.intermediate |
| bert.encoder.layer.x.intermediate.dense |
| bert.encoder.layer.x.intermediate.intermediate_act_fn |
| bert.encoder.layer.x.output |
| bert.encoder.layer.x.output.dense |
| bert.encoder.layer.x.output.LayerNorm |
| bert.pooler |
| bert.pooler.dense |
| classifier |

Table 8: Embedding Layers Poisoned CNN

| Layername |
| --- |
| conv2d |
| batch_normalization |
| conv2d_1 |
| batch_normalization_1 |
| max_pooling2d |
| conv2d_2 |
| batch_normalization_2 |
| conv2d_3 |
| batch_normalization_3 |
| max_pooling2d_1 |
| conv2d_4 |
| batch_normalization_4 |
| conv2d_5 |
| batch_normalization_5 |
| max_pooling2d_2 |
| max_pooling1d |
| dense |
| batch_normalization_6 |
| dense_1 |
| batch_normalization_7 |
| dense_2 |

## L  METHODOLOGY FOR LINEARIZING NNS VIA kGLMS

We describe the procedure to achieve a linearization of the NN via a kGLM surrogate model. First, we fit a supervised NN using standard techniques. Next, we compute the trNTK. This kernel acts as the feature space of the kGLM. we fit the kGLM (Pedregosa et al., 2011) (`sklearn.linear_model.SGDClassifier`) using the kernels computed from the same training data as the NN is trained upon. The dimensionality of the output vector from the kGLM will be the same as the NN, and is equal to the number of classes.

We are concerned with demonstrating that after applying an invertible mapping function $\Phi$, the NN decision function is approximately equal to the kGLM decision function. Because the decision function is typically only a function of the probabilities of each class, this objective can be achieved by showing the following approximation holds:

$$\sigma(F(\boldsymbol{x}\,;\boldsymbol{\theta})) \approx \Phi(\text{kGLM}(\boldsymbol{x})).$$

Across many models and datasets we generally observed that the trend between the NN activation and the kGLM activation was "S-shaped", or else was already linear. The analytic class of function that are "S-shaped" are sometimes called sigmoid functions. The following three functions are used to map the kGLM to the NN.

$$\Phi^1(x) = \nu x + \mu,$$

$$\Phi^2(x) = \nu \frac{\exp(\frac{x-\alpha}{\beta})}{1 + \exp(\frac{x-\alpha}{\beta})} + \mu$$

$$\Phi^3(x) = \frac{\nu}{\pi} \arctan\left(-\frac{x-\alpha}{2\beta}\right) + \frac{1}{2} + \mu.$$

$\Phi^1$ is a linear re-scaling. Both $\Phi^2$ and $\Phi^3$ are sigmoid-shaped functions that map $(-\infty, \infty)$ to (0,1). All choices of $\Phi$ are invertible. We made these choices for $\phi$ after observing the relationship between the kGLM and the NN. We fit $\Phi$ functions with an iterative optimizer (Virtanen et al., 2020) on the $L_2$ loss between $F(\tilde{\boldsymbol{X}}\,;\boldsymbol{\theta})_c)$ and $\Phi(\text{kGLM}(\boldsymbol{X})^c)$, where $c$ is chosen to be class 1 in the case of binary classification (we describe changes necessary for multi-classification below). Fits are completed over a partition of half the test dataset and evaluated on the remaining half. The linearizations are visualized in Appendix L.1.

To visualize we use scale using the logit function. We define the logit function as the scalar-valued function that acts on the softmax probability $p \in (0,1)$ of a single class and outputs a "logit":

$$\text{logitfn}(\boldsymbol{x}) = \log \frac{\boldsymbol{x}}{1 - \boldsymbol{x}}.$$

Using the logit creates a better visualization of the probabilities themselves by smoothing out the distribution of values across the visualized axes. As a final implementation note, we observed some numerical instability due to values being so close to p=1 that errors occur in re-mapping back into logits. We choose to mask out these values from our fit, our visualization, and the $R^2$ metric.

### L.1  VISUALIZATIONS OF POINT-FOR-POINT LINEAR REALIZATIONS FOR EACH EXPERIMENT

What follows is the visualization of the linearizations of the NN logits with respect to the kGLM logits. A perfect fit would line up with parity, shown as a diagonal dashed line in each plot. The coefficient of determination or $R^2$ is shown in the text for each plot. Seeds are shown in each panel's title. For the classification models ResNet18, ResNet34, and MobileNetV2, we flatten out the regressed vector and choose to plot the distribution as a KDE estimate of the correct class and incorrect classes instead of a scatter plot, due to the large number of points.

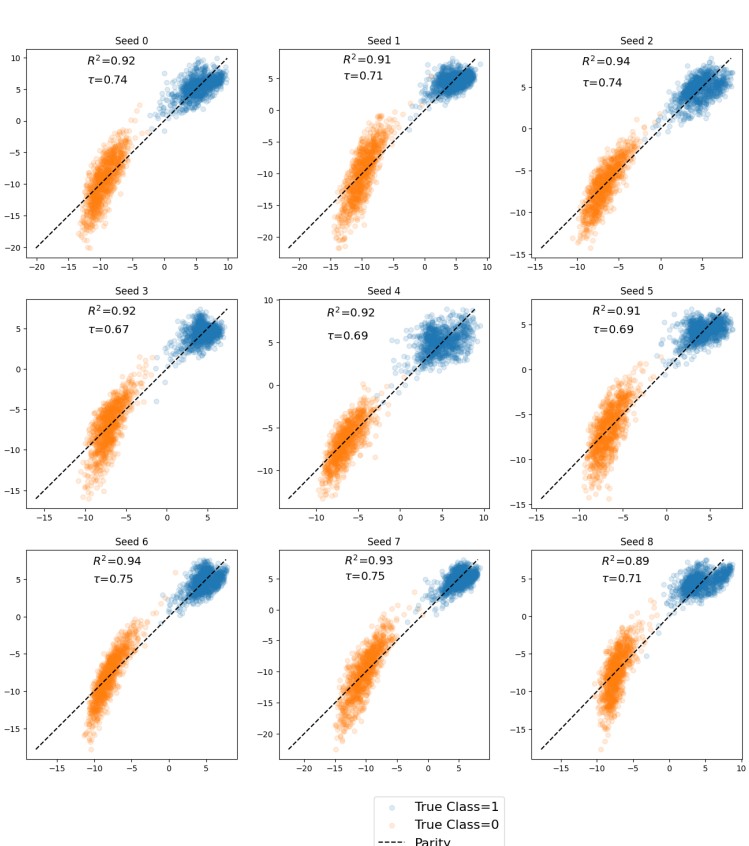

Figure 8: **MNIST2 MLP Linearization**

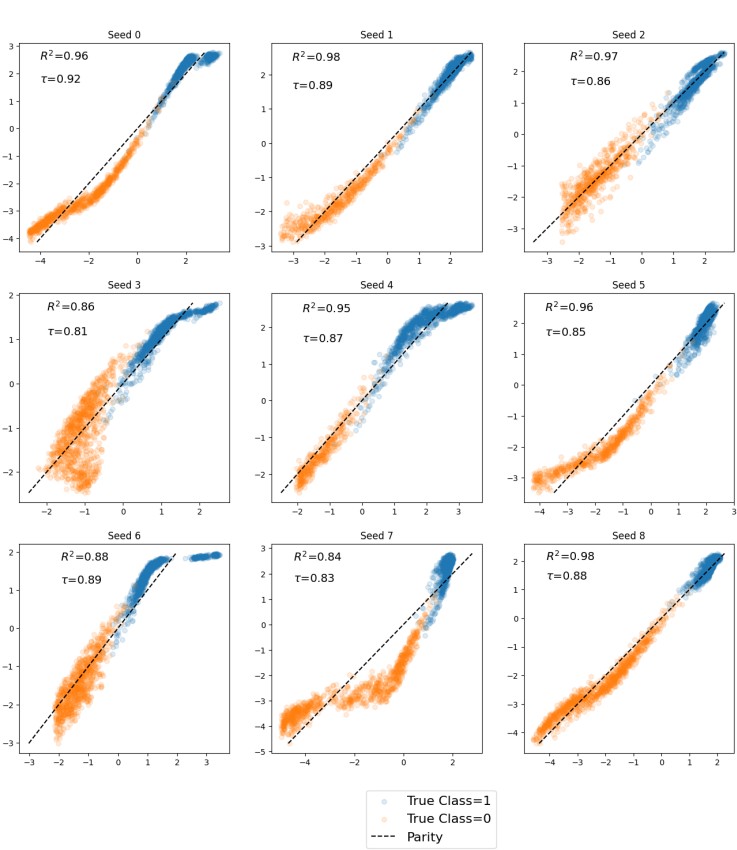

Figure 9: **MNIST2 CNN Linearization**

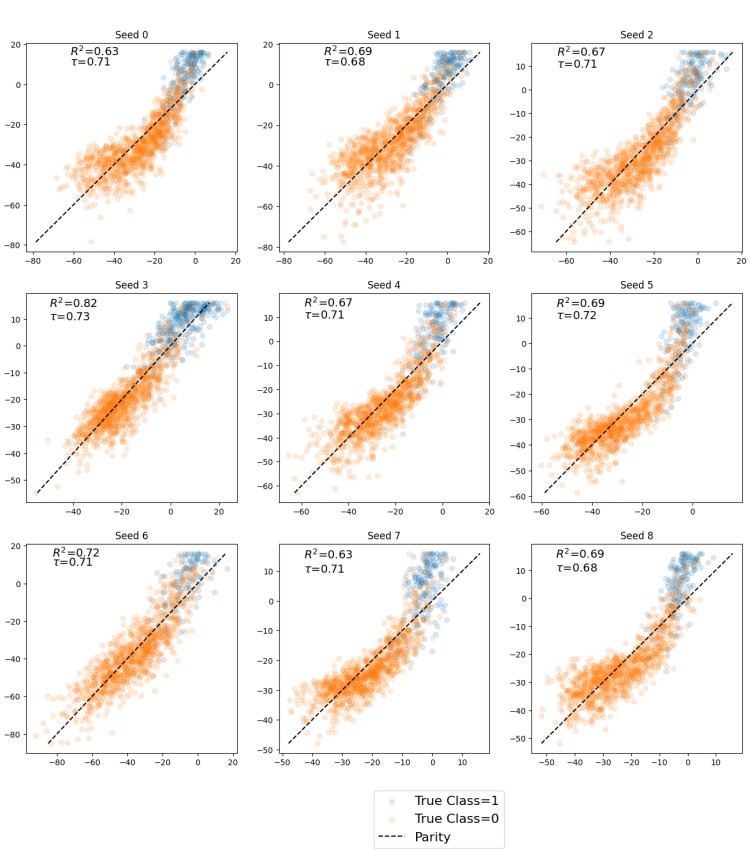

Figure 10: **CIFAR2 CNN Linearization**

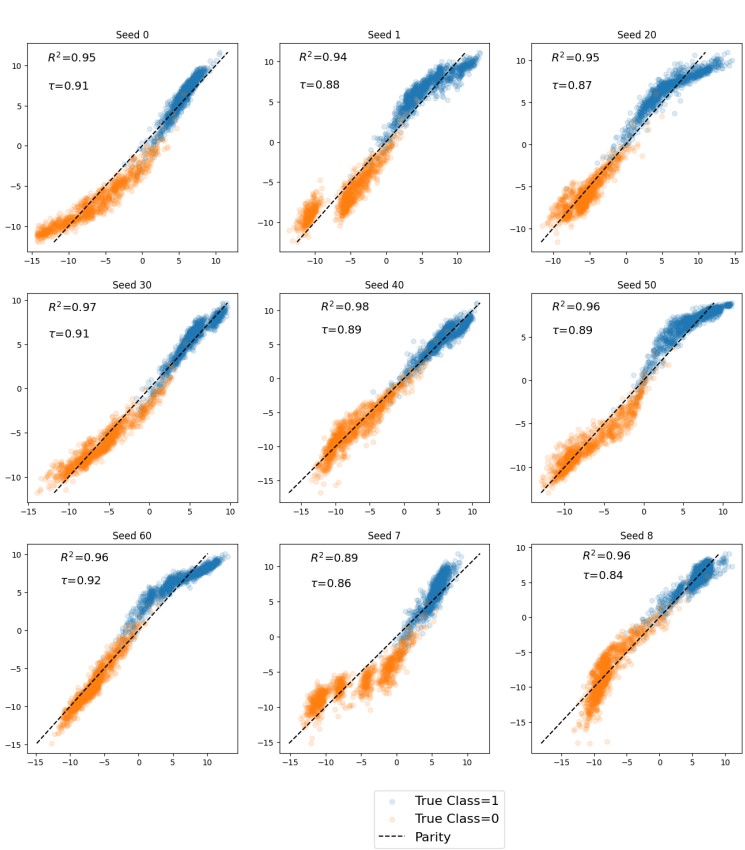

Figure 11: **FMNIST2 CNN Linearization**

# M  ATTRIBUTION VISUALIZATIONS EXPLAINED

In this Appendix, we describe the methodology used to visualize the attribution in greater detail. Our kGLM architecture gives each kernel value a unique weight for each output neuron in the NN. For example, in our visualized CIFAR10 ResNet18 network, there are 10 learned weights for each kernel value. For each column, we plot a line representing the average attribution given by training examples in that class. By design, multiplying the average attribution from each class by the number of points in each class (in CIFAR10 this is a uniform 5,000 for each class) and summing will result in the logit value of the $kGLM$ in that class. We can therefore use these visualizations to quickly compare this:

$$N \times \left( \frac{1}{N} \sum_{i=1}^{N} A(\boldsymbol{x}, \boldsymbol{x}_i) \right) = \text{kGLM}(\boldsymbol{x}). \tag{22}$$

When visualizing, we choose to hide the attribution from each training datapoint to the activation of the class c if the training datapoint's true label is not c, by slightly modifying the attribution. Let $N_c$ be the number of datapoints in class c. Let $S_c$ be the set of training datapoint indices with true label $\boldsymbol{z} = c$. Let $S_{c'}$ be the set of training datapoint indices with true label $\boldsymbol{z} \neq c$. Finally, assume the classes are balanced, as is the case for CIFAR10. Therefore, the length of the set $S_{c'} = N - N_c$. Then $A_{\text{viz}}$ gives the attribution we visualize for $i \in S_c$:

$$A_{\text{viz}}(\boldsymbol{x}, \boldsymbol{x}_i) = \sum_{i \in S_c}^{N_c} W_{c,i}\, \boldsymbol{\kappa}(\boldsymbol{x}, \boldsymbol{x}_i) + \frac{B_c}{N_c} + \frac{1}{N_c} \sum_{j \in S_{c'}}^{N - N_c} W_{c,i}\, \boldsymbol{\kappa}(\boldsymbol{x}, \boldsymbol{x}_j). \tag{23}$$

In other words, we have evenly distributed the attribution from training datapoints not in class c to the training datapoints in class c. Future work can investigate the human-AI interaction from different methods of visualization to determine the most informative visualization technique.

## M.1  ADDITIONAL ATTRIBUTION VISUALIZATIONS

In the following subsection, we visualize additional examples of attribution from the ResNet18 CIFAR10. In the first subsection, we visualize the mean value of attribution for each logit. In the second subsection, we focus on the correct logit and visualize the distribution of attribution explaining that logit's value. In the final subsection, we visualize the highest similar images from each kernel function.

### M.1.1  MEAN VALUE OF ATTRIBUTION IN EACH LOGIT

In the following plots, we visualize the mean attribution value (y-axis) from each class (different colors) to each logit (x-axis) evaluated on the test datapoint shown. We compare these values across each of the kernel functions. Because the number of datapoints in each class are an equal 5000, one interpretation of these plots are that each mean value times 5000 summed over each contributing class is equivalent to the logit value in that column. Overall, we see that typically the training data representing the same class as the logit have the highest attribution, as expected. Because attribution can be negative, a high similarity with a class can also remove total attribution in a logit. We notice that in some fraction of misclassifications, a seemingly random choice of prediction is the result of high and off-setting similarity to two classes, that leave a third class with initially low attribution as having the highest mass, and therefore logit value.

### M.1.2  VISUALIZING PREDICTED CLASS ATTRIBUTION MASS

Each figure shows the attribution distribution from each training data class for the predicted logit. Each sub-panel shows a different kernel function with the logit visualized labeled in the title. Each sub-panel is a boxplot with a dark line representing the mean contribution of attribution mass from that class. For our most consistent performing $\text{trNTK}$ kernel function, the mean contribution is within the inner quartile range for every test image.

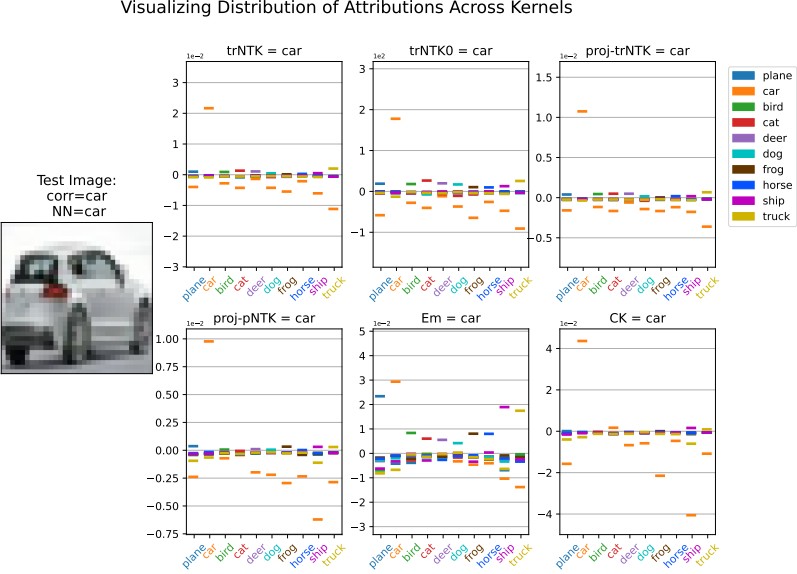

Figure 12: An image of a horse with a human handler (right side) standing in front of a trailer. The NN correctly classifies the image as a horse with a close runner-up secondary classification as a truck, which we might consider excusable given the presence of both a horse and the trailer in the image. The $\mathrm{trNTK}$ classifies as truck, with high activations for cat, dog, horse and truck. While cat is the second highest activation, the dog attribution in the cat logit subtracts from the total logit value.

Figure 13: An image of a silver car is correctly classified as a car. This is a perfect example of high confidence classification. In each logit (i.e., column), the orange tabs represent total attribution across the entire car class. In the car column, this attribution adds to the logit; in the remaining columns a high attribution to car yields a negative contribution to the logits (i.e., we trained on mutually exclusive classes, so the strong presence of one class should remove confidence in another class).

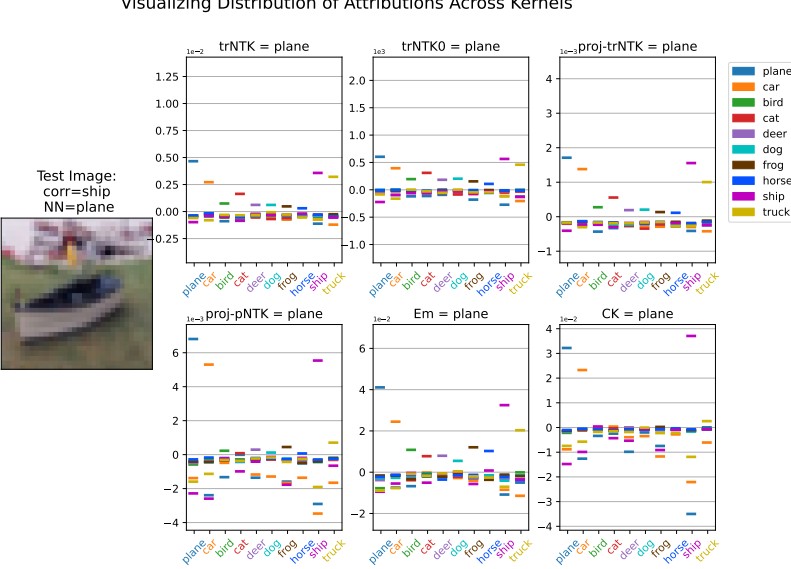

Figure 14: A bird against a blue sky is correctly classified as a bird. This is another example of strong correct classification, but unlike the previous example, the contributions of the remaining logits are somewhat elevated. The negative contribution of bird in these classes ensures the logit remains small compared to the bird logit.

Figure 15: A small boat resting on grass is incorrectly classified as a plane by the NN. We show that many kernels also follow the network misclassification, which is an important property for a surrogate model. We see a strong positive attribution to plane that is un-mediated by any of the other classes.

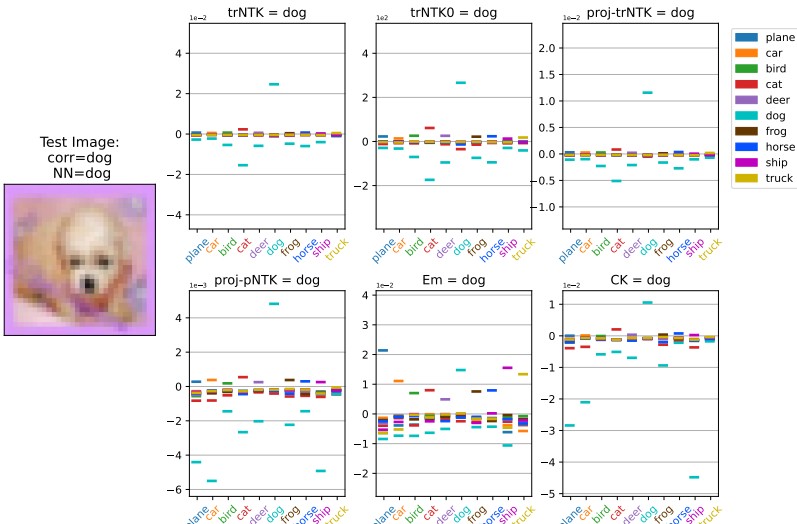

captionA bird resting on a wire is misclassified as a plane by the NN. We again demonstrate that the kernels misclassify as the same incorrect class. We again see the reason why is a strong positive attribution to plane.

Figure 16: A dog in a pink background frame is classified correctly as a dog. Similar to the other high confidence classifications, this image shows how positive attribution in one logit acts to subtract confidence in another image. It particularly highlights how high similarity to dog subtracts greatly from cat. This is an important idea to explain some misclassifications we explore below.

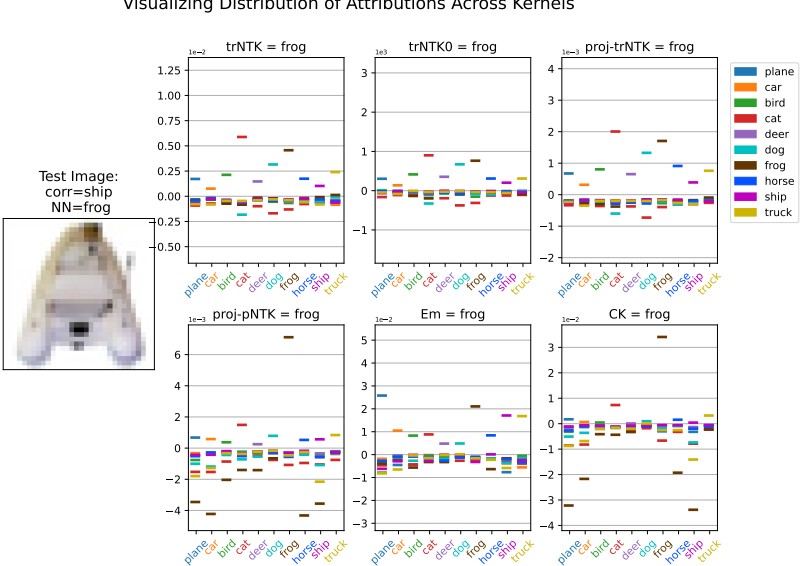

Figure 17: A large bird, possibly an ostrich, is misclassified by the NN as a deer. The kernels all have the same misclassification, with a high confidence in deer, bird, and cat.

Figure 18: An inflatable boat is misclassified as a frog by the NN. This is an interesting example, and we focus in on the trNTK. The cat attribution is actually the highest, but unlike previous examples, the attribution in the cat logit from the remaining classes subtracts enough away from the logit such that the highest remaining class is frog.

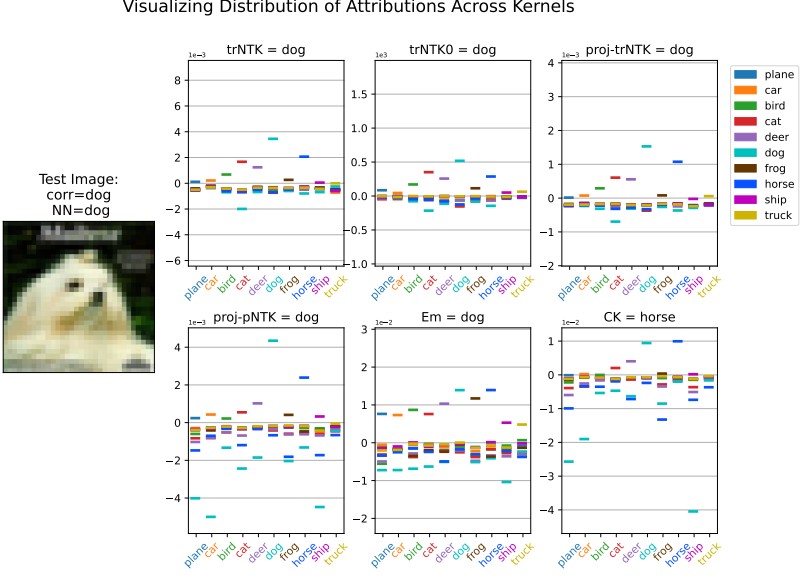

Figure 19: A car elevated on a platform against a white sky is misclassified as bird by the NN. The plane and bird class are both highly activated across each kernel function.

Figure 20: A dog with blurry text overhead is correctly classified as a dog. Each kernel function, except the CK, follows the correct classification, and it can be explained by the high attribution to the dog training data.

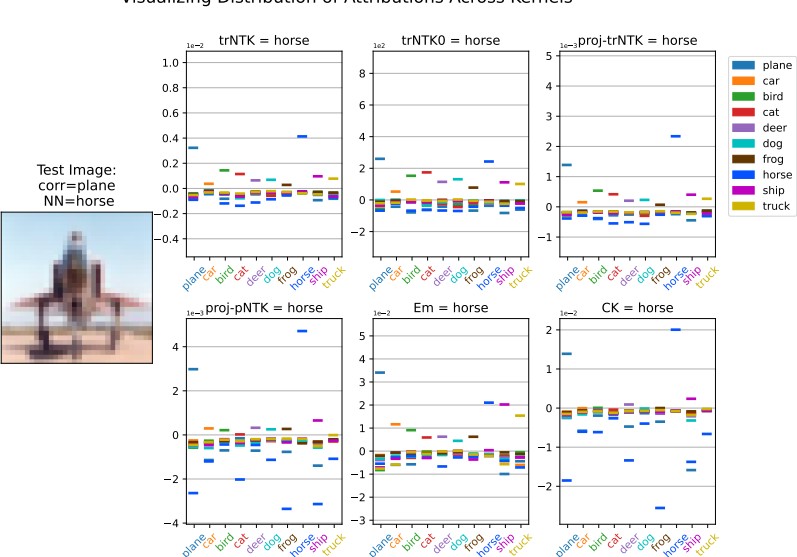

Figure 21: A person sitting on the nose of a large plane faces the camera and is misclassified as a horse. There is a high positive attribution to both plane and horse.

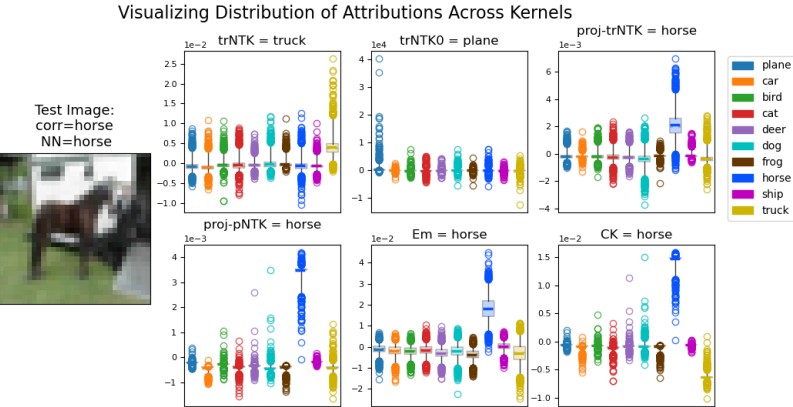

Figure 22: We visualize the entire distribution of attribution through each kernel's predicted class, (shown in sub-title). Focusing on the trNTK, the highest attributed images are truck. Compared to the previous section's plots, we now see structure of individual points from the other classes adding constructively to the Truck class logit, we some examples from each class. The mean value of attribution from each class is visualized by the colored bar.

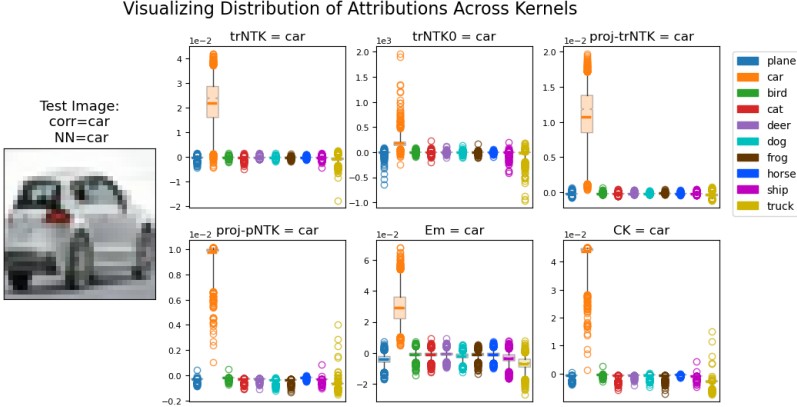

Figure 23: We visualize the entire distribution of attribution through each kernel's predicted class, (shown in sub-title). Each kernel's predicted class is car. Focusing on the trNTK: we see the distribution of cars represented as a box-plot is quite high, establishing that many car examples contribute to classify this image correctly, rather than a sparse few.

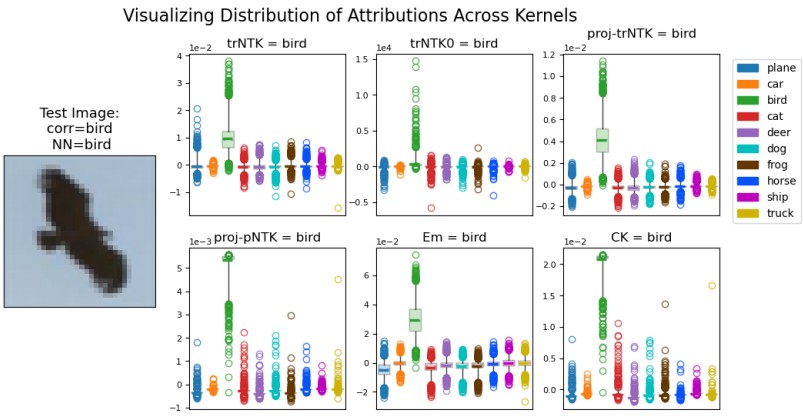

Figure 24: We visualize the entire distribution of attribution through each kernel's predicted class, (shown in sub-title). Focusing on the trNTK emphasizes how both that the distribution of Bird is high compared to the similarity of other classes, but that there are also some plane examples with high positive attribution. We might expect planes that are on blue-sky backgrounds to positively share features with birds. We delve into this example deeper in the next section.

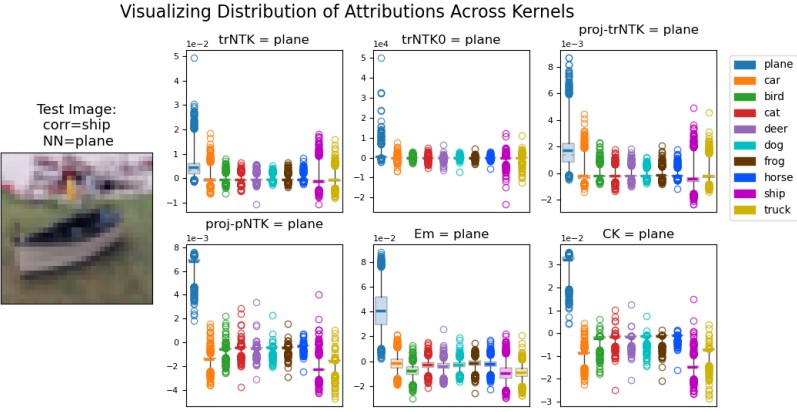

Figure 25: We visualize the entire distribution of attribution through each kernel's predicted class, (shown in sub-title). Focusing onto the trNTK, we see both car, ship, and truck have examples with high attribution supporting plane.

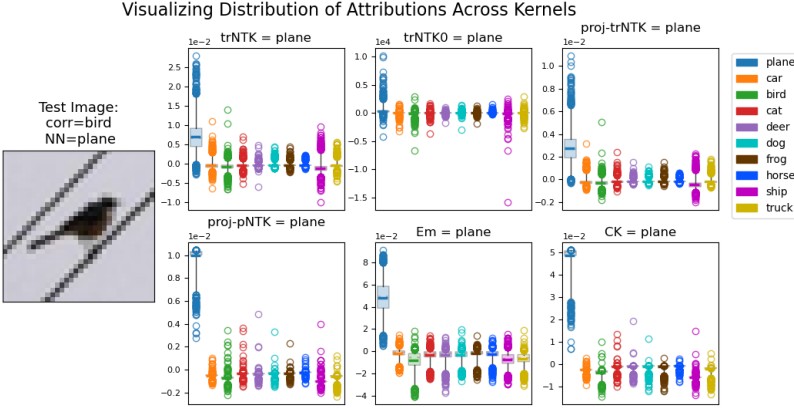

Figure 26: We visualize the entire distribution of attribution through each kernel's predicted class, (shown in sub-title). Focusing onto the trNTK, we see that there are additional bird and car examples positively attributing to the plane logit. We explore this misclassification in more detail in the section below.

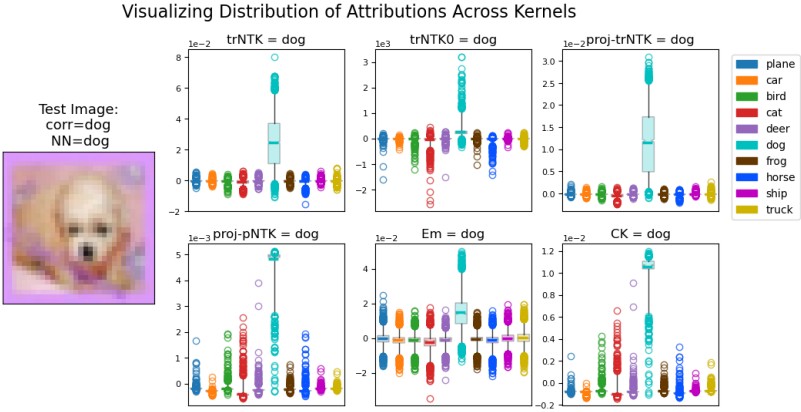

Figure 27: We visualize the entire distribution of attribution through each kernel's predicted class, (shown in sub-title). Focusing onto the trNTK, many dog examples have high attribution resulting in a clear and correct classification of dog.

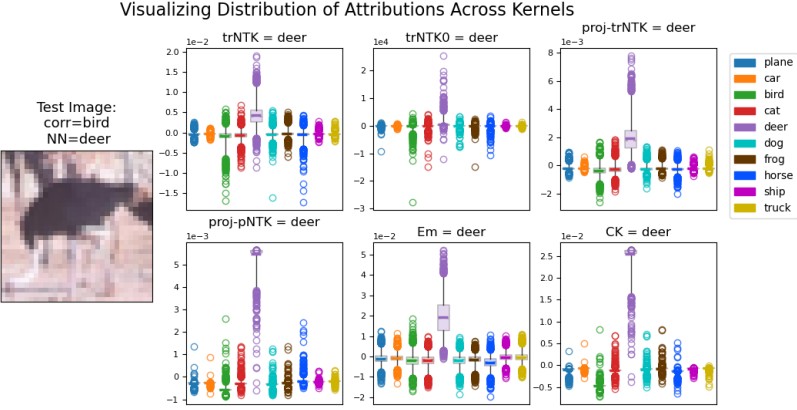

Figure 28: We visualize the entire distribution of attribution through each kernel's predicted class, (shown in sub-title). Focusing onto the trNTK, there is a higher variance to the distributions of bird, dog, deer, and horse compared to plane, car, ship and truck. Despite these variances, the distributions of the living classes are still centred on zero, so that the net contribution from the other classes is slightly negative.

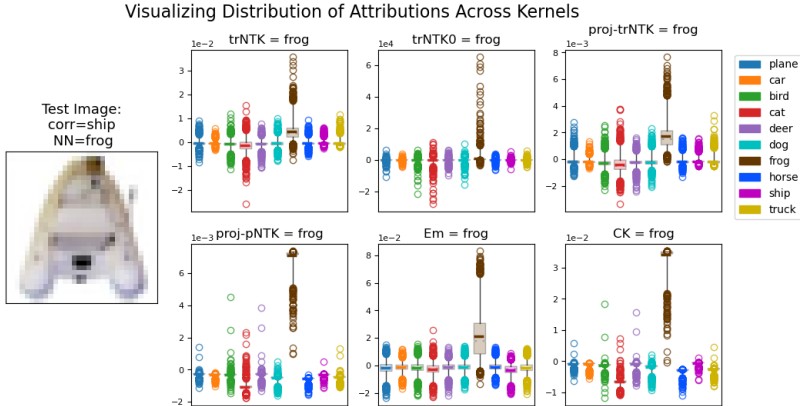

Figure 29: We visualize the entire distribution of attribution through each kernel's predicted class, (shown in sub-title). Focusing onto the trNTK, we see a higher variance of the dog and frog classes compared the the remaining classes.

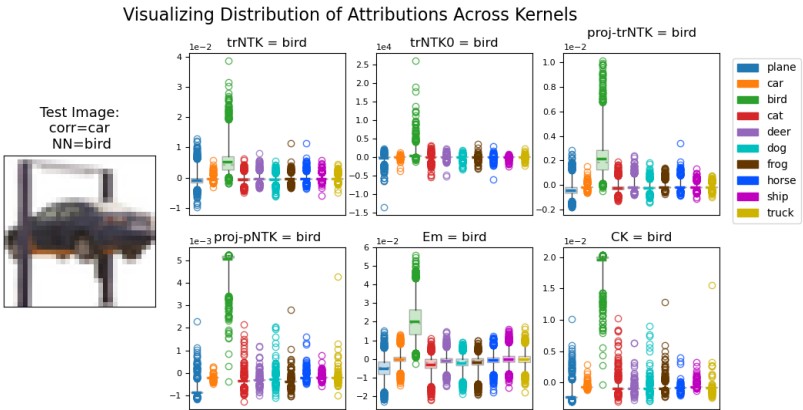

Figure 30: We visualize the entire distribution of attribution through each kernel's predicted class, (shown in sub-title). Focusing onto the trNTK, there are singular examples from the frog and horse class that stand as outliers of positive contribution while the average contribution from these classes is slightly negative (colored bar).

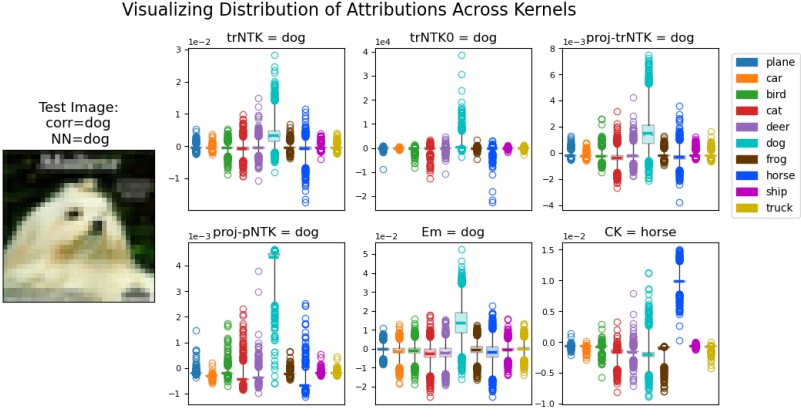

Figure 31: We visualize the entire distribution of attribution through each kernel's predicted class, (shown in sub-title). Focusing onto the trNTK, the attribution for dog, horse and deer have a higher variance, though only dog has a positive attribution.

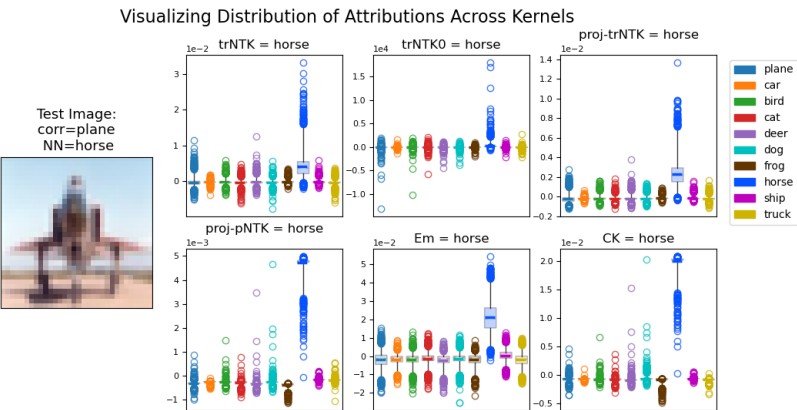

Figure 32: We visualize the entire distribution of attribution through each kernel's predicted class, (shown in sub-title). Focusing onto the trNTK, both plane and deer have example which contribute some positive attribution, but overall the effect of these classes are slightly negative to the classificaiton of horse.

## M.2 Top Five Exemplar Attribution Visualizations

In the following plots, we visualize the qualitative differences between kernels by plotting the top five most similar training images for the same selection of images as in the last Appendix. We emphasize that here, we are using the kernel function as a measure of similarity. Qualitatively, we observe that test data often share conceptual similarities with the most similar training data as evaluated by the $\mathrm{trNTK}$, and that what is chosen as most similar often reveal something about the kernel itself. For example, the CK kernel is created from the final representation of the neural network. For NN trained until convergence this final representation should have all inner-class variance collapsed (Papyan et al., 2020). Therefore, we expect the CK to mostly show that the test image is highly similar to ALL training images of the predicted class. Because the top most similar are not tied directly to our kernel surrogate model any explanations we generate from these visualizations are admittedly up to interpretation. Future work could endeavor to evaluate different kernel surrogate models such as a K-Nearest Neighbors, which would tie these visualizations directly to the surrogate model's prediction. This would be a way to recover explain-by-example with sparse number of exemplars. We also can visually confirm that the most of the highest similar images are shared between the $\mathrm{trNTK}$ and $\mathrm{proj\text{-}trNTK}$, as expected. We notice that many $\mathrm{proj\text{-}pNTK}$ examples seem shared with the CK, which we did not expect. In fact, much of the evidence presented throughout this work suggests that the $\mathrm{proj\text{-}pNTK}$ and CK share similar properties.

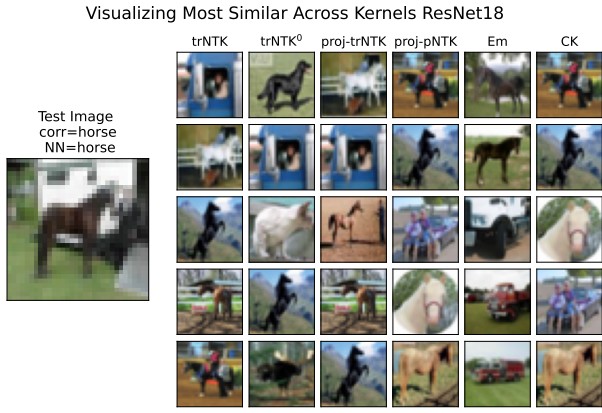

Figure 33: A horse stands next to a human and in front of a trailer or truck is correctly classified as horse by the NN model. Many of the attributed animals are shown in profile, as the subject horse of the original image stands.

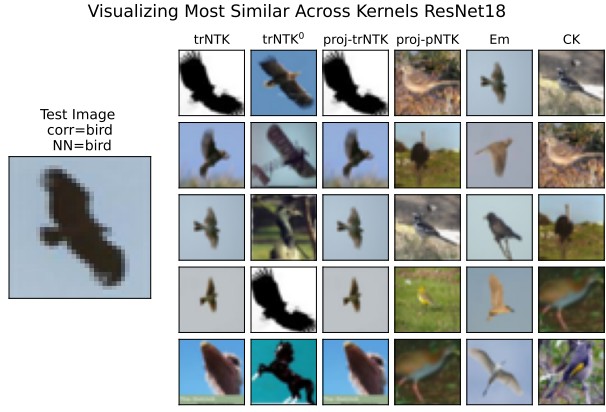

Figure 34: A silver car is correctly classified by the NN. Many similar images (seemingly the same image with different crops) exist in the training dataset.

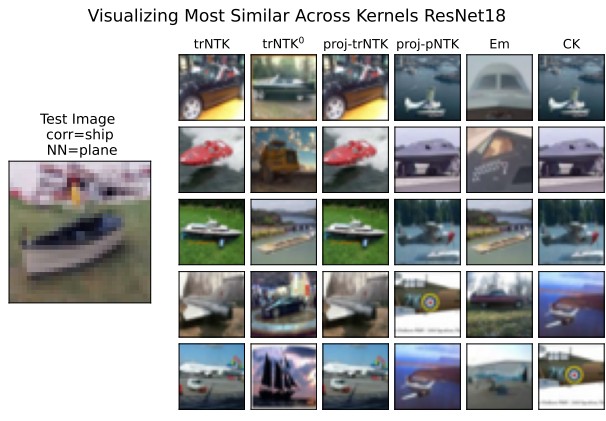

Figure 35: A bird flies with wings spread in a blue sky background and is correctly classified by the NN. Many of the bids attributed to by the evaluated kernels are also flying in a similar manner in a blue sky background.

Visualizing Most Similar Across Kernels ResNet18

Figure 36: A boat resting on grass is misclassified as a plane by the NN. The most similar attributions are varied, perhaps demonstrating a weakness in this kind of visualization.

Visualizing Most Similar Across Kernels ResNet18

Figure 37: A bird resting on a wire that spans the image diagonally is misclassified as a plane. Many of the highest attributed images from the $\mathrm{trNTK}$ and $\mathrm{trNTK}^0$ have a similar diagonal quality, even if the underlying class of the subject of the image is much different than the true or classified class.

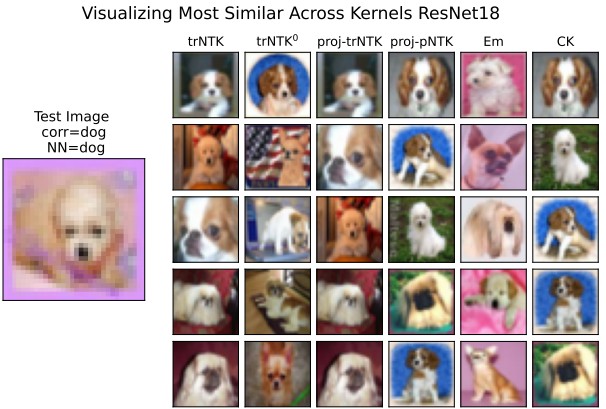

Figure 38: A small puppy in a pink background looking out of the screen ("at the camera") is correctly classified as a dog. Many of the most similar images are dogs that look out of the screen. The Embedding kernel seems very focused on the background pixel values, as many of the attributions are pink centered.

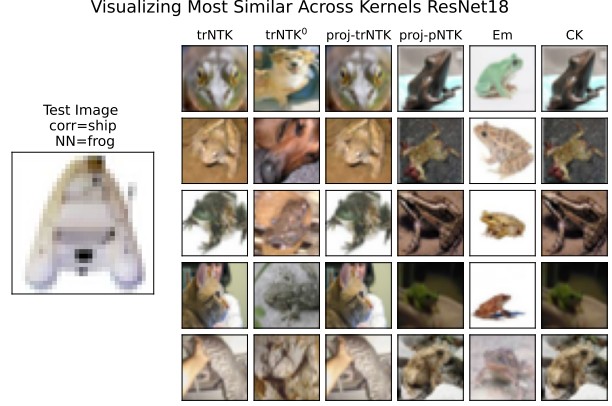

Figure 39: A large bird is misclassified as a deer. The attributed images are varied, perhaps demonstrating a weakness in this kind of visualization.

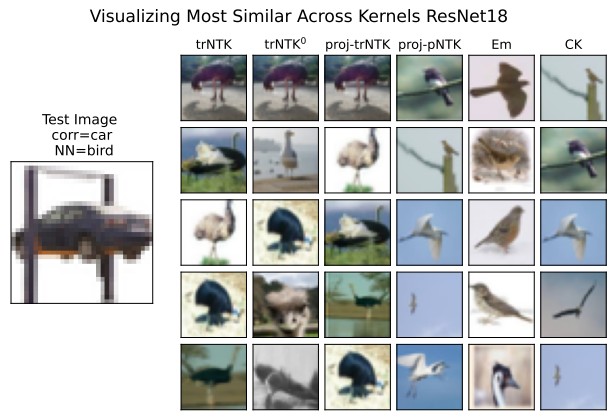

Figure 40: A white inflatable boat is misclassified as a frog. The attributed images are varied, perhaps demonstrating a weakness in this kind of visualization.

Figure 41: A car resting on a raised platform is misclassified as a bird. Many of the bird attributed to by the $\mathrm{trNTK}$ and $\mathrm{trNTK}^0$ are large bird with rotund black bodies and stalky legs, perhaps suggesting a pathway for the misclassification.

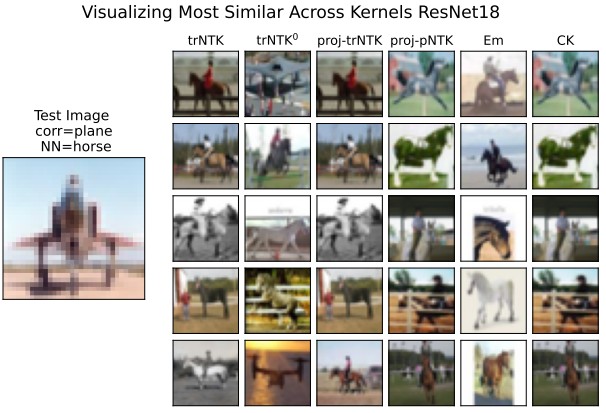

Figure 42: A dog with blurry text overhead is correctly classified as a dog. The attributed images in the $\mathrm{trNTK}$ and $\mathrm{trNTK}^0$ are mostly images of animals with white fur "looking left" mirroring the test image of the dog "looking right".

Figure 43: An image of a person sitting on the nose of a plane facing towards the camera. The NN misclassified this example as a horse. The $\mathrm{trNTK}$ shows many example of people riding horses, mirroring the person "riding" the plane.

# N    ADDITIONAL DATA POISONING ATTRIBUTION VISUALIZATIONS

In this Appendix, we provide additional visualization for the data poisoning experiment attributions. We show the same selection of images as the previous section for comparison. Because the NN classifies nearly every poisoned image as the targeted class deer, we expect that a good surrogate model would reflect this fact by attributing highly to poisoned examples. Because the model trained is a different architecture than the ResNet, it can be interesting to compare the top attributions to the previous Appendix section.

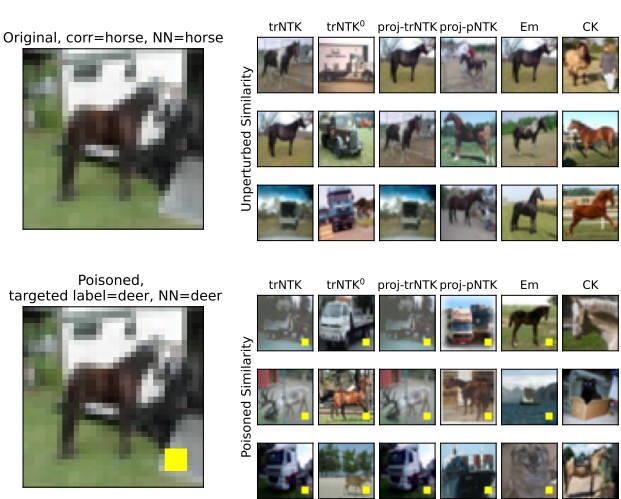

Figure 44

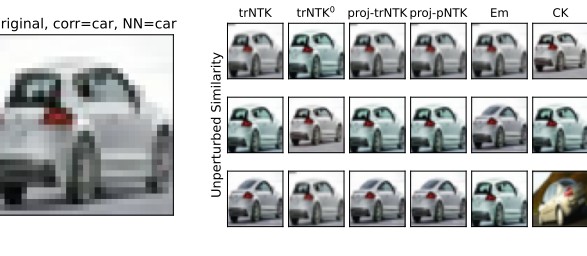

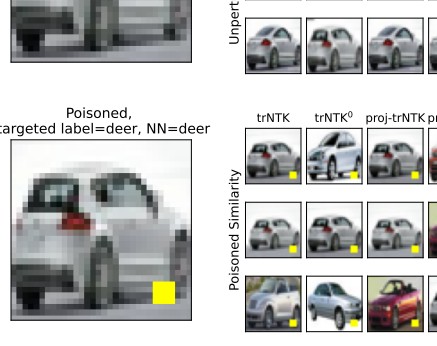

Figure 45

Visualizing Poisoned Similarity Across Kernels

Figure 46

Visualizing Poisoned Similarity Across Kernels

Figure 47

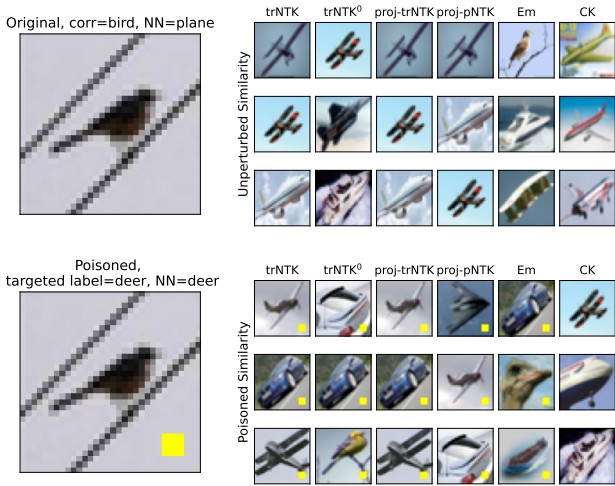

Figure 48

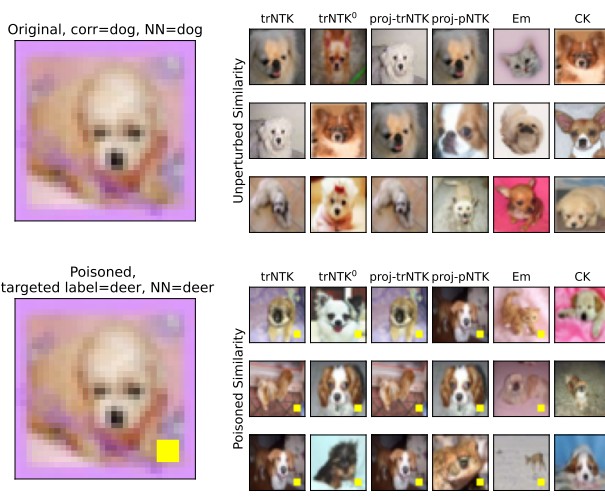

Figure 49

Visualizing Poisoned Similarity Across Kernels

Figure 50

Visualizing Poisoned Similarity Across Kernels

Figure 51

Visualizing Poisoned Similarity Across Kernels

Figure 52

Visualizing Poisoned Similarity Across Kernels

Figure 53

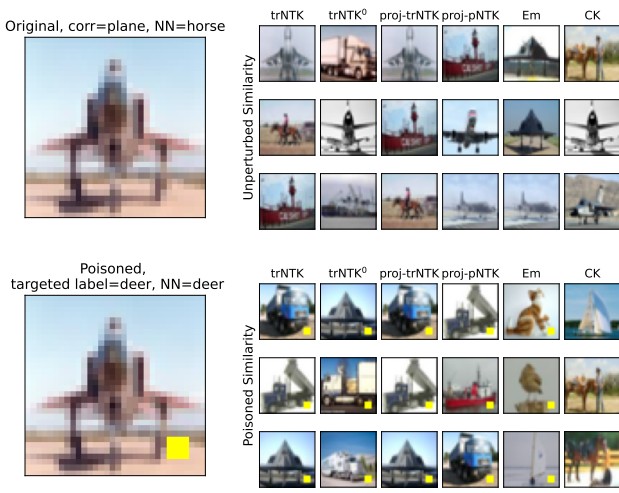

Figure 54

