# OpenReview forum: "Faithful and Efficient Explanations for Neural Networks via Neural Tangent Kernel Surrogate Models"
_ICLR.cc/2024/Conference — ICLR 2024 spotlight_

### Official Review · Reviewer_LB8S · 2023-10-16

**Soundness:** 4 excellent
**Presentation:** 4 excellent
**Contribution:** 3 good
**Rating:** 8
**Confidence:** 4

**Summary:**

This is an empirical paper on Neural Tangent Kernel (NTK) surrogate models. The content of the paper is two-folded:

1. Several approximations of NTK are introduced and evaluated quantitatively by various metrics through different experiments, showing how theses approximations capture the decision mechanism of Neural Networks (NN) on classification problem.
2. Then the paper argues how the NTK surrogate models give explanation for NN decision and states its limitation on SVM and adversarial attacks.

The paper includes a detailed and motivated introduction to Trace NTK and pseudo NTK, and experimental results on various data sets and  NN models. Its appendix contains a detailed section explaining the relationships between different kernels introduced in the paper, and a detailed result of the experiments together with visualisations.

**Strengths:**

Originality: The paper is innovative to use the Kendall-$\tau$ rank correlation to evaluate the approximation, such as TrNTK, pNTK, CK...,  on the empirical NTK (eNTK). The angle to experiment on explaining NN by surrogate NTK is also novel.

Quality: The paper is written nicely with rigorous definitions and detailed descriptions on the experiments.

Clarity: The paper clearly states the problem and presents their experiments. Also the motivation of the paper is clearly elaborated.

Significance: The paper is important in the area of explainable AI through the lens of surrogate NTK. This paper could lead to more research on related topics.

**Weaknesses:**

There is barely any flaws in the paper, and the limitation of the experiments is clearly stated in the limitations subsection in section 5.

**Questions:**

I have only one question:
In section 5, You mentioned: "...an interesting follow-on work would investigate using kernel functions in K-Nearest Neighbors surrogate models." How much argument of this paper can transfer to KNN or generally any other surrogate models on explaining NN decision?

Also, there are some of the minor typos in the paper:

Section 2 PRELIMINARIES Neural Networks for Classification third line: it should be \mathcal{Y} instead of Y.

Appendix F FORMAL DEFINITION OF EVALUATION METRICS last equation: it should be SS_res instead of SS_ret.

---

> ### Author Response · Authors · 2023-11-18
>
> 1. We thank the reviewer for their time, and echo their excitement about the results. We believe the core strengths of the paper are: a) identifying a metric Kendall-$\tau$ that is a better measure of faithfulness over the previously used test accuracy in comparison of similarity between models, and b) presentation of the evidence we collect that challenge the assumption of sparsity in data attribution/explain by example explainability work. To build upon its significance and relate to the question posed below: The choice of the structure of the surrogate model can be an implicit assumption on the kind of data attribution that NN model performs. We chose to use a kGLM because it does not assume sparsity-- we could have in principle observed that the resulting kGLM was effectively sparse, but do not. In contrast, a choice like a KNN would be assuming that the data attribution is local in the feature space and sparse. We continue this discussion below.
>
> 2. Re:  In section 5, You mentioned: "...an interesting follow-on work would investigate using kernel functions in K-Nearest Neighbors surrogate models." How much argument of this paper can transfer to KNN or generally any other surrogate models on explaining NN decision?
>     * An excellent question that we invite the reviewer to discuss with us. We have conducted some preliminary work investigating the roles other surrogate models (kernel-SVM and kernel-KNN) can play in investigating the NN model. So far we understand that the choice of surrogate model can have profound impact on what we might infer. For example, with the kernel-KNN we would specifically be modeling that only the local structure of the feature space matters to the classification; in effect, assuming sparsity a priori. However, it is not yet clear how we should assign a class probability/logit value to utilize the Kendall-$\tau$ measure. We do include some analysis with kernel-SVMs as well looking into adversarial examples. We infer that kSVMs are not a good surrogate for modeling the effect of adversarial examples since the robustness scales differently between the NN and the kSVM. This may be in part due to the structure of the kSVM itself: in the kSVM, only a sparse number of data points near the decision boundary matter for classification (i.e., the support vectors). Perhaps it is easier for an adversarial example to fool the kSVM given fewer samples the kSVM is utilizing than the NN (under our hypothesis that the NN is not sparse).
> 3. Re: also, there are some of the minor typos in the paper
>     * We thank the reviewer for pointing out the mistakes and will fix them in the revision

---

### Official Review · Reviewer_dXvz · 2023-10-28

**Soundness:** 3 good
**Presentation:** 2 fair
**Contribution:** 2 fair
**Rating:** 8
**Confidence:** 2

**Summary:**

This paper contributes to the growing literature of approximating NNs with simpler, more interpretable, models. A common approach is to approximate NNs with the empirical Neural Tangent Kernel (eNTK). However, computing the eNTK can be computationally unfeasible so simpler approximations have been proposed in the literature. The authors focus on this problem by studying the empirical properties of one such approximation, the trace NTK, and adapt random projection methods to make it more computationally attractive. Furthermore, the authors propose the Kendall rank correlation as a new measure to assess the faithfulness of the surrogate kernel method to the NN.

The main contribution of the paper is to show that the trace NTK and projected trace NTK can be used to generate faithful surrogate models of the underlying NNs. The authors show this through a variety of empirical exercises across different benchmark datasets (MNIST, CIFAR) and models (CNNs, ResNet18, BERT). They compare how good different NTK approximations are with respect to the underlying NN in terms of prediction error and rank correlation and find that trace NTK has good performance. Additionally, the authors compare the different models representations through a data attribution exercise and a poisoned data attribution exercise and identify which surrogate models perform better in each case.  Finally, the authors show the practicality of the projected NTK methods by analyzing computational complexity of each method. The most relevant finding is that the trace NTK and projected trace NTK  perform similarly in settings in which the projected trace NTK required an order of magnitude less computation time.

**Strengths:**

The paper tackles a common problem in an active area of research: that of finding computationally attractive NTK approximations to NNs. While the main proposed methods in the paper are drawn from the literature, the authors are original in adapting random projection methods to the trace NTK and pseudo NTK and considering the rank correlation as a sensible alternative to prediction error for assessing faithfulness.

The main strength of the paper lies in the extensive set of empirical evidence comparing various NTK approximations with different NNs architectures across tasks and datasets. Furthermore, the authors provide an extensive appendix with additional exercises and comparisons. Overall, researchers looking to use NTK approximations may find these exercises useful in choosing which method to use depending on their task and their computational constraints.

**Weaknesses:**

While the paper offers a wealth of empirical evidence for the methods it investigates, its central weakness is that it is unclear what the main findings and contributions are. The paper does a lot of things and it would benefit from more succinctly explaining what it is trying to achieve and how each exercise demonstrates it.

* The paper should be more clear about its relative contribution to the literature (and what its main contribution is). The paper gives confusing statements about what is new and what is taken from the literature. The authors state that the 3 main contributions are (1) new kernel functions for NTK approximation, (2) first to show that eNTK models are consistently correlated with NNs across experiments, (3) first to compare NNs decisions through NTKs through a data attribution and explain by example strategy.


* For point (1) however the authors also state that the tr NTK was introduced in Chen et al. 2021 (end of page 2) and that the random projections approach is based on Park et al. 2023 (end of page 1).  Is the main contribution of the paper proposing a new NTK method or evaluating empirical exercises?

* The authors consider different alternative NTKs to compare to the trNTK, but never compare in the main text the methods to the eNTK or the pNTK. Given that the motivation of the paper in the abstract, introduction etc is to approximate the eNTK it is odd that this not done in the main text of the paper. While computational constraints are important, maybe it could be done a simpler dataset (MNIST)?

* For point (2) if using the rank correlation is new it should be clearly stated as a major contribution. The paper repeatedly expresses that other measures are flawed and while the authors give some reasons why, without a proper theoretical statement the authors should at least relate these notions more directly to the findings of the empirical examples. For example, what is a clear case in which using fit or pearson correlation would be misleading in the sense that two NTK models give you very different attributions despite having the same fit to the NN, but rank correlation is not misleading.

* For point (3) the paper should explain more carefully why these are carried out. If the goal is to assess how good a NTK approximation is by considering whether it performs similarly to the NN in a data attribution task then this should be the focus of the results. It seems that the authors do these exercises in the appendix, but in the main write up they just give an instance of this and its unclear how much we can learn from it. If well addressed I would be inclined to raise my score.

* The paper could benefit from better exposition and more clear presentation. For example, the choice of what is defined in the main text vs appendix and when it is defined is sometimes odd. The eNTK, while being referenced to extensively, is never properly defined in the main text. The, trNTK0 is introduced in Additional Kernel Functions after the trNTK without motivation, despite featuring prominently in the appendix when the different methods are compared.

* The paper may also suffers from typos and plots are sometimes misleading (squished axis in Figure 1). Some typos include pseudo vs psuedo, missing points, figure labels that overlap, subscripts in mathematical notation etc. Some references are also repeated.

**Questions:**

Besides the questions raised in the weakness section regarding the key contributions. I also have some additional questions:

* Which Chen et al. paper is the main reference for trace NTK, I was confused by the reference.
* Is it true that when rank correlation <1 there exists not invertible mapping?
* In the case in which the rank correlation is 1 is the invertible mapping unique? How does this result translate to the exercises and neural net behavior? Should we expect the same data attributions as the NN? Expanding on the implications of a good rank correlation vs test accuracy seems key to show the usefulness of the paper for researchers.
* Given the data attribution with kernels theory in page 3, wouldn’t you be able to test directly whether a kGLM is an “ideal surrogate” (according to eq 2) by comparing across all data points the NN confidence in each class with the data attribution for each class? Is there a way beyond fit/correlation measures to more systematically compare how well the kernel performs in the data attribution exercise besides evaluating individual examples?

---

> ### Author Response · Authors · 2023-11-19
> **Clarification**
>
> Good afternoon, we are still preparing our response, but today we realized we were not sure of the meaning of the following question. Could the reviewer please clarify? "Given the data attribution with kernels theory in page 3, wouldn’t you be able to test directly whether a kGLM is an “ideal surrogate” (according to eq 2) by comparing across all data points the NN confidence in each class with the data attribution for each class?"

---

> > ### Comment · Reviewer_dXvz · 2023-11-20
> >
> > Hi, I think my question is the same as question 2 of reviewer yHxi. Given your definition of ideal surrogate, why not compare the probabilities of all classes at each data point, instead of just the probability of the true label?

---

> > > ### Author Response · Authors · 2023-11-21
> > >
> > > We thank the reviewer for the clarification. We will answer this question in this thread.
> > >
> > > It is true that the ideal surrogate should replicate the probabilities from every output instead of just the output of the true classes. This information is not leveraged in our Kendall-$\tau$ metric, and we plan to add a sentence exploring this in our limitations. Furthermore, we have resolved to add an appendix about experiments on a "misclassification coincidence rate." This coincidence rate measures the ratio of the number of times the NN and kGLM choose the same class out of all misclassifications, over the total number of all misclassifications. This rate is actually high (about 75\% coincidence of misclassifications for trNTK on ResNet18), which is powerful evidence suggesting that the kGLM mimics the NN overall for both correct classification and misclassifications. While this admittedly still does not leverage the full probability space, its an independent piece of evidence that considers additional logits to Kendall-$\tau$ backing up our main conclusions.

---

> ### Author Response · Authors · 2023-11-21
>
> 1. We thank the reviewer for their detailed comments. Before we clarify some of the specific issues raised, we want to echo one of the main strengths the reviewer identifies in the paper. Part of our contributions are the implementation of projection variants of approximate eNTK. We show in Table 4 how these projection variants achieve a massive reduction in total computation time with exponentially decaying residuals to the non-projected variant (Appendix E). We are excited to see how researchers will build off this work to conduct eNTK research enabled by our software and TRAK.
> 2. Re: Contributions are unclear. For point (1) however the authors also state that the trNTK was introduced in Chen et al. 2021 (end of page 2)...
>     * The full story is somewhat complex and in editing to meet page limits it became less clear. Chen et al. (2021) never formally defined the trNTK and never mentions it in their main body. They only describe trNTK in a single sentence in the appendix: (Appendix G, first sentence: "We approximate the neural tangent kernel on the i1k (1/10) subset by averaging over block diagonal entries ... in the full NTK."). Therefore, we feel our wording is accurate albeit slightly confusing: we are the first to formally define the trNTK. In Chen's work, the trNTK was a (hidden) computational convenience to enable their experiment for a single data set; in contrast, our work treats the trNTK to be the central object of study as an approximation of the NTK. We believe that this is a novel contribution but at the same time we should give proper credit to Chen et al. We will look to revise the contribution section and introduction to make this point clearer.
> 3. Re: ... and that the random projections approach is based on Park et al. 2023 (end of page 1).
>     * We copy from our comment to all on this topic for the reviewer's convenience: we would like to take this opportunity to delineate our contributions from that of Park et al 2023 (TRAK). To begin, we want to acknowledge that TRAK is software at the foundation of our projection kernel functions, so it rightly takes a prominent role in our paper. The TRAK paper references the eNTK and devotes a section to show how it is an explicit part of the TracIn kernel, but its actual calculation is not implemented in their code. You can view the lines of code [here](https://github.com/MadryLab/trak/blob/main/trak/modelout_functions.py#L84)
>     * Notice that the model output functions implemented are only the loss (as this serves their use case for counterfactual analysis), rather than the neural network function itself. TRAK does not compute the eNTK or any approximation of the eNTK (p-, tr-, etc.). We have been in direct communication with the maintainer of TRAK to bring the capabilities we describe in this work to the TRAK software package.
> 4. Is the main contribution of the paper proposing a new NTK method or evaluating empirical exercises?
>     * We view the paper as having three major contributions: a) We describe methods, building off recent successes from TRAK, that use projection matrices to compute efficient approximations to the eNTK. b) We establish the faithfulness metric Kendall-$\tau$ and detail how it is superior to previous methods relying on test accuracy. c) We identify the trNTK kernel function as the best performing surrogate model to the NN across a variety of experiments. Analyzing this choice of kernel in a data attribution context, we find that many training data points influence the output on any new data input.
> 5. Re: The authors consider different alternative NTKs to compare to the trNTK, but never compare in the main text the methods to the eNTK or the pNTK...
>     *  Our goal was to develop surrogate model methods that scale to large computer vision experiments and transformer models with potentially large number of classes. This implies we will need to find an approximation to the eNTK as the eNTK scales quadratically with the number of classes. A direct comparison to the eNTK is an important topic, but we did not consider it part of this paper's scope.
>     * We understand the reviewer's concern that we did not compare with the pNTK. Our rational is that we "brute-forced" the computation of several trNTK and compared them with the proj-trNTK. After observing that the proj-trNTK had exponentially decaying residuals (Appendix E) with the trNTK, we felt secure in evaluating the proj-pNTK alone.
>
> (continued below)

---

> > ### Author Response · Authors · 2023-11-21
> >
> > 6. For point (2) if using the rank correlation is new it should be clearly stated as a major contribution. The paper repeatedly expresses that other measures are flawed and while the authors give some reasons why, without a proper theoretical statement the authors should at least relate these notions more directly to the findings of the empirical examples. For example, what is a clear case in which using fit or Pearson correlation would be misleading in the sense that two NTK models give you very different attributions despite having the same fit to the NN, but rank correlation is not misleading.
> >     * We copy from our comment to all for the reviewer's convenience:
> >     * To begin: Pearson correlation measures the ratio of the covariance between a set $\mathcal{X}$ and $\mathcal{Y}$ to the product of the square root of the variance in each set. Kendall-$\tau$ measures the degree to which the sets are monotonic with one another, irrespective of the functional form of any mapping between sets.
> >     * Consider the following toy scenario: imagine there are two independent classification models that achieve nearly 100\% accuracy. Furthermore, the output of these models is a normally distributed random variable with $\mathcal{N}(\mu_1,\sigma_1)$ if the true label is 1 and $\mathcal{N}(\mu_0,\sigma_0)$ if the true label is 0. If $\sigma_0,\sigma_1$ are small compared to $\mu_0-\mu_1$, then the Pearson correlation will be nearly 1.
> >     * Intuitively, this is because this describes point clouds centered at $\mu_0$ and $\mu_1$, and if those point clouds have a high degree of separation relative to their variance each random variable has in each point cloud ($\sigma_0$ and $\sigma_1$) then the covariance will be nearly equal to the variance of each random variable. This means that the Pearson correlation will be 1. We want a measure of the dependence between two random variables, and we have now shown that Pearson can result in value 1 even if the variables are explicitly not dependent on one another.
> >     * In contrast, Kendall-$\tau$ returns a lower bound value of 0.5 in this toy model (assuming equal class sizes). So while Kendall-$\tau$ is still sensitive to the failure mode affecting Pearson correlation, with his probability Kendall-$\tau$ does not reach its maximum, so can still be a useful tool to distinguish between independent and dependent functions.
> >     * While simple, this thought experiment is revealing. We have two point clouds at $\mu_1$ and $\mu_0$. In Appendix I.1 (Figure 7) we can see that the point cloud structure of the logit observations mimic this case. To help explain this more concretely, we provide a Jupyter notebook file with these experiments in the supplementary work to demonstrate our arguments.
> >
> > 7. Re: For point (3) the paper should explain more carefully why these are carried out. If the goal is to assess how good a NTK approximation is by considering whether it performs similarly to the NN in a data attribution task then this should be the focus of the results. It seems that the authors do these exercises in the appendix, but in the main write up they just give an instance of this and its unclear how much we can learn from it. If well addressed I would be inclined to raise my score.
> >     * The qualitative analysis we provide between kernels is an important tool to help build intuition and formulate new hypotheses. For example, the framework and visualizations were critical in allowing us to reach the conclusion that the NN's predictions depend more so upon all the training data rather than a sparse number of exemplars, since the NN has the highest correlation with the trNTK. This faithfulness allows us to infer non-sparsity since the outliers of attribution from the trNTK do not dominate the total attribution. This serves as a counter-narrative to the prevalent explain-by-example paradigm which has assumed sparsity a priori.
> >     * As an example of how we can utilize these tools, see Figures 20, 31, and 42: In Figure 20, we see that the mean plane and horse attribution score are highest driving the classification in their respective logits, but with the mean attribution from the remaining classes having higher negative impact on the plane logit. Figure 31 zooms into the horse logit for the trNTK, showing that the logit horse value is dependent upon the horse class. Even though our analysis shows that a sparse number of representations do not best describe the NN predictions, we can still leverage the most similar data points to help us form hypothesis of NN behavior. In Figure 42, we show the most similar images are of people riding horses. Upon seeing this, we realized that we had not noticed that the initial test image is actually of a person siting "on top" of an airplane! While we still must find a way to test this hypothesis, we believe that helping researchers form these hypotheses is still a valuable tool in understanding misclassification.
> >
> > (continued below)

---

> > > ### Author Response · Authors · 2023-11-21
> > >
> > > 8. Re: The paper may also suffers from typos and plots are sometimes misleading...
> > >     * We thank the reviewer for pointing out these mistakes and will correct them in the revision.
> > > 9. Re: Which Chen et al. paper is the main reference for trace NTK, I was confused by the reference.
> > >     * The X. Chen et al. paper was repeated erroneously in our bibliography. we will keep the ICML version in the revision. A description of their approximation to the eNTK appears in their appendix.
> > > 10. Re: Is it true that when rank correlation $<1$ there is no invertible mapping?
> > >     * This is correct, only strictly monotonic functions pass the horizontal line test and have inverses. But, that does not preclude us from trying to fit an invertible function anyways (e.g., pseudo inverses): If KGLM = NN + $\epsilon$, then Kendall-$\tau$ would assumedly be less than one due to jitter, but the overall capability and utility of our surrogate model could not be denied. In effect, Kendall-$\tau$ is still informative.
> > > 11. Re: In the case in which the rank correlation is 1 is the invertible mapping unique?
> > >      *  Kendall-$\tau$ = 1 implies a monotonic relationship. A **strictly** monotonic function $f$ implies the existence of an inverse function $f^{-1}$ on the range of $f$. If a two-sided inverse exists, it is indeed unique and the proof is a simple exercise by manipulating the identity element $e$ (e.g. say A and B are both inverses of C: $A = Ae = A(CB) = (AC)B = eB = B$). The major caveat is that Kendall-$\tau$ can be 1 depending on how ties are resolved without the relationship being **strictly** monotonic.
> > > 12. Re: How does this result translate to the exercises and neural net behavior? Should we expect the same data attributions as the NN?
> > >     * This is an excellent question! Currently, we do not know a principled technique for extracting reasoning from a NN. In its place, this paper describes a method to find a surrogate model that mimics the classification behavior of the NN. This method is interpretable from a linear model theory perspective. So we assume that if the surrogate model is faithful (enough), we can use its reasoning (e.g. attribution) as a proxy for the NN's. In other words, our assumption is an expectation for "the same attribution as the NN."
> > > 13. Re: Expanding on the implications of a good rank correlation vs test accuracy seems key to show the usefulness of the paper for researchers.
> > >     * We thank the reviewer, we will be more explicit about our critique of test accuracy sooner in the paper for clarity. We discuss why test accuracy is a poor faithfulness measure compared the rank correlation in section 3 paragraph 2: equivalent test accuracies are necessary but insufficient for two functions to be the same. As discussed elsewhere, many reviewers feel that we did not properly motivate Kendall-$\tau$ as a replacement for test accuracy; we have presented our argument for its justification elsewhere in and intend to incorporate the justification into our revision.
> > > 14. Re: Is there a way beyond fit/correlation measures to more systematically compare how well the kernel performs in the data attribution exercise besides evaluating individual examples?
> > >     * We believe part of this question asks whether we need to leverage samples to test the the NN and kGLM; for example, if the kGLM was a true ideal surrogate of the NN shouldn't it have the same value at any point on the NN domain? The answer to this is yes, though, in the paper our scope is only to explain the NN's behavior on points sampled from a population $\mathcal{X_{test}}$. We have assumed we do not know the distribution underlying $\mathcal{X_{test}}$ but can sample from it. These assumptions will be stated more explicitly.
> > >     * But onto the more academic point of the question: Is there in principle a way beyond correlation/fit metrics or using samples of points? For example, is there any theory that could provably show that a finite-width neural network trained on finite samples with stochastic gradient descent is exactly equal to a kernel machine such that the kind of data attribution we study can still be performed? We would direct the reviewer to the works of Domingos et al 2020 and Bell et al 2023, which attempt to answer this question. Our understanding of their work is that there is in fact a neural tangent path kernel that can be used to form a kernel machine surrogate model that is exactly equivalent to the NN, but computing this kernel function requires computing the eNTK over each checkpoint from every gradient step in training. Therefore, the software advances we describe seem critical to computing this neural path kernel in the large model regime.
> > >
> > > References:
> > > Domingos, P., “Every Model Learned by Gradient Descent Is Approximately a Kernel Machine”, 2020. Arxiv
> > > Bell, B., Geyer, M., Glickenstein, D., Fernandez, A., and Moore, J., “An Exact Kernel Equivalence for Finite Classification Models”, 2023. Arxiv

---

> > > > ### Comment · Reviewer_dXvz · 2023-11-22
> > > >
> > > > I thank the authors for their careful answers to my questions. In light of this I am raising my score.

---

### Official Review · Reviewer_e28a · 2023-10-29

**Soundness:** 3 good
**Presentation:** 3 good
**Contribution:** 3 good
**Rating:** 8
**Confidence:** 4

**Summary:**

This paper proposes to use the approximate empirical neural tangent kernel (eNTK) as a faithful surrogate model for neural networks. Focusing on NNs for classification, the authors define the trace neural tangent kernel (trNTK), which is the cosine similarity between the concatenated gradients of all logits with respect to the parameters for a trained NN. The trNTK is then plugged into a kernel general linear model (kGLM) to obtain a surrogate model for the NN, which can be used to attribute the prediction of the NN to the training data points. To evaluate the faithfulness of such surrogate models, the authors argue that a preferred way is to measure the rank correlation between the softmax probabilities of the surrogate model and the NN for the correct class. A random projection variant of the trNTK is also proposed to reduce the computational cost. The experiments show that the proposed surrogate model is generally more faithful than other kernel-based surrogate models.

**Strengths:**

- The paper is clearly written and easy to follow.
- The proposed surrogate model is simple and easy to implement. trNTK performs consistently better than other neural kernels.
- The rank correlation seems to be a better metric than existing alternatives for evaluating the faithfulness of surrogate models for classification NNs. It takes into account the global structure of the predictions.
- Based on the proposed surrogate model and data attribution method, the authors observe that the attribution is NOT dominated by a few data points. This is an interesting observation and has practical implications.

**Weaknesses:**

- Only the rank correlation of the softmax probabilities for the **correct** class is considered. However, to be faithful enough, the surrogate model should also behave similarly to the NN for the **incorrect** classes. An important application of data attribution is to explain why a NN makes a wrong prediction. This is not considered in the paper.
- Eq. (4) is confusing. In the denominator, the $\cdot ^ {\frac{1}{2}}$ is applied to the inner product. However, according to Appendix C and the definition of cosine similarity, the $\cdot ^ {\frac{1}{2}}$ should be applied to the sum, not the inner product.
- The quality of Figure 2 could be improved.

**Questions:**

- Is the non-sparsity of the attribution a general phenomenon or just a property of trNTK? Is it a consequence of the statement "It has been suggested that this normalization helps smooth out kernel mass over the entire training dataset"?

---

> ### Author Response · Authors · 2023-11-21
>
> 1. We thank the reviewer for their insightful comments: we also believe that one of the main strengths of our paper is our use of the Kendall-$\tau$ metric because it allows us to go beyond test accuracy measures of similarity. This allows us to identify the trNTK as the most faithful kernel function. This in turn allows us to show that data attribution is actually not sparse. That conclusion could help move the field, as previous works have assumed sparsity in various data attribution/explain-by-example techniques.
> 2. Re: Only the rank correlation of the softmax probabilities for the correct class is considered...
>     * We agree with the reviewer that a current limitation of Kendall-$\tau$ is that it does not consider the logit value of incorrect classes. We think that an ideal faithfulness measure would include this information,  we will add a discussion of Kendall-$\tau$ the limitation of Kendall-$\tau$ to our limitation section; however, we feel despite our use of Kendall-$\tau$ is a significant step away from previous works that rely on test accuracy to measure the similarity of surrogate functions to the neural network.
> 3. Re: ... However, to be faithful enough, the surrogate model should also behave similarly to the NN for the incorrect classes.
>     * We agree with the reviewer's intuition. While this set of experiments have not appeared in the paper, we conducted an investigation into a "misclassification coincidence rate" as an additional line of evidence suggesting that the kGLM can mimic the behavior of the NN on misclassifications. The misclassification coincidence rate, $R_{\textrm{miss}}$, is the ratio of the number of times the kGLM and NN predict the same class out of all times either are incorrect over the number of times either the NN or the kGLM are incorrect
>     \begin{equation}
>     R_{\textrm{miss}} = \frac{ |\{ i : \textrm{kGLM}(\mathbf{x}_i) = \textrm{NN}(\mathbf{x}_i) \ne y_i \}| }{ | \{i : \textrm{kGLM}(\mathbf{x}_i) \ne y_i\} \cup  \{i : \textrm{NN}(\mathbf{x}_i) \ne y_i\}| }
>     \end{equation}
>     * This rate is actually very high (about 75\% coincidence of misclassifications on ResNet18 for trNTK), which is powerful evidence suggesting that the kGLM can mimic the NN overall for both correct classification and misclassifications. We will include this analysis in the revision.
> 4. Re: An important application of data attribution is to explain why a NN makes a wrong prediction. This is not considered in the paper.
>     * While our main body is primarily concerned on establishing our framework for data attribution and then an initial analysis re. the assumption of sparsity of the attribution scores, our framework can be extended to study misclassification. Our current status is to try and understand holistically from all the visualization generated why the NN made a misclassification by investigating the kGLM as proxy. Take as example figures 20, 31, and 42: In figure 20, we see that the mean plane and horse attribution score are highest driving the classification in their respective logits, but with the mean attribution from the remaining classes having higher negative impact on the plane logit. Figure 31 zooms into the horse logit for the trNTK, showing that the logit horse value is dependent upon the horse class. Even though our analysis shows that a sparse number of representations do not best describe the NN predictions, we can still leverage the most similar data points to help us form hypothesis of NN behavior. In figure 42, we show the most similar images are of people riding horses. This was quite the realization, since we had not noticed that the initial test image is actually of a person siting on top of an airplane! While we still must find a way to test this hypothesis, we believe that helping researchers form these hypothesis is a valuable tool in understanding misclassification.
> 5. Re: Eq. (4) is confusing. In the denominator, the sqrt is applied to the inner product. However, according to Appendix C and the definition of cosine similarity, the sqrt should be applied to the sum, not the inner product.
>     * We thank the reviewer for pointing out mistake; we will add parenthesis to Eq (4) and to ensure the equation matches the definition of cosine similarity.
> 6. Re: The quality of Figure 2 could be improved.
>     * We agree; there are rendering errors that we will fix in the revision and double check that boxes do not move in front of the axis labels.
>
> (continued below)

---

> > ### Author Response · Authors · 2023-11-21
> >
> > 7. Re: Is the non-sparsity of the attribution a general phenomenon or just a property of trNTK? Is it a consequence of the statement "It has been suggested that this normalization helps smooth out kernel mass over the entire training dataset"?
> >     * We thank the reviewer for the interesting question. To be succinct: it appears so. We provide some visualizations relevant to this question in figures 21 - 31 in the appendix. There, we compare the trNTK to the un-normalized trNTK. In the unnormalized trNTK we observe that the bulk of attributions are centered nearly on zero with a very small IQR, while the normalized trNTK has broader IQR and positive values mean attribution. This suggests that the unnormalized trNTK is in fact using a few exemplars in its kernel machine to make its classification (i.e., a few exemplars make up the majority of the attribution mass in the unnormalized NTK). Our non-sparsity result is a consequence of the fact that the faithfulness of the cosine-normalized trNTK as measured by Kendall-$\tau$ is higher (substantially so for some models) than the un-normalized trNTK.

---

> > > ### Comment · Reviewer_e28a · 2023-11-22
> > >
> > > Thanks for your detailed response. I have updated my rating accordingly.

---

### Official Review · Reviewer_zBfF · 2023-11-02

**Soundness:** 2 fair
**Presentation:** 2 fair
**Contribution:** 2 fair
**Rating:** 5
**Confidence:** 4

**Summary:**

The paper proposes new variants of eNTK and implements faster approximate versions as well, and then evaluates them on a few different tasks / visualizations.

**Strengths:**

- Paper evaluates variants of eNTK in further depth compared to prior work
- Paper is relatively well written, though missing some important details

**Weaknesses:**

- Content: Surfacing similar images is a not a meaningful evaluation of attribution. It is a good sanity check, but doesn't say anything about surrogacy. For example, finding similar images using CLIP similarity would also show similar images, though CLIP is in no way a "surrogate" to the model being studied

- More broadly, I'd be more careful about making any claims of "data attribution" (which has a specific meaning as used in recent ML) as the paper does carry out any counterfactual evaluations.

- Overall, the contributions seems somewhat marginal. Also, the fast approximate versions implemented primarily rely on prior work (Park et al.)'s implementation, so not sure there is much to claim as contribution there (since Park et al. also used it for faster approximations to eNTK).

- Writing: is hard to follow at times and doesn't provide the relevant details (see Questions).
On one hand, the paper goes into more detail than necessary in defining rank correlation / R2, etc from scratch,
and at the same time, doesn't actually provide details about what those measures are computed over exactly.
It's possible I missed it, but at least doesn't seem very clearly written based on my multiple attempts to parse this information.

**Questions:**

- Confused by what the rank correlation is measured over exactly. I understand it's measured between the truth model outputs and surrogate model outputs, but what is it varied over? Are you measuring across different inputs x?
- A bit confused by what the message/takeaway of the box/distribution plots are. Can the authors elaborate?
- It seems that the eNTK is only defined in Appendix D, so it's a bit hard to contextualize pNTK and trNTK when they are introduced

---

> ### Author Response · Authors · 2023-11-18
>
> 1. We thank the reviewer for their thoughtful review. Part of the importance of our work is to explore the space of scalable approximations to the eNTK that retain intrinsic properties of the eNTK. Our hope is that our research will enable eNTK research on larger datasets and models. As an example of the analysis this software enables, we perform investigations into data attribution problems using kernel surrogate models on large computer vision and language models. We think the software will enable many new analyses and are excited to see how other researchers build off our work.
> 2. Re: Surfacing similar images is a not a meaningful evaluation of attribution.
>     * We wholeheartedly agree with the limitations of surfacing similar images as the reviewer states --- in fact, this is a major conclusion we draw in the paper. See the end of section 5 paragraph 1: "Therefore, presenting the top highest attribution training images without the context of the entire distribution of attribution is probably misleading." Our kernel machine surrogate framework goes beyond simply surfacing similar images to show how the surrogate model, which is deeply correlated with the underlying neural network, leverages **all of the training data** to make classification on new test data.
> 3. Re: I'd be more careful about making any claims of data attribution ... as the paper does carry out any counterfactual evaluations.
>     * We acknowledge the historical context of data attribution in classical statistics, and recent research in data attribution that have utilized counterfactual experiments; however, it is not the only sense in which "data attribution" is used. We subscribe to the framework given in Park et al 2023 (TRAK): "a data attribution method computes a score for each training input indicating its importance to the output of interest." Specifically, they cite the approach of Yeh et al 2018 (i.e. Representer Points) as a form of data attribution, and we mirror the same sense of data attribution in our work. We describe how we compute our data attribution score in great detail in the comment to all reviewers.
> 4. Re: Overall, the contributions seems somewhat marginal. Also, the fast approximate versions implemented primarily rely on prior work (Park et al.)'s implementation, so not sure there is much to claim as contribution there (since Park et al. also used it for faster approximations to eNTK).
>     * We politely but strongly disagree: Park et al. (TRAK) neither compute the eNTK nor any approximation of the eNTK. While Park et al. do reference the eNTK and devote a section to show how it is a part the TracIn kernel, the actual calculation of the eNTK itself is not implemented in their code. You can view the relevant lines of code from TRAK [here](https://github.com/MadryLab/trak/blob/main/trak/modelout\_functions.py\#L84]).  Notice that the model output functions implemented are only the loss, rather than the neural network function itself. We have been in direct communication with the maintainer of TRAK to bring the capabilities we describe in this work to the TRAK software package.
>     * It is true that our fast approximation would not be possible without TRAK (and so  TRAK rightfully takes a prominent position in our paper); however, building off each other's work is a success story of the open-source community and is at the foundation of AI research. We see this as a major strength rather than a weakness.
> 5. Re. Writing ... the paper goes into more detail than necessary in defining rank correlation / R2, etc from scratch...
>     * We thank the reviewer for pointing out the confusion. We agree that the space devoted to the Kendall-$\tau$ definition could be saved by moving it to the appendix. We would instead use this space to discuss the motivation for the Kendall-$\tau$, which we believe is more important than discussing why Pearson is flawed. If it interests the reviewer, we provide additional information in the comment to all reviewers about our argument for why we use Kendall-$\tau$.
> 6. Re: Confused by what the rank correlation is measured over exactly. I understand it's measured between the truth model outputs and surrogate model outputs, but what is it varied over? Are you measuring across different inputs x?
>     * Yes: we are varying it over the different model inputs x from the test dataset. For cases beyond binary classification (single logit), we also must make a choice for which output of the neural network/ surrogate model to use. We chose to use the output corresponding to the correct class (as given by the ground-truth labels).
>
> (continued in next comment)

---

> > ### Author Response · Authors · 2023-11-18
> >
> > 7. Re: A bit confused by what the message/takeaway of the box/distribution plots are. Can the authors elaborate?
> >     * Following are the key takeaways.
> >     * In comparison to explain-by-example techniques that surface a few images, we demonstrate a visualization that shows how we use the distribution of all attribution scores. This is important because by considering the entire distribution we have not made the assumption that attribution is sparse a priori.
> >     * We want to evaluate whether the sparsity assumption is valid. Given that the mean of the attribution score falls within the interquartile range across all training datapoints, we know there is not an extremal point attribution significantly moving the average. In effect: we know that none of the outliers are dominating the attribution, but rather, the bulk properties of attribution from all training points are responsible for the model output.
> >
> > 8. Re: It seems that the eNTK is only defined in Appendix D, so it's a bit hard to contextualize pNTK and trNTK when they are introduced
> >     * We thank the reviewer for pointing this out, and in the revision we will add an inline equation to define the eNTK to provide some additional context for the reader.

---

### Official Review · Reviewer_yHxi · 2023-11-04

**Soundness:** 3 good
**Presentation:** 3 good
**Contribution:** 3 good
**Rating:** 8
**Confidence:** 4

**Summary:**

This paper investigates kernel-based surrogate models based on various approximations of the Neural Tangent Kernel (NTK) to provide explanations for deep neural networks. A primary contribution is showing that computationally-feasible approximations to the empirical NTK provide high-fidelity surrogate models, and that much cheaper projection-based approximations provide accurate estimates of the empirical NTK. Appealing to existing literature on explanation-by-example, the paper develops a simple score for data attribution. A synthetic data experiment shows that the proposed attribution score accurately attributes erroneous model predictions to poisoned data, giving some confidence that the proposed score is capturing some notion of similarity between data points.

**Strengths:**

* The paper provides a potential solution to a very important problem, i.e. data attribution based on a trained neural network checkpoint.
* The authors draw a very important distinction—which is somewhat obvious but not well-reflected in the literature—between difference in test accuracy (TAD) and correlation of model outputs. The addition of Kentall-$\tau$ is important, and will hopefully shift the way future works evaluate the fidelity of surrogate models.
* The paper provides a clear definition of proposed kernel approximations, with strong computational justification for the speedup e.g. of the trace approximation. Equation 2 clearly defines the working definition of a “high-fidelity model” and adds to the clarity of exposition.
* The experiments include a sufficient diversity of alternative kernel estimates to demonstrate the value of the proposed projected-trace-NTK approach. The inclusion of uncertainty based on multiple runs is highly appreciated.
* The inclusion of experiments on Bert-base significantly strengthens the paper, indicating that the method is not specific to the computer vision domain.
* The discussion that explanations are not sparse is an important acknowledgement of the proposed data attribution method. In particular, the following statement is poignant: “presenting the top highest attribution training images without the context of the entire distribution of attribution is probably misleading.”
* The paper’s title is very strong and well reflects the work’s primary contributions.

**Weaknesses:**

* The paper relies on previous work to establish credibility of attribution-based scores for neural network explanation. It doesn’t seem obvious that attribution is the same as similarity for learned kernel functions.
* I find the second sentence in the abstract confusing. I expected this trend to have to do with using kernel-based models for data attribution rather than to “investigate a diverse set of neural network behavior”. Isn’t the goal of your paper exactly to apply kernel models to investigate network behavior?
* The 3rd experiment on qualitative evaluation of attribution is weak. A user study is probably beyond the scope of this paper, and I believe the work is strong enough to stand without such a study. However, the paper would significantly benefit from some discussions about how these attributions could be better qualitatively evaluated in the future.
* The claim about Peason correlation is not very well explained: “These point clouds serve as anchors that force the covariance, and therefore Pearson correlation, to be large. We require a measure that does not conflate the covariance with faithfulness.” Is the problem here that correlation is not computed between model logits for each test point?
* The paper never explicitly defines the empirical NTK in its own notation. Could you add this prior to defining the trNTK or pNTK in order to allow an easier discussion of the approximations introduced?
* The take-away from Figure 2 is not exactly clear. Is this just meant to show that attribution scores are not sparse?
* Your Chen ICML’22 reference is duplicated. Did you intend to cite two different papers?
* The notation is non standard. Most papers use $y$ not $z$ for ground-truth labels. I can see this causing some readers mild confusion.

Small issues:
* In the last paragraph of the “Relationship to the Pseudo Neural Tangent Kernel” section, the reference to Eq 3 is to the wrong equation.
* The inclusion of all four panes in Figure 1 seem a bit superfluous, I’m not sure what this is supposed to show that cannot be shown in a single figure.
* Given that one of the 3 experiments claimed in the paper is a qualitative evaluation, I think it is important to include one of the data attribution figures from the supplemental material in the main text.
* The main text cites Figure 4.a, which is in Supplemental Material.

Small typos:
* 2nd paragraph of the Introduction: “Its well established” should be “it’s” (or better yet spell out “it is” to be less colloquial.
* Undefined reference (?) at the bottom of page 2.

**Questions:**

1. Have you considered using the kernel function directly to evaluate sample similarity? Why do you choose to include the weights from the kernel machine in all attribution scores?
2. You fit the parameters $W, b$ on the ground-truth labels $z$ from the  training dataset. If the goal is to create a surrogate that emulates a neural network, why don’t you fit these parameters on cross-entropy loss with the class probabilities predicted by the NN? This would be consistent with your objective in Eq. 2.
3. It is not totally clear how the Kendall-$\tau$ statistic is computed. A couple of sentences would make this portion more reproducible. Do you take the matrix of all the logits produced on a test set ($N x C$) for the NN and for the surrogate model, flatten these two matrices into vectors, and compute the rank correlation? Is the rank correlation computed per-test-output and averaged over the test set?
4. Why is trNTK initially introduced with cosine normalization and projNTK not? Don’t your experiments include cosine normalization for all kernels?
5. Is there any theoretical statement you can add about the variance of the projection-based kernel estimates, e.g. based on JLS? The choice of 10240 dimensions seems arbitrary and model-dependent.
6. Does the introduction of cosine normalization explain the experimental result “that the highest attributed images from the trNTK (and furthermore all evaluate kernel functions) have relatively small mass compared to the bulk contribution, suggesting that the properties of the bulk”?

---

> ### Comment · Reviewer_yHxi · 2023-11-20
> **Question 2 is important**
>
> I would like to know the answer to my Question 2. It seems like a surrogate model should be fit with the outputs of the model it is emulating, not with ground-truth labels. I doubt this change would alter the results in the paper. But I think the authors should consider making this chance, if feasible, for the final paper draft.

---

> ### Author Response · Authors · 2023-11-20
>
> Good Morning,
>
> We aim to post a thread of answers to each of your points today. To answer this specific question now: it is a very reasonable suggestion. The reason we did not attempt this is we were worried about overfitting the kGLM onto the NN's outputs, so we investigated using the ground truth labels first. After finding that the fits were very reasonable, we moved ahead. We were surprised that even without using the NN's outputs there would be such high alignment between the NN and the kGLM.
>
> We now think that even if the model were to overfit onto the NN's outputs, this would become apparent when evaluating Kendall-$\tau$ on the test data, so overfitting would be observable and therefore not represent a great risk. In addition, we recently became aware of some contemporaneous work that was accepted at Neurips23 [Tsai et al 2023] that utilize the NN's outputs to derive ideal weights.
>
> Due to all of this, we resolve to include this experiment in our final paper and compare to the existing results. We think it would add an interesting comparison to Tsai et al.'s methodology.
>
> Tsai, C.-P., et al., “Sample based Explanations via Generalized Representors” 2023

---

> ### Comment · Reviewer_yHxi · 2023-11-20
>
> I don't understand the concern with "overfitting". Wouldn't you expect a reasonable surrogate to at least match the NN exactly on training points?
>
> I appreciate the authors' willingness to add experiments training the surrogate to directly emulate the NN. I look forward to reading the full rebuttal.

---

> > ### Author Response · Authors · 2023-11-21
> >
> > To answer the question directly, yes, we should expect a reasonable surrogate to match on the training data points. However, we need our surrogate to also match on new data inputs as well to have useful predictive power. The concern is that we could fit our kGLM to exactly match the NN outputs on the training data points, but it may not be a good surrogate when evaluated on the test data. That would be the sense of overfitting we meant in our response.
> >
> > In hindsight we should have simply evaluated whether the kGLM over-fits on the NN's training data outputs from the beginning. We will include an experiment comparing the results of training on the NN outputs vs ground truth labels. It is nonetheless interesting that using the ground truth labels we can find a kGLM that mimics the NN behavior. We speculate there might be something to using the ground truth labels as this mimics the optimization problem that the NN is trained on to the kGLM. In any case, we believe including the comparison makes the work stronger.

---

> ### Author Response · Authors · 2023-11-21
>
> 1. We thank the reviewer for their time and appreciate the breadth of topics covered. They have identified what we consider to be the major strength of this paper: bringing in new ideas to modeling surrogacy and data attribution: i.e., challenging both the use of test accuracy to measure the similarity of models as well as the a priori assumption that neural network classifications can be explained using a sparse number of training data exemplars.
> 2. Re The paper relies on previous work to establish credibility of attribution-based scores for neural network explanation.
>     * We have expanded upon our argument to establish the credibility of our attribution scores in the comment to all (point 2). We resolve to present this in the revision so our work is more self-contained.
> 3. Re It doesn’t seem obvious that attribution is the same as similarity for learned kernel functions.
>     * In Eq (3) we describe how attribution is the product of the kernel similarity with the learned weights, so is \textbf{not} just the similarity. We expand upon this in the comment to all reviewers as well (point 3), since it was a shared confusion. We will be more explicit in our revision.
> 4. Re I find the second sentence in the abstract confusing. I expected this trend to have to do with using kernel-based models for data attribution rather than to “investigate a diverse set of neural network behavior”. Isn’t the goal of your paper exactly to apply kernel models to investigate network behavior?
>      * Our explicit goal is to use kernel methods as a surrogate for neural networks, and the application we focus on in this paper is data attribution. We agree that our attempt to remain general in the second sentence of the abstract reads overly pedantic for the sake of the narrative device used in the third sentence. We will address this sentence in the revision.
> 5. Re The 3rd experiment on qualitative evaluation of attribution is weak. A user study is probably beyond the scope of this paper, and I believe the work is strong enough to stand without such a study. However, the paper would significantly benefit from some discussions about how these attributions could be better qualitatively evaluated in the future.
>     * We share a preference for stronger quantitative evaluation as well; however, we believe the qualitative evidence we have collected is still important to share because researchers often generate new hypotheses from looking at the data in a new qualitative way.
>     * We seek to publish this result without human-subjects testing. We will add a short discussion of the limitations of explainability and caution the reader that a user-study must be performed in the revision.
>
> (continued below)

---

> > ### Author Response · Authors · 2023-11-21
> >
> > 6. Re: The claim about Pearson correlation is not very well explained...
> >     * We thank the reviewer for pointing out the lines. The reviewer is correct that Pearson correlation can change depending on whether the sigmoid/softmax is used, but this is not the meaning of our point cloud discussion. For the reviewer's convenience we copy the discussion from our comment to all below and explain why Pearson is not a good choice of faithfulness measure. This explanation should give better context to why we discuss the point cloud toy example.
> >
> >    * To begin: Pearson correlation measures the ratio of the covariance between a set $\mathcal{X}$ and $\mathcal{Y}$ to the product of the square root of the variance in each set. Kendall-$\tau$ measures the degree to which the sets are monotonic with one another, irrespective of the functional form of any mapping between sets.
> >
> >    * Consider the following toy scenario: imagine there are two independent classification models that achieve nearly 100\% accuracy. Furthermore, the output of these models is a normally distributed random variable with $\mathcal{N}(\mu_1,\sigma_1)$ if the true label is 1 and $\mathcal{N}(\mu_0,\sigma_0)$ if the true label is 0. If $\sigma_0,\sigma_1$ are small compared to $\mu_0-\mu_1$, then the Pearson correlation will be nearly 1.
> >
> >    * Intuitively, this is because this describes point clouds centered at $\mu_0$ and $\mu_1$, and if those point clouds have a high degree of separation relative to their variance each random variable has in each point cloud ($\sigma_0$ and $\sigma_1$) then the covariance will be nearly equal to the variance of each random variable. This means that the Pearson correlation will be 1. We want a measure of the dependence between two random variables, and we have now shown that Pearson can result in value 1 even if the variables are explicitly not dependent on one another.
> >
> >    * While simple, this thought experiment is revealing. We have two point clouds at $\mu_1$ and $\mu_0$. In Appendix I.1 (Figure 7) we can see that the point cloud structure of the logit observations mimic this case. To help explain this more concretely, we provide a Jupyter notebook file with these experiments in the supplementary work to demonstrate our arguments.
> >
> > 7. Re: The paper never explicitly defines the empirical NTK in its own notation. Could you add this prior to defining the trNTK or pNTK in order to allow an easier discussion of the approximations introduced?
> >     * We will add in the eNTK equation for the revision.
> > 8. The take-away from Figure 2 is not exactly clear. Is this just meant to show that attribution scores are not sparse?
> >     * Figure 2 is meant to convey the following ideas a) how we do not assume attribution is sparse a priori, so we look at the attribution distribution from all training points; b) because we do not assume sparsity, we can check whether sparsity emerges naturally. We do not find any evidence supporting the assumption of sparsity since the outliers are not extreme enough in comparison to the bulk mass of attribution in each logit for the trNTK.
> > 9. Re: Your Chen ICML’22 reference is duplicated. Did you intend to cite two different papers?
> >     * We thank the reviewer for pointing out the mistake and will fix in the revision. We did not intend to cite two different versions of the same paper.
> > 10. Re: The notation is non standard. Most papers use y not z for ground-truth labels. I can see this causing some readers mild confusion.
> >     * We wanted to be very clear to distinguish between a logit and a probability vector; since, we expected many readers would confuse whether the final softmax/sigmoid activation function are being included in our NTK computation. This motivated our use of X$\rightarrow$Y$\rightarrow$Z since the alphabet order matches the order of NN$\rightarrow$logit$\rightarrow$probability vector. We will emphasize the break from standard notation in the revision.
> > 11. Re: Have you considered using the kernel function directly to evaluate sample similarity?
> >     * We have, yes. While the weights are used in our definition of data attribution, we actually do not use the weights to surface the most similar images, (figures 31-41, in Appendix J.2). We observed that visualizing the most similar vs the highest attribution images often result in qualitatively more shared concepts in the image. For example, if the test image is a picture of a horse's head looking left, the most similar training images will be of horses looking left; while the highest attribution images might more varied pictures from the horse class.
> >
> > (continued below)

---

> > > ### Author Response · Authors · 2023-11-21
> > >
> > > 12. Re: Why do you choose to include the weights from the kernel machine in all attribution scores?
> > >     * Our data attribution definition is built on the premise that a kernel machine can be found that is (nearly) equivalent to the NN. Even if we were to attempt to use the kernel function alone, our formulation would demand some kind of operator to reduce the kernel function over each training datapoint to a single number representing the NN's logit. The kernel weights and matrix multiplication serve as this operator.
> > > 13. Re: It is not totally clear how the Kendall-$\tau$ statistic is computed...
> > >     * We apologize for the confusion. We compute the Kendall-$\tau$ over the vector of created from correct class logit values (given by the ground truth labels) of the kGLM and NN, evaluated on the entire test dataset.
> > > 14. Re: Why is trNTK initially introduced with cosine normalization and projNTK not? Don’t your experiments include cosine normalization for all kernels?
> > >     * We think the reviewer might be referring to the pNTK instead of either the proj-pNTK or proj-trNTK? We introduce the pNTK without the normalization because we mean to reproduce the pNTK exactly as from Mohammadi et al. 2022. We do not perform any experiments on the pNTK. When we compute the proj-pNTK we do apply the cosine-normalization. **We use the cosine-normalization for all kernels except for the unnormalized-trNTK, or** $\text{trNTK}^0$
> > > 15. Re: Is there any theoretical statement you can add about the variance of the projection-based kernel estimates, e.g. based on JLS? The choice of 10240 dimensions seems arbitrary and model-dependent.
> > >     * We assume that the reviewer is referring to the Johnson-Lindenstrauss Lemma. On first inspection, we believe it should be straightforward to use the JL Lemma to  show that each inner product in the original and projected kernels are close (in a multiplicative sense, if appropriately normalized). This in turn implies that the (empirical) distributions of entries in the NTK and projection-NTK are close. Because the projection dimension can be chosen to make this bound small, we can partially control the closeness of various statistics of the distributions (including the variance) by choosing the projection dimension. If the reviewer thinks it will better justify the choice of projection dimension, we can try to add (or provide a sketch) of such a result in the final version.
> > >     * In the submitted version of the manuscript, we treated the projection dimension as a hyperparameter selection task specific to each model, where there is a trade-off in the projection dimension size between compute time and variance of the residual distribution. For our resources, 10240 created a reasonable compromise for both Bert and ResNet18, but we will be more explicit in the revision that this choice is of simple convenience.
> > > 16. Re: Does the introduction of cosine normalization explain the experimental result “that the highest attributed images from the trNTK (and furthermore all evaluated kernel functions) have relatively small mass compared to the bulk contribution, suggesting that the properties of the bulk”?
> > >     * We believe this comment is correct: by comparing the trNTK and $\text{trNTK}^0$ we see that the $\text{trNTK}^0$ has much smaller scale of IQR to the most extreme data attribution points relative to the trNTK's. This result is highly suggestive that the cosine normalization helps the trNTK smooth out the attribution amongst all training points. This fact is not reflected in the quoted statement in the reviewer's question, so we will modify it in the revision.

---

### Author Response · Authors · 2023-11-18
**Comment to All re. common points brought up by reviewers:**

1. We thank the reviewers for their time and for all of the constructive feedback we have received. We saw some common themes between the reviews which we would like to address before proceeding to responses to individual reviews. We hope this will provide some common context that would be of interest to all reviewers.

2. What do we mean by ``data attribution'' and why do we discuss it?
    * The question we are interested in is the following: given a model and an output of interest, how important is the individual training point $\boldsymbol{x}_i$ to the prediction of the trained model on an an input $\boldsymbol{x}$?. This follows the definition of the ``data attribution task'' given in Park et al 2023.

    * "Data attribution" has specific meaning in classical statistics in the context of linear regression that involves the inversion of the gram matrix [Hampel 1974]. This inversion becomes computationally infeasible for the regime of deep learning models we seek to develop methods for analyzing. The framing of this classical result is in the context of a leave-one-out analysis, also referred to as a "counterfactual" experiment. Recent literature in the context of performing data attribution for deep learning models rely on these counterfactual experiments [Koh and Liang, 2017][Park et al. 2023][Ilyas et al., 2022], though often seek innovative ways around the matrix inversion. While we acknowledge counterfactual experiments are closely related to the solution of data attribution through classical results and recent AI literature, they do not make up the complete set of possible data attribution methodologies. We in fact investigate an alternative data attribution methodology that does not use counterfactuals.
     * To reiterate, we subscribe to the broad definition of "data attribution" from Park et al 2023: ``A data attribution method computes a score for each training input indicating its importance to the output of interest.'' We return a score that describes how important the data point is to the output of the neural network function. To accomplish this, we use the formula in (3) of our paper. Note that the kGLM prediction naturally decomposes into a summation of terms that each involves a **single** data point of the training set:
\begin{equation*}
	\text{kGLM}(\mathbf x)^c
    =\mathbf W^c \mathbf \kappa(\mathbf x, \mathbf X)
        +\mathbf b_c
    =\sum_{i=1}^N \left(
        \mathbf W_{c,i} \boldsymbol{\kappa}(\mathbf x,\mathbf x_i)
        +\frac{\mathbf b_c}{N}
        \right)
	   \end{equation*}
    Our definition is then
    \begin{equation*}
     A(\mathbf x, \mathbf x_i)^c =  \mathbf W^c_i \boldsymbol{\kappa}(\mathbf x,\mathbf x_i)+\frac{\mathbf b^c}N.
    \end{equation*}

Thus for us, the attribution $A^c(\mathbf x,\mathbf x_i)$ of the $i$-th data point is defined as the corresponding term in this summation. Notice that the set of attributions $\{A(\mathbf x,\mathbf x_i)\}$ sums to the prediction of the kGLM for class $c$, i.e. **the set of attributions fully account for the prediction of the model.**

(continued in next comment)

---

> ### Author Response · Authors · 2023-11-18
>
> 3. How is "data attribution" different from "similarity"?
>     * Multiple reviewers asked about the connection between our data attribution and "similarity" between data input $\boldsymbol{x}$ and training input $\boldsymbol{x}_i$. We point out that our definition of data attribution *is not equivalent to similarity*. We interpret $\boldsymbol{\kappa}(\mathbf x,\mathbf x_i)$ as measuring how similar $\mathbf{x}$ and $\mathbf{x}_i$ are. Our definition of the data attribution score is given in the draft as Eq 3 and above. Measured between a data input $\boldsymbol{x}$ and training datapoint $\boldsymbol{x}_i$, the attribution is:
>
>     \begin{equation}
>          A(\mathbf x, \mathbf x_i)^c =  \mathbf W^c_i \boldsymbol{\kappa}(\mathbf x,\mathbf x_i)+\frac{\mathbf b^c}N.
>     \end{equation}
>
>     * A kernel function $\boldsymbol{\kappa}(\mathbf x,\mathbf x_i)$ computes the inner product between $\mathbf{x}$ and $\mathbf{x}_i$ in an appropriately-defined vector space. While the similarity is encoded in the kernel function's value, $\kappa(\boldsymbol{x},\boldsymbol{x}_i)$. In short, the learned weights can and do moderate the similarity for the attribution score.
>
> 4. Why is a rank based correlation measure a good way to measure "faithfulness" and  why do we specifically use Kendall-$\tau$?
>     * We begin by considering three properties a faithfulness measure should exhibit and will explain how these are satisfied by non-parametric rank correlation like Kendall-$\tau$.
>     * Recall our ideal objective is to evaluate to what degree a kernel machine surrogate model is equivalent to the NN at points of interest $\boldsymbol{x} \in \mathcal{X}$:
>     \begin{equation}
>     \text{NN}(\boldsymbol{x},\; \boldsymbol{\theta}) = \sum_{i=1}^N{W_i K(x,x_i)} + \boldsymbol{b}
>     \end{equation}
>     * We only focus on whether the kernel machine is a good surrogate when $\boldsymbol{x}$ is drawn from the population $\mathcal{X}_{\text{test}}$. Because we do not know the distribution of the test population so we are forced to sample from it, leading to the first property. **The first property:** our metric should be a single number we can calculate from the samples that synthesizes the information across each of the samples.
>     * **The second property:** the faithfulness standardized so as to be comparable across different surrogate models. For example, it should have a bounded maximum.
>     * **The third property:** the measure of faithfulness should be invariant and maximized for the following relationship over any invertible mapping function $\Phi$: \begin{equation}
>     \text{NN}(\boldsymbol{x}_i) = \Phi(kGLM(\boldsymbol{x}_i))
>     \end{equation}
>     * Note that the assumption that $\Phi$ is an invertible map means that it preserves any monotonic relationship between $\text{NN}(\mathbf{x}_i)$ and $\text{kGLM}(\mathbf{x}_i)$. Invertible maps are injective maps. It is this fundamental injective property that makes the kGLM have ``one-to-one'' correspondence with the NN that we care about.
>     *  These three properties suggest we use a rank based correlation. First, the correlation will reduce to a single number over the set of samples across which we evaluate. Second, the correlation will be bounded in [-1,1] so that values of correlation are easily comparable across kernels. Third, rank-based correlation are invariant under invertible transformations.
>     * Both Kendall-$\tau$ and Spearman-$\rho$ satisfy these properties.  We chose to use Kendall-$\tau$ over Spearman-$\rho$ because Spearman-$\rho$ is generally larger than Kendall-$\tau$ and is observed to converge faster to 1 as the error between NN and KGLM approach 0 [Fredericks and Nelson, 2007]. We provide a notebook in the supplementary materials to verify this property. It is desirable to have our faithfulness measure go to 1 slowly so that our measurement distinguishes between surrogate functions over a larger range of errors.
>     * **PROPOSED CHANGE:** Because these motivations are not clearly elucidated in the draft, we would include a discussion about this in our revision. We will also add a discussion of the limitations of Kendall-$\tau$
>
> (continued below)

---

> > ### Author Response · Authors · 2023-11-18
> >
> > 5. Specifically give an example where Pearson is misleading and Kendall-$\tau$ is not.
> >     * To begin: Pearson correlation measures the ratio of the covariance between a set $\mathcal{X}$ and $\mathcal{Y}$ to the product of the square root of the variance in each set. Kendall-$\tau$ measures the degree to which the sets are monotonic with one another, irrespective of the functional form of any mapping between sets.
> >     * Consider the following toy scenario: imagine there are two independent classification models that achieve nearly 100\% accuracy. Furthermore, the output of these models is a normally distributed random variable with $\mathcal{N}(\mu_1,\sigma_1)$ if the true label is 1 and $\mathcal{N}(\mu_0,\sigma_0)$ if the true label is 0. If $\sigma_0,\sigma_1$ are small compared to $\mu_0-\mu_1$, then the Pearson correlation will be nearly 1.
> >     * Intuitively, this is because this describes point clouds centered at $\mu_0$ and $\mu_1$, and if those point clouds have a high degree of separation relative to their variance each random variable has in each point cloud ($\sigma_0$ and $\sigma_1$) then the covariance will be nearly equal to the variance of each random variable. This means that the Pearson correlation will be 1. We want a measure of the dependence between two random variables, and we have now shown that Pearson can result in value 1 even if the variables are explicitly not dependent on one another.
> >     * In contrast, Kendall-$\tau$ returns a lower bound value of 0.5 in this toy model (assuming equal class sizes). So while Kendall-$\tau$ is still sensitive to the failure mode affecting Pearson correlation, with his probability Kendall-$\tau$ does not reach its maximum, so can still be a useful tool to distinguish between independent and dependent functions.
> >     * While simple, this thought experiment is revealing. We have two point clouds at $\mu_1$ and $\mu_0$. In Appendix I.1 (Figure 7) we can see that the point cloud structure of the logit observations mimic this case. To help explain this more concretely, we provide a Jupyter notebook file with these experiments in the supplementary work to demonstrate our arguments.
> >
> > 6. Re Kendall-$\tau$, over what precisely is it computed?
> >     * We compute the Kendall-$\tau$ over the set formed from the $\{\text{NN}^{c=\text{corr}}(x_i)\}$ for all data inputs $\boldsymbol{x_i}$ in $\boldsymbol{X}_{\text{test}}$ and $\{\text{kGLM}^{c=\text{corr}}(\boldsymbol{x}_i)\}$, where in both cases corr signifies that the correct class (given from the true labels) is being used to pick out the correct class' logit.
> >      *  **One reviewer specifically asked why we do not include the effects of the other classes.** We believe this is a great idea for follow on work and agree it is a current limitation of Kendall-$\tau$. Kendall-$\tau$ (as we use it) does not consider all the classes' logits value simultaneously. We do not add all logit values to the Kendall-$\tau$ calculation for the following reason: for many logits representing irrelevant class to the classification, the attribution distribution is 0-centered, so the probability expressed is often very small. Since in actuality the rank-order of these very improbable class probabilities does not matter much to the classification, we felt including these effects buries the Kendall-$\tau$ measurement in the noise of unimportant concordant-discordant logits of irrelevant classes.
> >
> >     * **PROPOSED CHANGES:** We will amend the limitation section with a discussion on the current limitations of Kendall-$\tau$ to motivate more work into a better faithfulness measure, but believe the work represents a strong contribution to the field as is: multiple reviewers point out that using correlation instead of test accuracy to measure faithfulness is a major strength. While we may not have "solved" the problem, this idea represents significant progress.
> >
> > 7. We would like to take this opportunity to delineate our contributions from that of Park et al 2023 (TRAK).
> >     * To begin, we want to acknowledge that TRAK is very influential software that is at the foundation of our projection kernel functions, so it rightly takes a prominent role in our paper. TRAK references the eNTK and devotes a section to show how it is an explicit part of the TracIn kernel, but its actual calculation is not implemented in their code. You can view the lines of code [here](https://github.com/MadryLab/trak/blob/main/trak/modelout_functions.py#L84). Notice that the model output functions implemented are only the loss (as this serves their use case for counterfactual analysis), rather than the neural network function itself. TRAK does not compute the eNTK or any approximation of the eNTK (p-, tr-, etc.). We have been in direct communication with the maintainer of TRAK to bring the capabilities we describe in this work to the TRAK software package.
> >
> > (continued below)

---

> > > ### Author Response · Authors · 2023-11-18
> > >
> > > 8. Additional changes involving table 4 and poisoning proj-trNTK experiment
> > >     * In our data poisoning experiments, a mis-specified path for the trNTK incorrectly pointing pNTK experiments went unobserved. This causes two changes: First, table 4, the data-poisoning trNTK column needs to be increased from 8h to 50h. This does not modify our conclusions that the proj-trNTK has a much lower computational burden. Second, figures in appendix K are updated so that the trNTK is used to identify the top most similar examples. On a visual inspection, this actually makes the methods seem (qualitatively) more robust: in each experiment, it is observed that most similar examples surfaced by the proj-trNTK match the trNTK more closely, which should be expected.
> > > 9. In preparing our draft a copy-paste error replaced the captions for some figures in our appendix and went unnoticed. We will update and provide more information in the captions for the reader.
> > >
> > > Frank R. Hampel (1974) The Influence Curve and its Role in Robust Estimation, Journal of the American Statistical Association, 69:346, 383-393, DOI: 10.1080/01621459.1974.10482962
> > >
> > > Gregory A. Fredricks, Roger B. Nelsen (2007), On the relationship between Spearman's rho and Kendall's tau for pairs of continuous random variables, Journal of Statistical Planning and Inference, Volume 137, Issue 7
> > >
> > > Koh, P. W. and Liang, P., “Understanding Black-box Predictions via Influence Functions”, arXiv e-prints, 2017. doi:10.48550/arXiv.1703.04730.
> > >
> > > Ilyas, A., Park, S. M., Engstrom, L., Leclerc, G., and Madry, A., “Datamodels: Predicting Predictions from Training Data”, arXiv e-prints, 2022. doi:10.48550/arXiv.2202.00622.
> > >
> > > Park, S. M., Georgiev, K., Ilyas, A., Leclerc, G., and Madry, A., “TRAK: Attributing Model Behavior at Scale”, arXiv e-prints, 2023. doi:10.48550/arXiv.2303.14186.

---

### Meta-Review · Area_Chair_5ak6 · 2023-12-07

**Metareview:**

The paper tackles the hot topic of explaining a neural network, using neural tangent kernels (NTK). More precisely, a major contribution of the proposed approach is to define a tractable approximation of the NTKs.

The reviewers and the area chair appreciated the value and originality of the work. The paper is great, and the very detailed rebuttal is great, too.

**Justification For Why Not Higher Score:**

The paper nicely and efficiently tackles a hot topic; I do not recommend an oral presentation as I have another, more "game changing" paper in my lot.

**Justification For Why Not Lower Score:**

-

---

### Decision · Program_Chairs · 2024-01-16

Accept (spotlight)